# QUANTILE-FREE UNCERTAINTY QUANTIFICATION IN GRAPH NEURAL NETWORKS

## ABSTRACT

Uncertainty quantification (UQ) in graph neural networks (GNNs) is crucial in high-stakes domains but remains a significant challenge. In graph settings, message passing often relies on strong assumptions such as exchangeability, which are rarely satisfied in practice. Moreover, achieving reliable UQ typically requires costly resampling or post-hoc calibration. To address these issues, we introduce Quantile-free Prediction Interval GNN (QpiGNN), a framework that builds on quantile regression (QR) to enable GNN-based UQ by directly optimizing coverage and interval width without requiring quantile inputs or post-processing. QpiGNN employs a dual-head architecture that decouples prediction and uncertainty, and is trained with label-only supervision through a quantile-free joint loss. This design allows efficient training and yields robust prediction intervals, with theoretical guarantees of asymptotic coverage and near-optimal width under mild assumptions. Experiments on 19 synthetic and real-world benchmarks show QpiGNN achieves average 22% higher coverage and 50% narrower intervals than baselines, while ensuring efficiency and robustness to noise and structural shifts.

## 1 INTRODUCTION

Graph Neural Networks (GNNs) are increasingly applied to node regression, enabling accurate prediction of values at graph nodes. They have also shown strong potential in high-stakes domains such as healthcare (Li et al., 2020) and criminal justice (Zhou et al., 2024). Yet most GNNs rely on deterministic architectures that produce point estimates without uncertainty, and their message-passing mechanisms can propagate and amplify data biases (Jiang et al., 2024; Lin et al., 2024), thereby heightening risks in real-world decision-making (Kwon et al., 2022). Although uncertainty-aware modeling has gained attention, uncertainty quantification (UQ) in GNN regression remains underexplored, hindering trustworthy applications. In regression, UQ typically takes the form of prediction intervals that capture plausible ranges (Huang et al., 2023; Pouplin et al., 2024).

Recent works on UQ in GNNs have primarily taken two directions: Bayesian inference and frequentist approaches (Chen et al., 2024; Wang et al., 2024a). Bayesian approaches (Zhao et al., 2020; Stadler et al., 2021) estimate posterior distributions but suffer from scalability issues and sensitivity to prior specification. Frequentist approaches are generally more efficient, but they often rely on resampling or post-hoc calibration, which can be costly or unstable in graph settings. Resampling-based methods (Kang et al., 2022; Liao et al., 2023) incur computational overhead due to repeated inference, while post-hoc calibration methods (Huang et al., 2023) generally require calibration sets and additional processing; in particular, some approaches rely on strong assumptions such as exchangeability, which are often violated in graphs with structural dependencies (Zhou et al., 2020b).

These limitations motivate the development of alternative frequentist approaches that reduce reliance on strong assumptions or post-processing while remaining computationally practical. We propose quantile regression (QR) (Koenker & Bassett Jr, 1978) as a promising alternative for graph settings, due to its ability to handle non-Gaussian and heteroscedastic targets without restrictive distributional assumptions. Unlike resampling- or post-hoc methods, QR directly estimates conditional quantiles, avoiding repeated inference and reducing the need for post-hoc calibration. These properties make QR suitable for UQ in GNNs, where relational dependencies and structural noise violate assumptions, and existing methods struggle with graph-specific challenges such as correlated observations,

noise sensitivity, and heterophily (Angelopoulos & Bates, 2021; Ma et al., 2022). Therefore, to the best of our knowledge, this work represents the first attempt to apply QR to UQ in GNNs.

However, standard QR assumes i.i.d. data and requires quantile-level inputs, where the model either takes a quantile parameter $\tau$ (e.g., $\tau = 0.05, 0.95$) as input or trains separate predictors for each quantile to estimate the conditional quantile function $f(\mathbf{X}; \tau)$. This design increases model complexity and introduces issues such as quantile crossing (Zhou et al., 2020a). When combined with graph message passing, quantile supervision entangles node representations and often yields poorly calibrated, non-compact intervals (Rusch et al., 2023). Moreover, QR-based interval estimation methods that have not been applied to GNNs still rely on explicit quantile-level inputs or fail to distinguish predictions from uncertainty (Tagasovska & Lopez-Paz, 2019; Pouplin et al., 2024), which limits their stability and expressiveness in graph settings.

These limitations highlight the need for a quantile-free, graph-aware approach that retains QR's flexibility while avoiding reliance on quantile-level inputs and strong assumptions such as exchangeability in GNNs. To this end, we propose **Quantile-free Prediction Interval GNN (QpiGNN)**[1], a framework for uncertainty quantification in GNNs through prediction interval estimation. QpiGNN is built on two key ideas. First, a *dual-head architecture* that separates prediction from uncertainty estimation, reducing oversmoothing and entanglement. Second, a *quantile-free joint loss* directly optimizes coverage and interval width from label-only supervision, eliminating the need for quantile-level inputs and post-processing. Collectively, these components enable stable training and calibrated intervals, while ensuring coverage and near-optimal width under mild assumptions for UQ in graphs. Our main contributions are summarized as follows:

1. **Quantile-free UQ for GNNs**: In Section 3, we discuss the structural limitations of applying QR-based interval estimation to GNNs and introduce QpiGNN, the first quantile-free framework that enables calibrated and compact node-level uncertainty quantification.

2. **Framework Design and Theory**: We develop a dual-head architecture with a quantile-free joint loss that decouples prediction from uncertainty, mitigates oversmoothing, and—under mild assumptions—offers theoretical guarantees of coverage and near-optimal width.

3. **Empirical Results**: Across 19 diverse synthetic and real-world benchmarks, QpiGNN achieves on average 22% higher coverage and 50% narrower intervals than competitive baselines, while still remaining efficient and robust to noise and structural shifts.

## 2   RELATED WORK

**Uncertainty Quantification (UQ) in GNNs**   UQ in GNNs has been studied through ensemble, Bayesian, and frequentist approaches. Ensemble methods (Bazhenov et al., 2022; Wang et al., 2024b) improve robustness via multiple predictions but are costly and sensitive to shifts, limiting scalability in large graphs. Bayesian methods (Zhao et al., 2020; Stadler et al., 2021) provide principled probabilistic estimates by modeling posterior uncertainty but often face scalability issues and sensitivity to prior specification. Frequentist methods (Kang et al., 2022; Liao et al., 2023) are more efficient but often depend on resampling or post-hoc calibration, causing significant overhead. Among such post-hoc approaches, Conformal Prediction (CP) provides distribution-free coverage guarantees but relies on exchangeability (often simplified as i.i.d.), an assumption that can be violated in graph data (Vovk et al., 2005). Although recent works attempt to relax these via partial exchangeability (Huang et al., 2023; Zhao et al., 2024), CP remains sensitive to heterogeneity such as hubs and heterophily, highlighting the need for alternative approaches.

**Quantile Regression (QR)**   QR models asymmetric and heteroscedastic targets without strong distributional assumptions (Koenker & Bassett Jr, 1978). However, existing graph-based QR methods such as GSL-QR (Zhang et al., 2023) and PE-GQNN (de Amorim et al., 2024) primarily target prediction or representation learning rather than uncertainty estimation. Recent extensions of QR for interval estimation, such as SQR (Tagasovska & Lopez-Paz, 2019) and RQR (Pouplin et al., 2024), improve flexibility but are not designed for GNNs, often causing calibration failures or oversmoothing when combined with message passing. In contrast, we integrate QR into GNNs in a quantile-free manner, removing explicit quantile-level inputs and post-processing, and thereby achieving calibrated and robust interval prediction for graphs.

---

[1]Code available at `https://anonymous.4open.science/r/QpiGNN-15366`

## 3 PRELIMINARIES AND BACKGROUND

### 3.1 NODE-WISE PREDICTION INTERVALS IN GRAPH NEURAL NETWORKS

Let $G = (\mathcal{V}, \mathcal{E})$ be a graph, where $\mathcal{V}$ is the set of nodes with $n = |\mathcal{V}|$, and $\mathcal{E} \subseteq \mathcal{V} \times \mathcal{V}$ is the set of edges. Each node $v \in \mathcal{V}$ is associated with a feature vector $\mathbf{x}_v \in \mathbb{R}^d$, where $d$ is the dimensionality of node features. The feature matrix is denoted by $\mathbf{X} \in \mathbb{R}^{n \times d}$. Each node also has a scalar regression target $y_v \in \mathbb{R}$, and the full target vector is $\mathbf{y} = [y_1, y_2, \ldots, y_n]^\top \in \mathbb{R}^n$. The goal is to predict, for each node $v$, a *prediction interval* $[\hat{y}_v^{\text{low}}, \hat{y}_v^{\text{up}}]$ such that $\hat{y}_v^{\text{low}} \leq y_v \leq \hat{y}_v^{\text{up}}$ holds with high probability.

GNNs iteratively update node representations by aggregating information from their local neighborhoods. The final representation of node $v$, denoted as $\mathbf{h}_v$, is used to estimate the target value $y_v$. At layer $l$, the representation is updated as follows:

$$\mathbf{h}_v^{(l)} = \sigma \left( W^{(l)} \cdot \text{AGG} \left( \{ \mathbf{h}_u^{(l-1)} \mid u \in \mathcal{N}(v) \} \right) \right), \tag{1}$$

where $\text{AGG} \in \{\text{MEAN, SUM, MAX}\}$ is an aggregation function and $\sigma$ a non-linearity. Aggregation varies by architecture, using mean, sum, or attention (e.g., GraphSAGE (Hamilton et al., 2017), GCN (Kipf & Welling, 2017)). Here, $\mathcal{N}(v)$ denotes the neighbors of node $v$.

### 3.2 QUANTILE REGRESSION AND EXTENSIONS

QR estimates the conditional quantile function $q_\tau(x)$ with $P(y \leq q_\tau(x)) = \tau$, where $\tau \in (0, 1)$ is the quantile level. Prediction intervals are obtained by setting the bounds to $\tau^{\text{low}} = \alpha/2$ and $\tau^{\text{up}} = 1 - \alpha/2$, where $\alpha$ is the miscoverage rate and $1 - \alpha$ the target coverage (e.g., $\alpha = 0.1$ yields a 90% interval). A key limitation of QR is quantile crossing (Zhou et al., 2020a), where predictions at different quantile levels violate monotonicity, causing lower quantiles to exceed higher ones. To overcome this issue, several extensions of QR have been proposed.

Tagasovska & Lopez-Paz (2019) proposed Simultaneous Quantile Regression (SQR), which jointly estimates multiple quantiles using a single model $f_\theta(x, \tau)$ conditioned on both input and quantile level. Rather than training separate models for each quantile, SQR optimizes a unified objective. The standard QR loss, known as the pinball loss, is defined as $\mathcal{L}_{\text{QR}}(y, \hat{y}) = (\tau - \mathbb{I}(y < \hat{y}))(y - \hat{y})$, where $\tau \in (0, 1)$ is a fixed quantile level. In contrast, SQR minimizes the expected pinball loss over a continuous range of quantiles $\tau \sim \mathcal{U}(0, 1)$:

$$\mathcal{L}_{\text{SQR}} = \mathbb{E}_{\tau \sim \mathcal{U}(0,1)}[\mathcal{L}_{\text{QR}}(y, f_\theta(x, \tau))]. \tag{2}$$

This formulation mitigates quantile crossing and improves parameter efficiency by sharing representations across quantile levels. However, our experiments show that SQR suffers from instability and calibration failure when combined with the message passing of GNNs as defined in Equation (1).

Beyond quantile-based approaches, Pouplin et al. (2024) proposed Relaxed Quantile Regression (RQR), originally developed for MLPs, which estimates prediction intervals by learning the conditional center and spread without relying on pre-defined quantile-level inputs. RQR predicts input-dependent bounds $[\hat{y}^{\text{low}}(x), \hat{y}^{\text{up}}(x)]$ and optimizes them for target coverage and minimal width using a width-regularized objective:

$$\mathcal{L}_{\text{RQR-W}} = \left( \alpha + 2\lambda - \mathbb{I}\left[ \hat{y}^{\text{low}} \leq y \leq \hat{y}^{\text{up}} \right] \right)(y - \hat{y}^{\text{low}})(y - \hat{y}^{\text{up}}) + \frac{\lambda}{2} \left( \hat{y}^{\text{up}} - \hat{y}^{\text{low}} \right)^2, \tag{3}$$

where $\lambda \geq 0$ controls the trade-off between coverage and compactness, and $\mathbb{I}(\cdot)$ is the indicator function. The first term enforces coverage of $y$, while the second penalizes excessive width.

While effective in tabular MLPs, our experiments reveal that directly extending RQR to GNNs introduces significant challenges. Message passing tends to produce overly smooth and global intervals (Rusch et al., 2023), limiting node-wise adaptivity. Moreover, the single-head design of RQR entangles the learning of center and spread, reducing representation flexibility and degrading calibration performance under graph-structured data. To improve interval validity in graph-based tasks, we further add an ordering penalty $\gamma_{\text{order}}$, which also facilitates comparison with our proposed model:

$$\mathcal{L}_{\text{RQR}^{adj.}} = \mathcal{L}_{\text{RQR-W}} + \gamma_{\text{order}} \cdot \text{ReLU}(\hat{\mathbf{y}}^{\text{low}} - \hat{\mathbf{y}}^{\text{up}}), \tag{4}$$

where $\gamma_{\text{order}} \geq 0$ is a hyperparameter. This penalty, *absent in the original formulation*, is added mainly as a practical fix: applying RQR to GNNs causes quantile crossing that hinders evaluation, rather than reflecting any graph-specific mechanism.

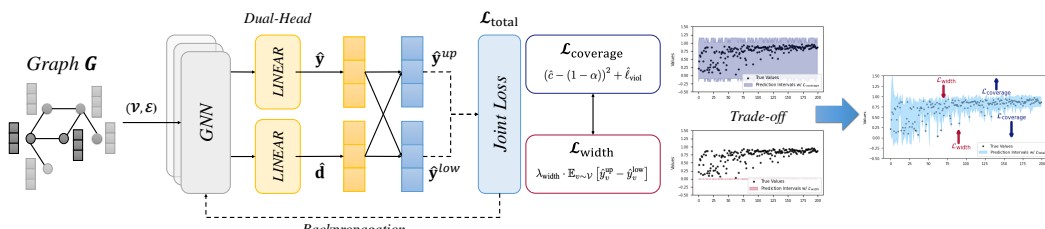

Figure 1: **Overview of Quantile-free Prediction Interval Graph Neural Network.** QpiGNN estimates node-wise prediction intervals using a *dual-head GNN* trained with a *quantile-free joint loss* that trades off coverage and compactness. One head predicts the target value $\hat{y}$, while the other predicts the interval width $\hat{d}$. The loss includes a coverage term $\mathcal{L}$coverage (⬆) encouraging wide enough intervals to maintain coverage, and a width term $\mathcal{L}$width (⬇) promoting tighter intervals. As illustrated, models trained with only the coverage term produce overly wide intervals, whereas the full joint loss yields tighter, locally adaptive intervals that still satisfy the target coverage $1 - \alpha$.

## 4 QUANTILE-FREE PREDICTION INTERVAL GRAPH NEURAL NETWORK

### 4.1 PROBLEM FORMULATION

We address node-wise uncertainty quantification in graph-structured data by learning calibrated prediction intervals without relying on explicit quantile-level inputs, resampling, and post-processing. Given a graph $G = (\mathcal{V}, \mathcal{E})$ with node features $\mathbf{X}$, the objective is to learn a function $f : \mathcal{V} \to \mathbb{R}^2$ that assigns each node $v \in \mathcal{V}$ a prediction interval $f(v) = [\hat{y}_v^{\text{low}}, \hat{y}_v^{\text{up}}]$, such that:

$$\mathbb{P}_{v \sim \mathcal{V}} \left( \hat{y}_v^{\text{low}} \leq y_v \leq \hat{y}_v^{\text{up}} \right) \geq 1 - \alpha, \quad \text{and} \quad \mathbb{E}_{v \sim \mathcal{V}} \left[ \hat{y}_v^{\text{up}} - \hat{y}_v^{\text{low}} \right] \text{ is minimized.}$$

Here, $1 - \alpha$ denotes the target coverage, the probability that true values fall within the predicted intervals. Thus, the learned intervals should be both *calibrated*—achieving the target coverage across nodes—and *compact*—minimizing their average width. We propose QpiGNN, a GNN-based framework for node-wise uncertainty quantification via calibrated prediction intervals.

### 4.2 DUAL-HEAD GNN FOR CALIBRATED PREDICTION INTERVALS

GNNs generate node representations by aggregating neighborhood information (Hamilton et al., 2017; Kipf & Welling, 2017). While effective for prediction, single-head architectures that estimate prediction bounds jointly can suffer from over-smoothing (Li et al., 2018), making it difficult to capture node-wise uncertainty. To mitigate this, QpiGNN employs a dual-head architecture that decouples the estimation of central prediction $\hat{\mathbf{y}}$ and interval width $\hat{\mathbf{d}}$, allowing each to be optimized for a distinct objective—accuracy for $\hat{\mathbf{y}}$, and coverage for $\hat{\mathbf{d}}$.

Separating the two heads also prevents representational conflicts and gradient interference between prediction and uncertainty estimation, enabling each branch to learn its own function class more effectively. Similar dual-head structures have demonstrated strong performance in heteroscedastic and Bayesian regression (Kendall & Gal, 2017; Lakshminarayanan et al., 2017). As shown in Figure 1, this design enables expressive, node-wise uncertainty modeling by structurally separating prediction and uncertainty components, with the full training algorithm provided in Appendix A.

The effectiveness of this design is validated through an ablation study in Section 5.4. Across nine synthetic datasets, the dual-head GNN with a learnable margin outperforms fixed-margin or single-output variants, achieving better empirical trade-offs by maintaining high coverage with significantly narrower intervals. QpiGNN uses a GNN encoder to compute node embeddings $\mathbf{H} = \text{GNN}(\mathbf{X}, \mathcal{E})$, followed by two linear heads $\hat{\mathbf{y}} = \mathbf{W}_{\text{pred}}\mathbf{H} + \mathbf{b}_{\text{pred}}$, $\hat{\mathbf{d}} = \text{Softplus}(\mathbf{W}_{\text{diff}}\mathbf{H} + \mathbf{b}_{\text{diff}})$. The softplus activation ensures $\hat{\mathbf{d}} > 0$, yielding half-widths. The final prediction interval for node $v$ is given by:

$$\hat{y}_v^{\text{low}} = \hat{y}_v - \hat{d}_v, \quad \hat{y}_v^{\text{up}} = \hat{y}_v + \hat{d}_v.$$

This architecture allows QpiGNN to estimate calibrated and compact prediction intervals in an end-to-end manner—without requiring quantile-level input, resampling, or post-processing.

### 4.3 QUANTILE-FREE JOINT LOSS FOR COMPACT PREDICTION INTERVALS

To train QpiGNN to produce prediction intervals that are both calibrated and compact, we define a loss function $\mathcal{L}_{\text{total}}$ consisting of a coverage term $\mathcal{L}_{\text{coverage}}$ and a width regularization term $\mathcal{L}_{\text{width}}$.

$$\mathcal{L}_{\text{total}} = \underbrace{(\hat{c} - (1-\alpha))^2 + \hat{\ell}_{\text{viol}}}_{\mathcal{L}_{\text{coverage}}} + \underbrace{\lambda_{\text{width}} \cdot \mathbb{E}_{v \sim \mathcal{V}} \left[\hat{y}_v^{\text{up}} - \hat{y}_v^{\text{low}}\right]}_{\mathcal{L}_{\text{width}}}, \tag{5}$$

where the empirical coverage is defined as $\hat{c} := \mathbb{P}_{v \sim \mathcal{V}} \left(\hat{y}_v^{\text{low}} \leq y_v \leq \hat{y}_v^{\text{up}}\right)$, and the empirical violation loss is given by $\hat{\ell}_{\text{viol}} := \mathbb{E}_{v \sim \mathcal{V}} \left[|y_v - \hat{y}_v^{\text{low}}| \cdot \mathbb{I}[y_v < \hat{y}_v^{\text{low}}] + |y_v - \hat{y}_v^{\text{up}}| \cdot \mathbb{I}[y_v > \hat{y}_v^{\text{up}}]\right]$.[2] The violation penalty provides fine-grained feedback when predicted intervals fail to capture true targets.[3] The term $\lambda_{\text{width}}$ balances calibration against interval compactness.

Unlike RQR-W in Equation (3), which entangles coverage and width into a single conditional loss—often resulting in oversmoothed intervals in GNNs (Rusch et al., 2023)—our formulation separates these objectives. This separation yields more stable intervals under graph-based learning, as it directly optimizes empirical coverage and width over nodes. We further validate this in Appendix B by applying our loss to an RQR formulation, showing that our objective yields more stable intervals.

Specifically, it penalizes deviations from the target coverage $(1-\alpha)$ via $(\hat{c} - (1-\alpha))^2$, while independently regularizing interval width via $\lambda_{\text{width}} \cdot \mathbb{E}_{v \sim \mathcal{V}}[\hat{y}_v^{\text{up}} - \hat{y}_v^{\text{low}}]$. This disentangled design enables stable training and precise control over node-level uncertainty. Its linear additive penalty prevents excessive amplification and yields more stable behavior across datasets.[4] As shown in Figure 1, jointly minimizing $\mathcal{L}_{\text{coverage}}$ and $\mathcal{L}_{\text{width}}$ yields prediction intervals while maintaining the coverage.

**Asymptotic and Finite-sample Coverage** A goal of QpiGNN is to ensure that predicted intervals achieve the desired coverage level $1-\alpha$ at both finite and asymptotic sample sizes. We provide justification for why the empirical coverage $\hat{c}$ remains close to the target under mild conditions.

**Proposition 1.** *Assume the following mild conditions: (i) the label noise $\varepsilon_v = y_v - f(x_v)$ is bounded and weakly dependent across nodes, (ii) the predicted mean $\hat{y}_v$ and interval half-width $\hat{d}_v$ converge in probability to their targets, and (iii) node embeddings remain sufficiently diverse. Then, as $N \to \infty$, the empirical coverage converges in probability to the target level $1-\alpha$:*

$$\hat{c} \xrightarrow{P} 1-\alpha, \quad \text{equivalently,} \quad \forall \varepsilon > 0, \lim_{N \to \infty} \mathbb{P}(|\hat{c} - (1-\alpha)| > \varepsilon) = 0.$$

*Sketch of Proof.* Define each prediction interval as $[\hat{y}_v^{\text{low}}, \hat{y}_v^{\text{up}}]$ and let $Z_v := \mathbb{I}[\hat{y}_v^{\text{low}} \leq y_v \leq \hat{y}_v^{\text{up}}]$. Under assumptions (i)–(iii), the expected coverage satisfies $\mathbb{E}[Z_v] \to 1-\alpha$. By the Weak Law of Large Numbers (WLLN) (Penrose & Yukich, 2003; Gama & Ribeiro, 2019), the empirical coverage $\hat{c} = \frac{1}{N} \sum_{v=1}^{N} Z_v \xrightarrow{P} 1-\alpha$. $\square$

**Finite-sample Guarantees** For finite samples, deviation of empirical coverage from the target can be controlled using classical concentration inequalities. Under localized message passing, the bounded-difference condition holds approximately, so the inequalities of McDiarmid (1989) and Hoeffding (1994) still apply up to a graph-dependent constant. This yields $|\hat{c} - (1-\alpha)| = \mathcal{O}(1/\sqrt{N})$, implying that even for moderate $N$, empirical coverage $\hat{c}$ remains close to the target $1-\alpha$ with high probability, consistent with the stability observed in our experiments.

This section focuses on finite-sample guarantees derived from concentration inequalities rather than asymptotic arguments. As detailed in Appendix C.5, perturbing a single node affects the coverage estimator by at most $(1/N + \delta_G)$, providing the theoretical basis for the approximate bounded-difference behavior used in our analysis. Moreover, our finite-sample guarantees bound the deviation $|\hat{c} - \mathbb{E}[\hat{c}]|$ even in the absence of exact independence, since the estimator exhibits controlled sensitivity to single-node perturbations, which provides the approximate bounded-difference needed for applying concentration arguments in practical graph regimes.

---

[2]For notational compactness, an equivalent form is $\hat{\ell}_{\text{viol}} = \mathbb{E}_{v \sim \mathcal{V}}[\max(0, (y_v - \hat{y}_v^{\text{low}})(y_v - \hat{y}_v^{\text{up}}))]$; we retain the decomposed form for interpretability and slightly more stable gradients.

[3]In finite samples, the violation loss provides a useful training signal; once the model is calibrated, its effect diminishes and we omit it from the asymptotic analysis (Appendix C.4).

[4]Conversely, when the target range is narrow, multiplicative terms may operate more stably.

**Width Optimality under Coverage Constraints**   To encourage compact prediction intervals, we introduce a width penalty term $\mathcal{L}_{\text{width}}$, weighted by a regularization parameter $\lambda_{\text{width}} \geq 0$, which can be tuned via Bayesian optimization (Snoek et al., 2012) to balance calibration and compactness (typically 0.2–0.5; see Appendix E). We adopt an L1-based mean width penalty for robustness and interpretability, avoiding the instability of L2 losses under outliers (Pearce et al., 2018; Tagasovska & Lopez-Paz, 2019) (See Appendix F). Theoretically, if the conditional distribution $P(y \mid x_v)$ is symmetric around the predicted mean $\hat{y}_v$, the minimum-width interval satisfying the coverage constraint $\hat{c} \geq 1 - \alpha$ is $[\hat{y}_v - d_v^*, \ \hat{y}_v + d_v^*]$, where $d_v^* = F_v^{-1}(1 - \alpha/2)$ is the conditional quantile of $|y_v - \hat{y}_v|$. Since the true distribution is unknown, QpiGNN instead minimizes the total loss $\mathcal{L}_{\text{total}} = \mathcal{L}_{\text{coverage}} + \lambda_{\text{width}} \cdot \mathcal{L}_{\text{width}}$, which can be interpreted as a Lagrangian relaxation (Franceschi et al., 2019) of the constrained optimization problem. When $\hat{c}$ approaches the target $1 - \alpha$, the model prioritizes width reduction, yielding near-optimal intervals, as detailed in Appendix C.2.

**Convergence Properties of the Joint Loss**   The joint loss $\mathcal{L}_{\text{total}}$ is non-convex, but each component is continuous and piecewise smooth, which allows the use of standard stochastic approximation techniques. Following results from stochastic non-convex optimization (Ghadimi & Lan, 2013), we show that training with a diminishing learning rate ensures convergence to a stationary point. Specifically, under the assumptions that (i) each component is continuous and piecewise smooth, (ii) gradients are bounded, and (iii) the step size decays appropriately, the training dynamics satisfy $\lim_{t \to \infty} \mathbb{E}\left[\|\nabla_\theta \mathcal{L}_{\text{total}}(\theta_t)\|\right] = 0$, implying that the model parameters converge to a point where the expected gradient norm vanishes (Kingma & Ba, 2015; Reddi et al., 2018).

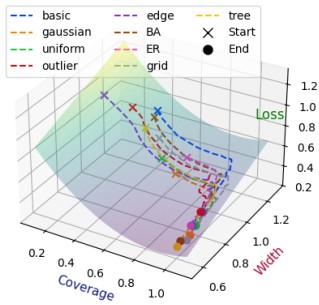

Figure 2: Loss convergence trajectory on synthetic datasets.

This property is crucial given the hybrid nature of the loss, which balances interval width and coverage. As shown in Figure 2, training first reduces sharp coverage-violation penalties and then progressively narrows intervals, following a natural trajectory toward minimizing the overall loss while maintaining target coverage. A formal proof and empirical validation are provided in Appendix C.3.

## 5 EXPERIMENTS

### 5.1 EXPERIMENTAL SETTINGS

**Datasets and Metrics**   We evaluate QpiGNN on 19 datasets. The top half of Table 1 presents synthetic graphs, while the bottom half lists real-world node-level regression benchmarks (Rozember-czki et al., 2021). Detailed descriptions of all datasets are provided in Appendix G. For evaluation, we primarily report Prediction Interval Coverage Probability (PICP) (Rana et al., 2015) and Mean Prediction Interval Width (MPIW) (Khosravi et al., 2010a), which together assess calibration and compactness. Additional metrics are included in Appendix H.

**Baselines**   We compare QpiGNN with six baselines: SQR-GNN, an extension of the MLP-based SQR to GNNs; RQR$^{adj.}$-GNN, which extends RQR to GNNs by enforcing interval ordering in Equation (4); and CF-GNN (Huang et al., 2023). For CF-GNN, we report both our reimplementation[5] and the original version with tuned hyperparameters[6], denoted CF-GNN$^{opt.}$. We also include Evidential Regression (ER)-GNN, a deterministic method that estimates uncertainty by predicting evidential parameters (Amini et al., 2020). For comparison with approximate Bayesian inference methods, we also evaluate BayesianNN (Kendall & Gal, 2017) and MC Dropout (Gal & Ghahramani, 2016).

**Implementation**   All models target $1 - \alpha = 0.90$, and use GraphSAGE with two layers and hidden size 64. We train with the Adam optimizer (learning rate and weight decay = $10^{-3}$) for 500 epochs, averaging over 5 or 10 runs. MC Dropout uses 0.2 dropout rate and 100 stochastic passes. RQR$^{adj.}$-GNN is trained with fixed $\lambda = 1.0$ [7] and order penalty $\gamma_{\text{order}} = 1.0$. All baselines are trained with standard MSE loss. Full details are in Appendix I and our codebase[8].

---

[5]Same GNN architecture, layers, hidden size, and epochs.

[6]Including $\tau$, learning rate, target interval size, loss weights, and architecture.

[7]Using $\lambda = 0.5$ or 0.1 yielded prediction intervals performance differences within 0.01 compared to $\lambda = 1$.

[8]https://anonymous.4open.science/r/QpiGNN-15366

Table 1: Prediction intervals performance (PICP/MPIW) on 19 synthetic and real datasets. Models are grouped by dataset and evaluated on test splits. **Bold** indicates coverage above the 90% target $(1-\alpha)$, and underline marks the lowest MPIW among those. This highlights models achieving both calibration and compactness. $\lambda^{opt.}$ denotes the width penalty selected via Bayesian optimization.

| Model | Basic | | Gaussian | | Uniform | | Outlier | | Edge | | BA | | ER | | Grid | | Tree | |
|---|---|---|---|---|---|---|---|---|---|---|---|---|---|---|---|---|---|---|
| | PCIP | MPIW | PCIP | MPIW | PCIP | MPIW | PCIP | MPIW | PCIP | MPIW | PCIP | MPIW | PCIP | MPIW | PCIP | MPIW | PCIP | MPIW |
| SQR-GNN | 0.85 | 0.33 | 0.88 | 0.50 | 0.88 | 0.51 | **0.90** | 0.10 | **0.91** | 0.32 | 0.81 | 0.72 | 0.75 | 0.60 | 0.78 | 0.53 | 0.80 | 0.26 |
| RQR$^{adj.}$-GNN | **0.90** | 0.82 | 0.88 | 0.53 | **0.90** | 0.68 | **0.90** | 0.36 | **0.93** | 0.83 | 0.78 | 0.75 | 0.88 | 0.77 | 0.72 | 0.48 | 0.85 | 0.68 |
| ER-GNN | **1.00** | 1.24 | **1.00** | 1.85 | **1.00** | 1.53 | **0.97** | 0.88 | **1.00** | 1.07 | **1.00** | 6.89 | **0.72** | 1.86 | **1.00** | 11.92 | **1.00** | 11.21 |
| BayesianNN | **1.00** | 3.01 | **1.00** | 2.98 | **1.00** | 3.00 | **1.00** | 2.95 | **1.00** | 3.06 | **1.00** | 3.08 | **1.00** | 3.01 | **1.00** | 3.01 | **1.00** | 3.00 |
| MC Dropout | **0.99** | 0.32 | 0.55 | 0.20 | 0.65 | 0.26 | 0.58 | 0.06 | **1.00** | 0.30 | 0.67 | 0.26 | 0.76 | 0.23 | 0.33 | 0.16 | 0.64 | 0.20 |
| CF-GNN | **0.92** | 1.90 | **0.91** | 2.90 | **0.90** | 3.04 | **0.93** | 1.92 | **0.92** | 1.78 | **0.90** | 68.27 | **0.90** | 17.15 | **0.94** | 3.18 | **0.93** | 0.97 |
| QpiGNN ($\lambda^{0.5}$) | 0.89 | 0.30 | **0.92** | 0.55 | 0.88 | 0.43 | 0.89 | 0.47 | **0.94** | 0.39 | **0.98** | 0.49 | **0.98** | 0.63 | **0.98** | 0.87 | **0.96** | 0.39 |
| QpiGNN ($\lambda^{0.1}$) | **0.98** | 0.93 | **0.99** | 0.84 | **0.99** | 0.89 | **0.99** | 0.75 | **1.00** | 0.97 | **0.98** | 1.01 | **0.99** | 0.92 | **0.98** | 0.98 | **1.00** | 0.59 |
| QpiGNN ($\lambda^{opt.}$) | **0.90** | 0.30 | **0.95** | 0.64 | **0.93** | 0.62 | **0.90** | 0.49 | **0.94** | 0.54 | **0.98** | 0.48 | **0.98** | 0.63 | **0.99** | 0.93 | **0.96** | 0.39 |

| Model | Education | | Election | | Income | | Unemploy. | | Twitch | | Chameleon | | Crocodile | | Squirrel | | Anaheim | | Chicago | |
|---|---|---|---|---|---|---|---|---|---|---|---|---|---|---|---|---|---|---|---|---|
| | PCIP | MPIW | PCIP | MPIW | PCIP | MPIW | PCIP | MPIW | PCIP | MPIW | PCIP | MPIW | PCIP | MPIW | PCIP | MPIW | PCIP | MPIW | PCIP | MPIW |
| SQR-GNN | 0.88 | 0.32 | 0.89 | 0.47 | 0.86 | 0.22 | 0.87 | 0.33 | 0.30 | 0.03 | 0.37 | 0.01 | 0.44 | 0.01 | 0.22 | 0.01 | 0.88 | 0.32 | 0.87 | 0.21 |
| RQR$^{adj.}$-GNN | 0.87 | 0.49 | 0.89 | 0.54 | 0.89 | 0.36 | **0.90** | 0.38 | **0.91** | 0.42 | 0.86 | 0.15 | 0.87 | 0.08 | 0.89 | 0.15 | 0.85 | 0.50 | 0.88 | 0.30 |
| ER-GNN | **1.00** | 4.37 | **1.00** | 6.90 | **1.00** | 3.12 | **1.00** | 4.80 | **0.99** | 1.33 | **0.97** | 1.08 | **1.00** | 1.56 | **0.97** | 0.80 | **1.00** | 2.09 | **1.00** | 3.06 |
| BayesianNN | **1.00** | 2.96 | **1.00** | 2.98 | **1.00** | 2.97 | **1.00** | 2.98 | **1.00** | 3.07 | **1.00** | 2.95 | **1.00** | 3.07 | **1.00** | 2.97 | **1.00** | 2.94 | **1.00** | 2.99 |
| MC Dropout | 0.40 | 0.11 | 0.48 | 0.18 | 0.45 | 0.09 | 0.41 | 0.09 | **0.91** | 0.15 | 0.47 | 0.02 | 0.46 | 0.01 | 0.31 | 0.02 | 0.50 | 0.11 | 0.34 | 0.07 |
| CF-GNN | 0.88 | 2.78 | 0.89 | 1.08 | **0.92** | 3.48 | **0.90** | 3.16 | **0.92** | 3.53 | - | - | - | - | - | - | **0.90** | 3.22 | **0.90** | 3.12 |
| CF-GNN$^{opt.}$ | **0.90** | 3.10 | **0.91** | 0.94 | **0.91** | 2.92 | 0.89 | 2.61 | 0.89 | 2.34 | - | - | - | - | - | - | **0.90** | 2.82 | **0.91** | 2.26 |
| QpiGNN ($\lambda^{0.5}$) | **0.99** | 0.57 | **0.98** | 0.77 | **0.99** | 0.41 | **1.00** | 0.74 | 0.59 | 0.08 | 0.51 | 0.03 | **0.92** | 0.08 | 0.73 | 0.07 | **0.92** | 0.39 | **0.97** | 0.36 |
| QpiGNN ($\lambda^{0.1}$) | **0.99** | 0.90 | **1.00** | 0.97 | **1.00** | 0.72 | **1.00** | 0.93 | **0.98** | 0.54 | **0.98** | 0.40 | **1.00** | 0.54 | **0.99** | 0.47 | **0.99** | 0.74 | **0.99** | 0.60 |
| QpiGNN ($\lambda^{opt.}$) | **0.99** | 0.59 | **0.98** | 0.77 | **0.99** | 0.44 | **1.00** | 0.73 | **0.94** | 0.36 | **0.96** | 0.23 | **0.97** | 0.16 | **0.96** | 0.18 | **0.93** | 0.40 | **0.98** | 0.36 |

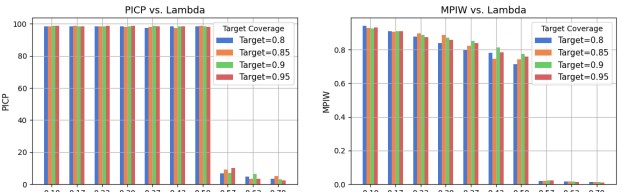 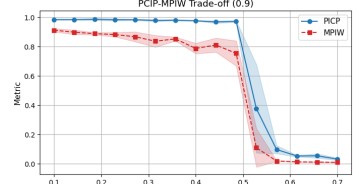

Figure 3: Impact of $\lambda_{\text{width}}$ on PICP and MPIW across $1-\alpha \in [0.8, 0.95]$, over 5 runs (500 epochs) on the ER graph.

Figure 4: PICP–MPIW trade-off under $\lambda_{\text{width}}$ at $1-\alpha = 0.90$ (ER graph).

## 5.2 RESULTS

**Quantitative Evaluation** Table 1 reports PICP and MPIW on 19 datasets under three width penalties: default ($\lambda^{0.5}$), conservative ($\lambda^{0.1}$), and Bayesian-optimized ($\lambda^{opt.}$). On synthetic graphs, $\lambda^{opt.}$ reliably meets target coverage with the lowest MPIW in most cases, while $\lambda^{0.5}$ yields tight but sometimes under-covering intervals and $\lambda^{0.1}$ attains near-perfect coverage at the cost of wider widths. On real graphs, $\lambda^{0.5}$ provides the narrowest intervals, whereas $\lambda^{opt.}$ balances coverage and compactness (e.g., *Chameleon*, *Squirrel*). Against baselines, QpiGNN shows superior trade-off: BayesianNN ensures coverage with wide intervals, MC Dropout is narrow yet unreliable, and SQR-GNN, RQR$^{adj.}$-GNN, and CF-GNN often suffer poor calibration or unstable widths. SQR-GNN can exhibit instability on heterophily graphs, but if slight under-coverage is permitted—as is sometimes allowed outside standard UQ conventions—QR-based baselines can also demonstrate competitive performance. QpiGNN with $\lambda^{opt.}$ improves coverage by 22% and reduces width by 50% on average, with further results in Appendix J.

**Qualitative Evaluation** Figure 9 and 10 in Appendix K illustrate prediction intervals on synthetic and real-world graphs. For example, on *Tree*, SQR-GNN and MC Dropout yield narrow but under-covering intervals, while BayesianNN and CF-GNN ensure coverage only with wide intervals. In contrast, QpiGNN attains the target coverage with balanced width. More broadly, QpiGNN adapts interval widths to data uncertainty across diverse graph settings. It adjusts to local variability in noisy real-world graphs (e.g., Anaheim, Chicago), yields tighter calibrated intervals on sparse graphs (e.g., Twitch), and maintains strong coverage with compact widths on high-variance synthetic graphs (e.g., Tree). These results demonstrate QpiGNN's adaptability and robustness.

Table 2: Robustness analysis on ER graphs with feature/target noise and edge dropout.

| Perturbation Type | Level | QpiGNN | | SQR-GNN | | RQR$^{adj.}$-GNN | |
|---|---|---|---|---|---|---|---|
| | | PICP | MPIW | PICP | MPIW | PICP | MPIW |
| Feature Noise ($\sigma$) | 0.1 | 0.89 | 0.66 | 0.76 | 0.78 | 0.85 | 0.78 |
| | 0.2 | 0.90 | 0.80 | 0.65 | 0.69 | 0.85 | 0.78 |
| | 0.3 | 0.92 | 0.83 | 0.71 | 0.81 | 0.84 | 0.76 |
| Target Noise ($\sigma$) | 0.1 | 0.96 | 0.44 | 0.49 | 0.52 | 0.95 | 0.87 |
| | 0.2 | 0.99 | 0.71 | 0.95 | 0.84 | 0.97 | 1.05 |
| | 0.3 | 1.00 | 1.05 | 0.98 | 0.99 | 0.98 | 1.32 |
| Edge Dropout ($p$) | 0.2 | 0.84 | 0.21 | 0.53 | 0.44 | 0.86 | 0.80 |
| | 0.4 | 0.88 | 0.23 | 0.60 | 0.44 | 0.89 | 0.84 |
| | 0.6 | 0.91 | 0.21 | 0.82 | 0.50 | 0.89 | 0.84 |

Table 3: Exchangeability analysis on real-world datasets with Random, Degree, and Community splits.

| Split Type | Education | | Election | | Income | | Unemploy. | | Twitch | |
|---|---|---|---|---|---|---|---|---|---|---|
| | PICP | MPIW | PICP | MPIW | PICP | MPIW | PICP | MPIW | PICP | MPIW |
| Random | 0.99 | 0.90 | 1.00 | 0.97 | 1.00 | 0.72 | 1.00 | 0.93 | 0.98 | 0.54 |
| Degree | 0.99 | 0.75 | 0.99 | 0.87 | 0.99 | 0.44 | 0.99 | 0.93 | 0.94 | 0.68 |
| Community | 0.99 | 0.54 | 0.99 | 0.86 | 1.00 | 0.46 | 0.97 | 0.84 | 0.99 | 0.49 |

| Split Type | Chameleon | | Crocodile | | Squirrel | | Anaheim | | Chicago | |
|---|---|---|---|---|---|---|---|---|---|---|
| | PICP | MPIW | PICP | MPIW | PICP | MPIW | PICP | MPIW | PICP | MPIW |
| Random | 0.98 | 0.40 | 1.00 | 0.54 | 0.99 | 0.47 | 0.99 | 0.74 | 0.99 | 0.60 |
| Degree | 0.97 | 0.57 | 1.00 | 0.72 | 0.97 | 0.45 | 0.95 | 0.58 | 0.96 | 0.46 |
| Community | 0.95 | 0.58 | 0.98 | 0.74 | 0.97 | 0.47 | 0.97 | 0.60 | 1.00 | 0.53 |

**Hyperparameter Sensitivity** Figure 3 shows that small $\lambda$ (e.g., 0.1) yields near-perfect coverage but wide intervals. As $\lambda$ increases to 0.5, intervals tighten (e.g., MPIW $\approx 0.59$ for $1-\alpha = 0.80$) with minimal loss in PICP. Beyond $\lambda \approx 0.57$, coverage collapses ($< 0.06$), revealing a steep calibration–sharpness trade-off. Figure 4 illustrates that at $1-\alpha = 0.90$, PICP remains stable ($\geq 0.96$) until $\lambda \approx 0.5$, after which both PICP and MPIW degrade rapidly, suggesting a critical tipping point. Bayesian optimization (Snoek et al., 2012) can be applied to select $\lambda_{\text{width}}$ for each dataset. As shown in Appendix E, optimal $\lambda$ values lie between 0.2 and 0.5, with synthetic datasets favoring stronger regularization. Additional analyses for the baselines and limitations are provided in Appendix D.

**Computational Efficiency** Appendix L compares efficiency, runtime, and complexity across baselines. QpiGNN balances accuracy and efficiency, with training cost comparable to lightweight baselines (e.g., SQR-GNN). All models share complexity $\mathcal{O}(Ed + Nd^2)$. MC Dropout and BayesianNN add overhead from sampling, while CF-GNN incurs the highest cost from calibration.

## 5.3 ROBUSTNESS AND GENERALIZATION

**Robustness to Perturbations** We evaluate robustness under three types of perturbations—feature noise, target noise, and edge dropout—on ER graphs in Table 2. Across all settings, QpiGNN consistently maintains valid coverage and compact intervals, outperforming SQR-GNN and RQR$^{adj.}$-GNN. Under feature noise, QpiGNN maintains high coverage (PICP $\approx 0.90$) with moderate widths, while SQR-GNN under-covers and RQR$^{adj.}$-GNN is less stable. For target noise, QpiGNN expands intervals to preserve near-perfect coverage (PICP $= 1.00$ at $\sigma = 0.3$, MPIW $\approx 1.05$), whereas SQR-GNN collapses early and RQR$^{adj.}$-GNN needs much wider intervals. With edge dropout, QpiGNN raises coverage ($0.84 \rightarrow 0.91$) while keeping MPIW compact ($\approx 0.21$), outperforming SQR-GNN's poor coverage and RQR$^{adj.}$-GNN's wide intervals.

**Exchangeability under Splits** Table 3 reports QpiGNN performance under three splits: Random, assigning nodes uniformly; Degree, separating nodes by degree to test generalization across low- and high-degree regions; and Community, splitting by graph communities to evaluate cross-community transfer. Results show that while Random splits yield stable coverage and widths, Degree splits often increase interval width (e.g., *Twitch*, MPIW $= 0.68$), and Community splits frequently produce narrower yet valid intervals (e.g., *Education*, $0.90 \rightarrow 0.54$). QpiGNN maintains coverage and adapts widths even under non-exchangeable splits, showing robustness to structural shifts.

**Structural Shift Generalization** We evaluate robustness to structural shifts by training QpiGNN on one graph type and testing on others. This setting reflects realistic scenarios where graphs evolve, are partially observed, or reconstructed over time, making robustness to such changes critical for GNN-based uncertainty quantification. As shown in Figure 5, models trained on expressive graphs like *BA* and *ER* generalize best, achieving PICP $\geq 0.89$ on most targets. In contrast, models trained on simpler sources (e.g., *Tree*, *Basic*) transfer poorly, especially to irregular graphs (e.g., PICP $< 0.2$ on *BA*). We also observe a coverage–width trade-off: *Basic* and *Tree* yield narrow but under-covering intervals, while *ER* strikes a better balance. Full results appear in Appendix M.

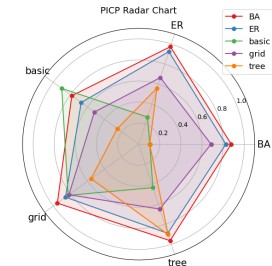

Figure 5: Radar plot of QpiGNN PICP on synthetic graphs, showing robustness under structural shifts.

### 5.4 ABLATION STUDY

We examine the contributions of QpiGNN's architecture and loss design on nine synthetic datasets. Figure 6 presents the PICP–MPIW trade-off, with results reported in Appendix N. *(1) Architecture:* The dual-head model with a learnable margin (green points) consistently outperforms fixed-margin and single-output variants, achieving superior calibration–compactness trade-offs. *(2) Loss:* The complete objective delivers the best overall performance (green dashed ellipse). Coverage-only training yields overly wide intervals, width-only training fails to calibrate, and pure MSE baseline collapses to trivial outputs, underscoring the importance of calibrated, uncertainty-aware losses.

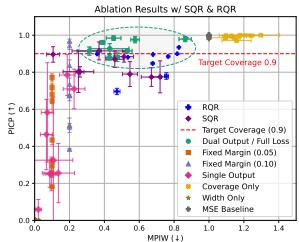

Figure 6: PICP–MPIW trade-off on nine synthetic datasets, comparing our ablation with SQR and RQR.

## 6 DISCUSSION

QpiGNN's design yields strong empirical performance: by decoupling prediction and uncertainty, it avoids extra procedures and provides stable, calibrated estimates under noise and dependencies. Consistently, it produces tighter reliable intervals than baselines, demonstrating efficiency and robustness in line with UQ principles that prioritize reducing width when exceeding target coverage.

**Constraints and Empirical Robustness** Theoretical guarantees of QpiGNN rely on mild assumptions that may be violated in real graphs due to heavy-tailed noise (Verma & Zhang, 2019; Jin et al., 2020), model bias (Pouplin et al., 2024), and structural redundancy (Tagasovska & Lopez-Paz, 2019). These factors represent practical violations of independence and symmetry, often limiting the reliability of uncertainty quantification in graph-structured data. Nevertheless, our results show that QpiGNN maintains calibrated and compact intervals across diverse settings—including noisy, sparse, and high-variance graphs—outperforming baselines on 7 of 10 real-world datasets (Table 1). It further demonstrates resilience to feature and edge noise (Table 2) and sustains performance even under splits violating exchangeability (Table 3), highlighting that our method does not rely on strong assumptions. QpiGNN's limitations may be mitigated by incorporating coverage-aware objectives or robust training strategies to reduce model bias and handle heavy-tailed noise and structural redundancy. Moreover, the bounded-difference assumption may behave differently in dense or high-degree graphs, and clarifying this limitation is a direction for theoretical refinement.

**Task Extensions** QpiGNN extends naturally beyond node regression. For graph-level regression, pooled embeddings $\mathbf{h}_G = \rho(\text{GNN}(\mathbf{X}, \mathcal{E}))$ are fed into two linear heads: $\hat{y}_G = \mathbf{W}_{\text{pred}}\mathbf{h}_G + b_{\text{pred}}$, $\hat{d}_G = \text{Softplus}(\mathbf{W}_{\text{diff}}\mathbf{h}_G + b_{\text{diff}})$, yielding prediction intervals $[\hat{y}_G - \hat{d}_G, \ \hat{y}_G + \hat{d}_G]$. For link prediction, edge embeddings (e.g., $\mathbf{h}_{uv} = \phi([\mathbf{h}_u \parallel \mathbf{h}_v \parallel \mathbf{h}_u \odot \mathbf{h}_v])$) are processed in the same way, producing calibrated predictions $\hat{y}_{uv} \pm \hat{d}_{uv}$. While classification is not directly supported, the quantile-free dual-head design can be adapted via predictive sets or confidence margins.

## 7 CONCLUSION

We proposed QpiGNN, a GNN-based framework for uncertainty quantification that estimates calibrated and compact prediction intervals for node-level regression. By combining a dual-head architecture with a quantile-free joint loss, QpiGNN disentangles prediction from uncertainty estimation in an end-to-end manner, eliminating the need for quantile-level inputs, resampling, or post-processing. Experiments on 19 synthetic and real-world datasets show that QpiGNN outperforms baselines, delivering reliable calibration, robustness to noise, and precise control of the coverage–width trade-off. It also remains effective without exchangeability and generalizes well under structural shifts, particularly when trained on expressive graphs. Our ablation studies further confirm the contribution of the dual-head design and quantile-free joint loss to these trade-offs. Future work includes extending QpiGNN to broader graph applications requiring reliable UQ and improving its uncertainty modeling by disentangling aleatoric and epistemic components, enabling more informed decision-making in high-stakes settings. Another important direction is applying QpiGNN to dynamic or evolving graphs, where structural drift and temporal dependencies introduce additional uncertainties that remain unexplored.

## ETHICS STATEMENT

This work relies exclusively on publicly available synthetic and real-world benchmark datasets, containing no sensitive or personally identifiable information. The proposed method aims to improve uncertainty quantification in graph neural networks. However, misuse or misinterpretation of uncertainty estimates could pose risks in high-stakes applications, and thus careful application is advised.

## REPRODUCIBILITY STATEMENT

We provide implementation details, hyperparameter settings, and dataset descriptions in the main text and appendix. All datasets are publicly accessible, and the code for reproducing our experiments is already released. Results are reported as averages over multiple runs with fixed random seeds to ensure reproducibility.

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
