# A  TRAINING ALGORITHM FOR QPIGNN

---

**Algorithm 1** Training QpiGNN with Coverage–Width Joint Loss

---

**Require:** Graph $G = (\mathcal{V}, \mathcal{E})$, node features $\mathbf{X} \in \mathbb{R}^{N \times d}$, ground truth targets $\mathbf{y} \in \mathbb{R}^N$, edge index $\mathcal{E}$, desired coverage level $1 - \alpha$, width penalty (regularization) weight $\lambda_{\text{width}}$, fixed margin (optional), total training epochs $T$, learning rate $\eta$, weight decay $\omega$

**Ensure:** Trained model parameters $\boldsymbol{\theta}$

1: **Model and Optimizer Initialization:**
2: Initialize QpiGNN with parameters $\boldsymbol{\theta}$ (with `dual_output=True` to estimate prediction and interval width separately)
3: Define composite loss:
$$\mathcal{L}_{\text{total}} = \mathcal{L}_{\text{coverage}} + \lambda_{\text{width}} \cdot \mathcal{L}_{\text{width}}$$
4: Initialize optimizer: `Adam`($\boldsymbol{\theta}$, $\eta$, $\omega$)
5: **for** $t = 1$ to $T$ **do**
6:     **Set model to training mode:** `model.train()`
7:     **Reset gradients:** `optimizer.zero_grad()`
8:     **Message Passing and Node Representation:**
$$\mathbf{H}_1 \leftarrow \texttt{ReLU}(\texttt{SAGEConv}_1(\mathbf{X}, \mathcal{E}))$$
$$\mathbf{H}_2 \leftarrow \texttt{ReLU}(\texttt{SAGEConv}_2(\mathbf{H}_1, \mathcal{E}))$$
9:     **Dual-Head: Prediction and Interval Width Estimation**
$$\hat{\mathbf{y}} \leftarrow \texttt{Linear}_{\text{pred}}(\mathbf{H}_2)$$
$$\hat{\mathbf{d}} \leftarrow \begin{cases} \texttt{sigmoid}(\texttt{Linear}_{\text{diff}}(\mathbf{H}_2)) & \text{if fixed margin is not set} \\ \texttt{Constant}(\text{fixed\_margin}) & \text{otherwise} \end{cases}$$
10:     **Interval Construction:**
$$\hat{\mathbf{y}}^{\text{low}} \leftarrow \hat{\mathbf{y}} - \hat{\mathbf{d}}, \quad \hat{\mathbf{y}}^{\text{up}} \leftarrow \hat{\mathbf{y}} + \hat{\mathbf{d}}$$
11:     **Loss Computation:**
$$\mathcal{L}_{\text{total}} \leftarrow \mathcal{L}_{\text{coverage}} + \lambda_{\text{width}} \cdot \mathcal{L}_{\text{width}}$$

$$\mathcal{L}_{\text{coverage}} = (\hat{c} - (1 - \alpha))^2 + \hat{\ell}_{\text{viol}}, \quad \mathcal{L}_{\text{width}} = \lambda_{\text{width}} \cdot \mathbb{E}_{v \sim \mathcal{V}} \left[ \hat{y}_v^{\text{up}} - \hat{y}_v^{\text{low}} \right]$$

$$\text{where} \quad \hat{c} := \mathbb{P}_{v \sim \mathcal{V}} \left( \hat{y}_v^{\text{low}} \leq y_v \leq \hat{y}_v^{\text{up}} \right),$$

$$\hat{\ell}_{\text{viol}} := \mathbb{E}_{v \sim \mathcal{V}} \left[ |y_v - \hat{y}_v^{\text{low}}| \cdot \mathbb{I}[y_v < \hat{y}_v^{\text{low}}] + |y_v - \hat{y}_v^{\text{up}}| \cdot \mathbb{I}[y_v > \hat{y}_v^{\text{up}}] \right]$$

12:     **Backward and Optimization Step:**
13:     $\mathcal{L}_{\text{total}}$`.backward()`
14:     `optimizer.step()`
15: **end for**
16: **return** Trained `QpiGNN`($\boldsymbol{\theta}$)

---

Algorithm 1 outlines the training process for QpiGNN, which learns to generate node-specific prediction intervals through a dual-head architecture: one head predicts the mean response, while the other estimates the interval width. The total loss $\mathcal{L}_{\text{total}}$ combines a coverage constraint $\mathcal{L}_{\text{coverage}}$, which penalizes incorrect interval coverage, and a width penalty $\mathcal{L}_{\text{width}}$, which encourages compact and informative intervals. The contribution of the width term is controlled by a regularization weight $\lambda_{\text{width}} \geq 0$, which governs the trade-off between calibration and interval tightness. By eliminating the need for explicit quantile estimation or conformal post-processing, QpiGNN enables fast, one-pass uncertainty quantification (UQ) that scales efficiently with graph size and structure.

Table 4: Comparison using QpiGNN loss replaced by the RQR loss across 19 synthetic and real datasets (10 runs, target coverage 90%).

| Model | Basic | | Gaussian | | Uniform | | Outlier | | Edge | | BA | | ER | | Grid | | Tree | |
|---|---|---|---|---|---|---|---|---|---|---|---|---|---|---|---|---|---|---|
| | PCIP | MPIW | PCIP | MPIW | PCIP | MPIW | PCIP | MPIW | PCIP | MPIW | PCIP | MPIW | PCIP | MPIW | PCIP | MPIW | PCIP | MPIW |
| QpiGNN ($\lambda = 0.5$) | 0.89 | 0.30 | 0.92 | 0.55 | 0.88 | 0.43 | 0.89 | 0.47 | 0.94 | 0.39 | 0.98 | 0.49 | 0.98 | 0.63 | 0.98 | 0.87 | 0.96 | 0.39 |
| QpiGNN ($\lambda = optimal$) | 0.90 | 0.30 | 0.95 | 0.64 | 0.93 | 0.62 | 0.90 | 0.49 | 0.94 | 0.54 | 0.98 | 0.48 | 0.98 | 0.63 | 0.99 | 0.93 | 0.96 | 0.39 |
| QpiGNN w/ RQRLoss ($\lambda = 0.1$) | 0.92 | 0.84 | 0.93 | 0.61 | 0.90 | 0.71 | 0.94 | 0.46 | 0.93 | 0.85 | 0.19 | 0.85 | 0.50 | 0.83 | 0.87 | 0.62 | 0.27 | 0.81 |

| Model | Education | | Election | | Income | | Unemploy. | | Twitch | | Chameleon | | Crocodile | | Squirrel | | Anaheim | | Chicago | |
|---|---|---|---|---|---|---|---|---|---|---|---|---|---|---|---|---|---|---|---|---|
| | PCIP | MPIW | PCIP | MPIW | PCIP | MPIW | PCIP | MPIW | PCIP | MPIW | PCIP | MPIW | PCIP | MPIW | PCIP | MPIW | PCIP | MPIW | PCIP | MPIW |
| QpiGNN ($\lambda = 0.5$) | 0.99 | 0.57 | 0.98 | 0.77 | 0.99 | 0.41 | 1.00 | 0.74 | 0.59 | 0.08 | 0.51 | 0.03 | 0.92 | 0.08 | 0.73 | 0.07 | 0.92 | 0.39 | 0.97 | 0.36 |
| QpiGNN ($\lambda = optimal$) | 0.99 | 0.59 | 0.98 | 0.77 | 0.99 | 0.44 | 1.00 | 0.73 | 0.94 | 0.36 | 0.96 | 0.23 | 0.97 | 0.16 | 0.96 | 0.18 | 0.93 | 0.40 | 0.98 | 0.36 |
| QpiGNN w/ RQRLoss ($\lambda = 0.1$) | 0.90 | 0.53 | 0.89 | 0.52 | 0.90 | 0.37 | 0.84 | 0.46 | 0.95 | 0.72 | 0.94 | 0.30 | 0.97 | 0.24 | 0.92 | 0.25 | 0.83 | 0.44 | 0.82 | 0.41 |

# B APPLYING QPIGNN LOSS TO AN RQR-STYLE FORMULATION

Keeping the architecture fixed, we replaced the QpiGNN objective with the RQR loss and evaluated both variants across the 19 synthetic and real datasets in Table 4. This controlled setup allows us to isolate the effect of the loss function while keeping the encoder and training pipeline identical.

Across the synthetic datasets, QpiGNN maintains near-target coverage with compact intervals, whereas the RQR-based variant often collapses, showing under-coverage (e.g., 0.19 on BA, 0.27 on Tree) and inflated intervals. A similar pattern appears on real-world graphs: QpiGNN provides reliable coverage with reasonable interval widths, while the RQR-style model shows degradation, including under-coverage and wider intervals on datasets such as Twitch, Chameleon, and Crocodile.

Overall, these results confirm that simply substituting the loss with an RQR formulation is insufficient for stable UQ on graphs. The proposed joint loss (Equation (5)) is crucial for achieving both coverage reliability and interval compactness under graph-dependent residual structures.

# C THEORETICAL FOUNDATIONS OF QPIGNN

In this appendix, we provide theoretical insights and guarantees underpinning the design of QpiGNN. Specifically, we analyze the method along three key dimensions:

1. The convergence of empirical coverage to the desired target level $1 - \alpha$
2. The optimality of predicted interval widths under the coverage constraint
3. The convergence behavior of the proposed loss function under gradient-based optimization.

These results offer a foundational understanding of why QpiGNN is both statistically sound and practically effective for node-level UQ in graph-structured data.

## C.1 COVERAGE CONSISTENCY

We formally establish that the empirical coverage $\hat{c}$, defined as the proportion of true targets contained within the predicted intervals, converges to the target level $1 - \alpha$ as the number of nodes increases. This result provides theoretical justification for the calibration behavior observed in QpiGNN.

**Proposition 2** (Asymptotic Coverage Consistency). *Let $\hat{y}_v$ and $\hat{d}_v > 0$ denote the predicted mean and interval half-width for node $v$, forming the prediction interval $[\hat{y}_v^{low}, \hat{y}_v^{up}] = [\hat{y}_v - \hat{d}_v, \ \hat{y}_v + \hat{d}_v]$. Suppose the following conditions hold:*

- *(A1) The noise $\varepsilon_v = y_v - f(x_v)$ is bounded and independent across nodes;*

- *(A2) The predicted mean and width satisfy $\hat{y}_v \xrightarrow{P} \mathbb{E}[y_v|x_v]$ and $\hat{d}_v \xrightarrow{P} d_v^*$, and the loss $\mathcal{L}_{total}$ is Lipschitz-continuous in the model parameters;*

- *(A3) Node embeddings $\mathbf{H}$ are sufficiently expressive and bounded in norm.*

*Then, as $N \to \infty$,*

$$\hat{c} := \mathbb{P}_{v \sim \mathcal{V}} \left( \hat{y}_v^{low} \leq y_v \leq \hat{y}_v^{up} \right) \xrightarrow{P} 1 - \alpha.$$

*Proof.* We verify the conditions of the Weak Law of Large Numbers (WLLN) (Penrose & Yukich, 2003; Gama & Ribeiro, 2019) for the sequence $\{Z_v\}_{v=1}^N$, where each $Z_v := \mathbb{I}[\hat{y}_v^{\text{low}} \leq y_v \leq \hat{y}_v^{\text{up}}]$ is:

- Bounded: $Z_v \in [0,1]$ for all $v$,

- Mean-convergent: $\mathbb{E}[Z_v] \to 1 - \alpha$ as $N \to \infty$,

- Weakly dependent (assumed negligible or bounded via local message passing).

Under these conditions, the empirical mean $\hat{c} = \frac{1}{N}\sum_{v=1}^N Z_v$ satisfies the WLLN:

$$\hat{c} \xrightarrow{P} 1 - \alpha.$$

$\square$

**Finite-sample Concentration Bounds**  While the above result ensures asymptotic calibration, we now provide finite-sample guarantees that quantify the deviation of $\hat{c}$ from its expected value. These bounds show that QpiGNN maintains reliable coverage even with moderate graph sizes.

**Proposition 3** (Hoeffding-type Bound (Hoeffding, 1994)). *If* $Z_v := \mathbb{I}[\hat{y}_v^{low} \leq y_v \leq \hat{y}_v^{up}] \in [0,1]$ *are independent for* $v = 1, \ldots, N$, *then for any* $\delta \in (0,1)$, *with probability at least* $1 - \delta$,

$$|\hat{c} - \mathbb{E}[\hat{c}]| \leq \sqrt{\frac{\log(2/\delta)}{2N}}.$$

**Proposition 4** (McDiarmid-type Bound (McDiarmid, 1989)). *Let* $\hat{c} = f(X_1, \ldots, X_N)$, *where* $X_v = (x_v, y_v)$, *and suppose that changing a single sample* $X_v$ *alters* $\hat{c}$ *by at most* $1/N$. *Then for any* $\epsilon > 0$,

$$\mathbb{P}\left(|\hat{c} - \mathbb{E}[\hat{c}]| > \epsilon\right) \leq 2\exp\left(-2N\epsilon^2\right).$$

These concentration bounds confirm that QpiGNN's empirical coverage remains close to its expected value with high probability, thus providing calibration guarantees under both asymptotic and finite-sample settings.

## C.2 WIDTH OPTIMALITY UNDER COVERAGE CONSTRAINTS

We now theoretically justify why the objective used in QpiGNN encourages compact prediction intervals while satisfying the coverage requirement. We formalize this as a constrained optimization problem where the goal is to minimize the average width of prediction intervals subject to a minimum coverage level.

**Theorem 1** (Width-Optimality under Coverage). *Let each* $y_v$ *follow a symmetric, continuous distribution centered at* $\hat{y}_v = \mathbb{E}[y_v \mid x_v]$. *Then, the solution to the optimization problem*

$$\min_{\{\hat{d}_v \geq 0\}} \frac{1}{N}\sum_{v=1}^N 2\hat{d}_v \quad \text{subject to} \quad \mathbb{P}_{v \sim \mathcal{V}}\left(\hat{y}_v^{low} \leq y_v \leq \hat{y}_v^{up}\right) \geq 1 - \alpha$$

*is given by* $\hat{d}_v^* = F_v^{-1}(1 - \alpha/2)$, *where* $F_v^{-1}(\cdot)$ *denotes the conditional quantile function of* $|y_v - \hat{y}_v|$.

*Proof.* Given the symmetry and continuity of $y_v$ around $\hat{y}_v$, the most compact interval capturing mass $1 - \alpha$ is symmetric about the mean. By the definition of quantiles, the smallest such symmetric interval corresponds to the threshold $\hat{d}_v^*$ such that $\mathbb{P}(|y_v - \hat{y}_v| \leq \hat{d}_v^*) = 1 - \alpha$, which gives $\hat{d}_v^* = F_v^{-1}(1 - \alpha/2)$. This value achieves the minimal width $2\hat{d}_v^*$ while satisfying the global coverage constraint. $\square$

**Lagrangian Relaxation.**  In practice, the true quantile function $F_v^{-1}(\cdot)$ is unknown and intractable to estimate directly. Instead, QpiGNN minimizes the following surrogate loss:

$$\left(\mathbb{P}_{v \sim \mathcal{V}}\left(\hat{y}_v^{\text{low}} \leq y_v \leq \hat{y}_v^{\text{up}}\right) - (1 - \alpha)\right)^2 + \lambda_{\text{width}} \cdot \mathbb{E}_{v \sim \mathcal{V}}\left[\hat{y}_v^{\text{up}} - \hat{y}_v^{\text{low}}\right].$$

which serves as a soft Lagrangian penalty formulation (Franceschi et al., 2019). The hyperparameter $\lambda_{\text{width}}$ modulates the trade-off between satisfying the coverage constraint and minimizing interval width. When empirical coverage exceeds the target, the model reduces interval sizes; when coverage falls short, it implicitly expands the intervals by adjusting the predicted margins. This leads to adaptive and stable training dynamics without requiring explicit quantile estimation.

## C.3 CONVERGENCE OF THE TRAINING OBJECTIVE

We analyze the convergence behavior of a simplified version of the training loss used in QpiGNN, omitting the violation loss term $\hat{\ell}_{\text{viol}}$. The objective is defined as:

$$\mathcal{L}_{\text{total}}(\theta) = \left(\mathbb{P}_{v \sim \mathcal{V}}\left(\hat{y}_v^{\text{low}} \leq y_v \leq \hat{y}_v^{\text{up}}\right) - (1 - \alpha)\right)^2 + \lambda_{\text{width}} \cdot \mathbb{E}_{v \sim \mathcal{V}}\left[\hat{y}_v^{\text{up}} - \hat{y}_v^{\text{low}}\right],$$

where $\theta$ denotes the trainable parameters. This loss consists of a quadratic coverage penalty and a linear interval width term, both of which are differentiable with respect to $\theta$.

**Proposition 5** (Convergence to a Stationary Point). *Assume:*

*(A1) Each component of $\mathcal{L}_{total}$ is continuous and piecewise smooth;*

*(A2) The gradient norm is bounded: $\|\nabla_\theta \mathcal{L}_{total}(\theta)\| \leq G$;*

*(A3) The learning rate $\eta_t$ satisfies: $\eta_t \to 0$, $\sum_{t=1}^{\infty} \eta_t = \infty$, and $\sum_{t=1}^{\infty} \eta_t^2 < \infty$.*

*Then the sequence of iterates $\theta_t$ generated by stochastic gradient descent satisfies:*

$$\lim_{t \to \infty} \mathbb{E}[\|\nabla_\theta \mathcal{L}_{total}(\theta_t)\|] = 0.$$

*Proof.* The total loss $\mathcal{L}_{\text{total}}(\theta)$ is composed of a differentiable quadratic coverage term and a linear width penalty, both of which are locally Lipschitz and smooth with respect to $\theta$. As such, $\mathcal{L}_{\text{total}}$ satisfies the regularity conditions required by standard results in stochastic non-convex optimization (Ghadimi & Lan, 2013). Therefore, under assumptions (A1)–(A3), stochastic gradient descent with diminishing step sizes converges to a first-order stationary point in expectation. Similar convergence guarantees also hold for adaptive optimizers such as Adam, provided the gradients remain bounded (Kingma & Ba, 2015; Reddi et al., 2018). □

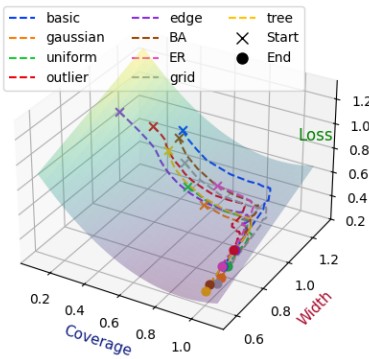
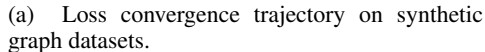
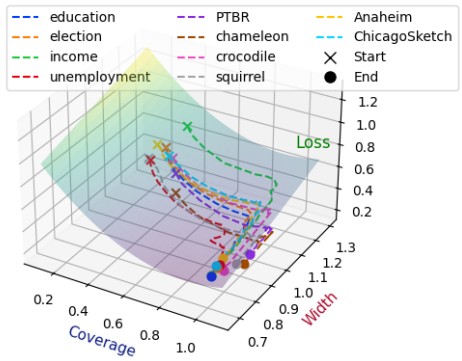

(a) Loss convergence trajectory on synthetic graph datasets.

(b) Loss convergence trajectory on real-world graph datasets.

Figure 7: 3D convergence trajectories in coverage–width–loss space. Each dashed line corresponds to a dataset, tracing optimization from initialization (✗) to convergence (●). The surfaces visualize the loss landscape, while the trajectories highlight stable descent toward calibrated and compact prediction intervals.

**Empirical verification.** We empirically validate the convergence dynamics of QpiGNN through 3D trajectory visualizations on both synthetic and real-world graph datasets. Figure 7 plots training paths in the coverage–width–loss space, showing how the model jointly optimizes calibration and compactness over time. In both settings, training starts with high loss (marked by ✗) and proceeds toward a low-loss region (marked by ●) along a smooth descent trajectory.

On synthetic datasets (Figure 7a), the paths show a structured progression: the model first corrects coverage errors and then reduces interval width, eventually converging near the optimal coverage level $1 - \alpha = 0.9$ while minimizing width. These trajectories confirm the effectiveness of the

sequential optimization behavior embedded in our loss design. On real datasets (Figure 7b), the convergence paths are more varied due to structural heterogeneity and noise. Some datasets exhibit greater fluctuations in coverage during early training, but all converge toward the loss-minimizing surface. This robustness across dataset types highlights the adaptability and stability of QpiGNN's training process.

## C.4 VIOLATION LOSS AND THEORETICAL COVERAGE GUARANTEES

Recall the total coverage-related loss is defined as $\mathcal{L}_{\text{coverage}} = (\hat{c} - (1 - \alpha))^2 + \hat{\ell}_{\text{viol}}$, where the empirical violation loss is

$$\hat{\ell}_{\text{viol}} := \frac{1}{N} \sum_{v=1}^{N} \left[ |y_v - \hat{y}_v^{\text{low}}| \cdot \mathbb{I}[y_v < \hat{y}_v^{\text{low}}] + |y_v - \hat{y}_v^{\text{up}}| \cdot \mathbb{I}[y_v > \hat{y}_v^{\text{up}}] \right].$$

This term penalizes not just whether a coverage violation occurs, but also by how far $y_v$ lies outside the predicted interval. It plays a key role in shaping the model's predictions during training, especially when $\hat{c} \ll 1 - \alpha$. However, note that $\hat{\ell}_{\text{viol}} \xrightarrow{P} 0$ as the model becomes well-calibrated, i.e., when

$$\mathbb{P}(y_v < \hat{y}_v^{\text{low}} \text{ or } y_v > \hat{y}_v^{\text{up}}) \to \alpha.$$

In this regime, the expected violation magnitude vanishes:

$$\mathbb{E}\left[ \hat{\ell}_{\text{viol}} \right] \leq \mathbb{E}\left[ |y_v - \hat{y}_v^{\text{low}}| \cdot \mathbb{I}[y_v < \hat{y}_v^{\text{low}}] + |y_v - \hat{y}_v^{\text{up}}| \cdot \mathbb{I}[y_v > \hat{y}_v^{\text{up}}] \right] \to 0.$$

Hence, $\hat{\ell}_{\text{viol}}$ is asymptotically negligible and does not interfere with the convergence of $\hat{c}$ to the target coverage level $1 - \alpha$. The proof of coverage consistency thus holds even in its absence.

## C.5 ROBUSTNESS OF THEORETICAL GUARANTEES UNDER PRACTICAL CONDITIONS

Our coverage consistency theorem assumes that the samples $X_v = (x_v, y_v)$ are i.i.d., and that model predictions converge under bounded noise. These assumptions, while standard, are idealized. In practical graph settings, particularly with GNNs, message passing introduces weak dependencies among node predictions due to shared neighborhood structures (Verma & Zhang, 2019; Jin et al., 2020).

Nonetheless, we observe that predictions $\hat{y}_v^{\text{low}}, \hat{y}_v^{\text{up}}$ still rely primarily on local features and limited-depth neighborhoods. As a result, these weak dependencies do not significantly impair the empirical convergence behavior of the coverage estimator $\hat{c}$. In particular, our empirical results demonstrate that QpiGNN maintains tight coverage and compact intervals even under node and edge perturbations, supporting the practical validity of the assumptions used in our finite-sample analysis. In fact, the bounded-difference property required by McDiarmid's inequality (McDiarmid, 1989) remains approximately satisfied:

$$\sup_{X_1, \dots, X_N, X_v'} |\hat{c}(X_1, \dots, X_v, \dots, X_N) - \hat{c}(X_1, \dots, X_v', \dots, X_N)| \leq \frac{1}{N} + \delta_G,$$

where $\delta_G$ is a graph-dependent residual term that diminishes under localized message aggregation. Similar forms of concentration under dependency graphs and weak mixing conditions have been studied in, supporting the plausibility of our empirical findings.

**Remark 1.** To build intuition, consider sparse Erdős–Rényi (ER) graphs with average degree $\mathcal{O}(\log N)$. In such graphs, the influence of a single node diminishes rapidly with distance. For example, in GraphSAGE with mean aggregation, each neighbor contributes at most $\mathcal{O}(1/\deg(v))$ per layer, yielding an overall effect of $\mathcal{O}(1/N)$. Over $k$ layers, the cumulative influence remains bounded by $\mathcal{O}(k/N)$. This intuition aligns with prior work showing that the WLLN can hold under weak dependence in graph processes (Gama & Ribeiro, 2019).

Empirically, QpiGNN maintains tight coverage and compact intervals under diverse perturbations, confirming that these assumptions hold in practice (see Section 5).

Moreover, while the theoretical analysis assumes symmetric conditional distributions (e.g., for width optimality), we observe that QpiGNN maintains valid coverage and produces compact intervals even

Table 5: Hyperparameter sensitivity analysis of target-coverage baselines (SQR, RQR, CF-GNN) evaluated on 10 real datasets over 10 independent runs.

| Model (target coverage) | Education | | Election | | Income | | Unemploy. | | Twitch | | Chameleon | | Crocodile | | Squirrel | | Anaheim | | Chicago | |
|---|---|---|---|---|---|---|---|---|---|---|---|---|---|---|---|---|---|---|---|---|---|
| | PCIP | MPIW | PCIP | MPIW | PCIP | MPIW | PCIP | MPIW | PCIP | MPIW | PCIP | MPIW | PCIP | MPIW | PCIP | MPIW | PCIP | MPIW | PCIP | MPIW |
| SQR (0.80) | 0.78 | 0.28 | 0.75 | 0.32 | 0.78 | 0.19 | 0.79 | 0.35 | 0.09 | 0.02 | 0.11 | 0.01 | 0.15 | 0.01 | 0.09 | 0.01 | 0.67 | 0.35 | 0.73 | 0.23 |
| SQR (0.85) | 0.82 | 0.30 | 0.78 | 0.36 | 0.83 | 0.21 | 0.83 | 0.38 | 0.12 | 0.03 | 0.11 | 0.01 | 0.16 | 0.01 | 0.10 | 0.01 | 0.72 | 0.38 | 0.76 | 0.24 |
| SQR (0.95) | 0.88 | 0.36 | 0.85 | 0.43 | 0.87 | 0.23 | 0.88 | 0.44 | 0.12 | 0.03 | 0.14 | 0.02 | 0.20 | 0.01 | 0.09 | 0.01 | 0.74 | 0.43 | 0.77 | 0.27 |
| RQR (0.80) | 0.78 | 0.37 | 0.77 | 0.40 | 0.78 | 0.26 | 0.76 | 0.35 | 0.62 | 0.42 | 0.67 | 0.09 | 0.72 | 0.03 | 0.68 | 0.10 | 0.65 | 0.22 | 0.72 | 0.30 |
| RQR (0.85) | 0.83 | 0.41 | 0.82 | 0.44 | 0.83 | 0.28 | 0.80 | 0.39 | 0.68 | 0.46 | 0.73 | 0.09 | 0.75 | 0.03 | 0.75 | 0.10 | 0.68 | 0.23 | 0.80 | 0.32 |
| RQR(0.95) | 0.94 | 0.65 | 0.93 | 0.60 | 0.94 | 0.42 | 0.90 | 0.51 | 0.83 | 0.56 | 0.92 | 0.17 | 0.86 | 0.05 | 0.94 | 0.19 | 0.83 | 0.62 | 0.91 | 0.51 |
| CF-GNN (0.80) | 0.80 | 2.66 | 0.80 | 0.99 | 0.80 | 2.45 | 0.80 | 2.28 | 0.81 | 2.77 | - | - | - | - | - | - | 0.81 | 2.55 | 0.80 | 2.44 |
| CF-GNN (0.85) | 0.85 | 3.21 | 0.85 | 1.07 | 0.85 | 2.63 | 0.85 | 2.79 | 0.85 | 3.00 | - | - | - | - | - | - | 0.86 | 3.42 | 0.85 | 2.89 |
| CF-GNN (0.95) | 0.95 | 5.10 | 0.95 | 1.44 | 0.95 | 4.48 | 0.95 | 4.77 | 0.95 | 7.91 | - | - | - | - | - | - | 0.96 | 5.65 | 0.95 | 4.78 |

under moderately skewed or heavy-tailed distributions. This suggests that the method is robust to deviations from theoretical assumptions in practice, consistent with observations in robust quantile regression literature (Tagasovska & Lopez-Paz, 2019; Pouplin et al., 2024).

Overall, the purpose of Appendix B.5 is to demonstrate that concentration-based reasoning remains appropriate in the regimes where QpiGNN is evaluated, even though the strict i.i.d. and symmetry assumptions are only approximately satisfied in practical graph settings. A rigorous extension of our theoretical framework to formally incorporate weak dependency models—e.g., via mixing conditions or graph-dependent concentration inequalities—remains an important direction for future work.

# D  ADDITIONAL HYPERPARAMETER SENSITIVITY RESULTS FOR BASELINES

We further evaluated SQR, RQR, and CF-GNN under multiple target-coverage levels to assess whether these baselines can realize different coverage–width trade-offs (Table 5).

Although SQR adjusts its quantile inputs based on the specified target coverage, its achieved coverage consistently fell short of the desired levels across most datasets and collapsed entirely on non-homophilous graphs. RQR exhibited similar behavior: modifying the target coverage produced only minor changes in interval width, while the realized coverage remained largely unchanged and often far below the target.

These results reveal a limitation of current UQ methods for graph regression: similar to OpiGNN, RQR shows limited sensitivity to the penalty magnitude, indicating that coverage may be inherently difficult to control through this mechanism. We therefore highlight this limitation and point to it as an area requiring further investigation to achieve stricter control over target coverage.

# E  DATASET-SPECIFIC $\lambda_{\text{WIDTH}}$ SELECTION

To investigate how the trade-off between calibration and sharpness varies across tasks, we perform dataset-wise tuning of the width penalty coefficient $\lambda_{\text{width}}$ using Bayesian optimization. The results are summarized in Table 14 and visualized in Figure 8.

We observe substantial variation in the optimal $\lambda_{\text{width}}$ across datasets. Synthetic graphs typically favor higher values ($\geq 0.4$) to prevent overly conservative intervals in structured or low-noise settings (e.g., *Tree*, *Edge*). Real-world datasets show greater diversity: high-variability graphs (e.g., *Election*, *Crocodile*) prefer stronger regularization, while noisy or non-homophilous graphs (e.g., *Chameleon*, *Squirrel*) work better with smaller values (0.22–0.28). Notably, similar interval widths can arise from different $\lambda$ values depending on structural factors. These results highlight the need for dataset-specific tuning, and Bayesian optimization offers a principled way to achieve this.

# F  COMPARISON OF L1 AND L2 WIDTH PENALTIES

We further validated this by replacing the L1 width penalty in Equation (5) with an L2 counterpart (Table 6). The L2 penalty consistently yielded much wider intervals—despite achieving similar or even overly conservative coverage—because it magnifies heavy-tailed residuals induced by

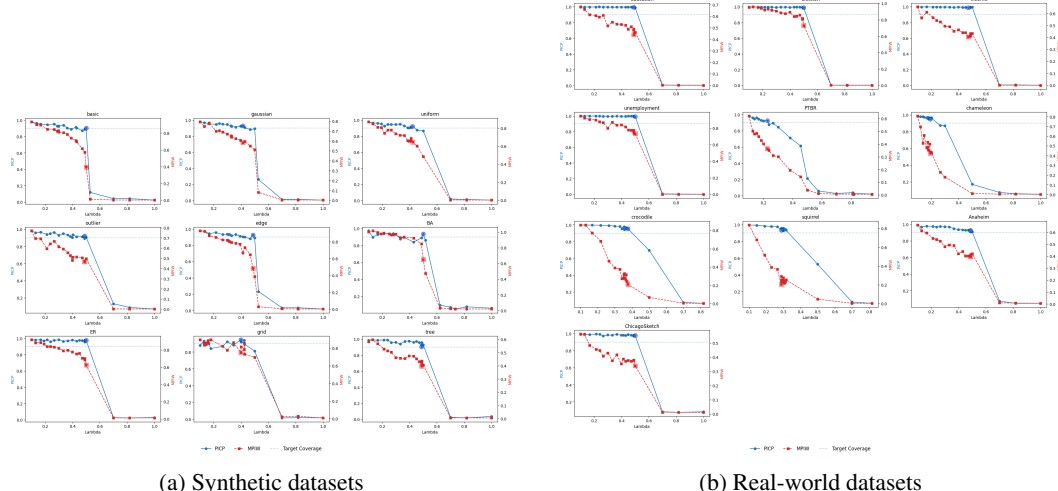

| (a) Synthetic datasets | (b) Real-world datasets |

Figure 8: Optimal values of the width penalty coefficient $\lambda_{\text{width}}$ identified via Bayesian optimization for (a) synthetic and (b) real-world datasets. The results illustrate dataset-specific variability in the calibration–sharpness trade-off, supporting the need for adaptive regularization.

Table 6: Prediction interval performance on 19 synthetic and real datasets. Comparison of L1- and L2-based width penalties in QpiGNN (10 runs, target coverage 90%, $\lambda$ selected optimally).

| Penalty | Basic | | Gaussian | | Uniform | | Outlier | | Edge | | BA | | ER | | Grid | | Tree | |
|---|---|---|---|---|---|---|---|---|---|---|---|---|---|---|---|---|---|---|
| | PCIP | MPIW | PCIP | MPIW | PCIP | MPIW | PCIP | MPIW | PCIP | MPIW | PCIP | MPIW | PCIP | MPIW | PCIP | MPIW | PCIP | MPIW |
| L1 | 0.90 | 0.30 | 0.95 | 0.64 | 0.93 | 0.62 | 0.90 | 0.49 | 0.94 | 0.54 | 0.98 | 0.48 | 0.98 | 0.63 | 0.99 | 0.93 | 0.96 | 0.39 |
| L2 | 0.90 | 0.81 | 0.94 | 0.69 | 0.91 | 0.75 | 0.94 | 0.63 | 0.90 | 0.79 | 0.17 | 0.94 | 0.87 | 0.87 | 0.99 | 1.01 | 0.31 | 1.01 |

| Penalty | Education | | Election | | Income | | Unemploy. | | Twitch | | Chameleon | | Crocodile | | Squirrel | | Anaheim | | Chicago | |
|---|---|---|---|---|---|---|---|---|---|---|---|---|---|---|---|---|---|---|---|---|
| | PCIP | MPIW | PCIP | MPIW | PCIP | MPIW | PCIP | MPIW | PCIP | MPIW | PCIP | MPIW | PCIP | MPIW | PCIP | MPIW | PCIP | MPIW | PCIP | MPIW |
| L1 | 0.99 | 0.57 | 0.98 | 0.77 | 0.99 | 0.44 | 1.00 | 0.73 | 0.94 | 0.36 | 0.96 | 0.23 | 0.97 | 0.16 | 0.96 | 0.18 | 0.93 | 0.40 | 0.98 | 0.36 |
| L2 | 0.99 | 0.68 | 0.99 | 0.79 | 0.99 | 0.43 | 0.98 | 0.86 | 0.97 | 0.75 | 0.97 | 0.70 | 1.00 | 0.69 | 0.97 | 0.53 | 0.94 | 0.52 | 0.94 | 0.45 |

heterophily, structural noise, and message passing. In contrast, the L1 penalty remained stable across all datasets and achieved the target coverage with substantially tighter intervals.

# G    EVALUATION DATASETS

We evaluate QpiGNN and all baselines on a diverse collection of graph-structured datasets, encompassing both synthetic and real-world settings. Table 7 summarizes key statistics across all datasets, including graph size, feature dimensions, and target distribution.

**Synthetic Datasets.**   We generate nine synthetic datasets to assess controlled uncertainty estimation behavior under varying structural and statistical conditions. These datasets include grid, chain, tree, and random graphs, each with distinct patterns of homophily, noise level, and target function. Node features and targets are synthetically generated based on predefined functional relationships (e.g., nonlinear or spatial functions with added noise). The generation scripts and documentation are publicly available on GitHub[9]. These datasets enable controlled ablation and stress testing of uncertainty quantification performance.

**Real-World Datasets.**   We also evaluate on ten publicly available real-world datasets from diverse domains, covering social networks, geographic data, knowledge graphs, and transportation systems.

- **U.S. County-Level Datasets (Education, Election, Income, Unemployment)** are constructed from county-level U.S. maps, using adjacency information derived from geographic boundaries. Node attributes include socioeconomic indicators, and targets reflect

---

[9]https://anonymous.4open.science/r/QpiGNN-11614

Table 7: Statistics of all datasets used in the experiments, including synthetic, real-world, and graph-structured domains. Each entry lists the number of nodes, edges, and input features.

| | Dataset | # Nodes | # Edges | # Features | Dataset | # Nodes | # Edges | # Features |
|---|---|---|---|---|---|---|---|---|
| **Synthetic** | Basic | 1,000 | 2,000 | 5 | Education | 3,234 | 12,717 | 6 |
| | Gaussian | 1,000 | 2,000 | 5 | Election | 3,234 | 12,717 | 6 |
| | Uniform | 1,000 | 2,000 | 5 | Income | 3,234 | 12,717 | 6 |
| | Outlier | 1,000 | 2,000 | 5 | Unemploy. | 3,234 | 12,717 | 6 |
| | Edge | 1,000 | 2,000 | 5 | Twitch | 1,912 | 31,299 | 3,169 |
| | BA | 1,000 | $\sim$2,994 | 5 | Chameleon | 2,277 | 36,101 | 3,132 |
| | ER | 1,000 | $\sim$5,000 | 5 | Crocodile | 11,631 | 180,020 | 13,183 |
| | Grid | 900 | 1,740 | 5 | Squirrel | 5,201 | 217,073 | 3,148 |
| | Tree | 1,000 | 998 | 5 | Anaheim | 914 | 3,638 | 4 |
| | - | | - | | Chicago | 2,176 | 14,961 | 4 |

either vote shares or demographic statistics. The base topology and election outcomes were obtained from an open GitHub repository[10], while additional attributes were sourced from the U.S. Department of Agriculture Economic Research Service[11].

- **Wikipedia & Twitch Graphs (Chameleon, Squirrel, Crocodile, PTBR)** were collected from the MUSAE project (Rozemberczki et al., 2021), which provides temporal and social graphs annotated with node features and continuous targets. These datasets are widely used for benchmarking node regression in non-homophilous graphs[12].

- **Transportation Networks (Anaheim, Chicago)** model urban road networks as graphs, with nodes corresponding to intersections and edges to road segments. Node features include traffic-related metrics, and the targets correspond to flow estimates or congestion levels. These datasets were obtained from the Transportation Networks for Research repository[13].

These datasets span a wide range of graph topologies, feature modalities, and target distributions, providing a robust testbed for evaluating node-level prediction interval quality across domains.

## H  EVALUATION METRICS

To evaluate the quality of UQ, we consider a comprehensive set of metrics that collectively measure two core aspects of prediction intervals: *reliability* (calibration) and *informativeness* (compactness). These metrics enable a nuanced analysis of model performance beyond simple point prediction accuracy.

There is a natural tension between reliability (PICP) and informativeness (MPIW/sharpness): increasing interval width improves coverage but reduces precision. A robust UQ model should strike a balance between the two. Metrics like WS and CWC explicitly model this trade-off, with CWC particularly effective in safety-critical domains where undercoverage must be avoided at all costs.

**Prediction Interval Coverage Probability (PICP)**  PICP (Rana et al., 2015) evaluates whether the model's prediction intervals capture the true target values at the desired rate. It is the proportion of samples whose ground truth values fall within the corresponding predicted intervals:

$$\text{PICP} = \frac{1}{N} \sum_{v=1}^{N} \mathbb{I}\left[\hat{y}_v^{\text{low}} \leq y_v \leq \hat{y}_v^{\text{up}}\right].$$

A well-calibrated model with target coverage $1-\alpha$ should achieve PICP close to that value. However, high PICP alone does not guarantee high-quality intervals if the width is excessive.

---

[10]https://github.com/tonmcg/
[11]https://www.ers.usda.gov/data-products/county-level-data-sets/
[12]https://github.com/benedekrozemberczki/MUSAE
[13]https://github.com/bstabler/TransportationNetworks

**Mean Prediction Interval Width (MPIW)** MPIW (Khosravi et al., 2010a) measures the average width of the predicted intervals:

$$\text{MPIW} = \frac{1}{N} \sum_{v=1}^{N} (\hat{y}_v^{\text{up}} - \hat{y}_v^{\text{low}}).$$

Smaller MPIW indicates tighter (sharper) intervals, which are generally more informative. However, overly narrow intervals risk missing the true target and reducing PICP. Thus, MPIW should be interpreted jointly with PICP.

**Mean Prediction Error (MPE)** MPE measures the average deviation between the center of the predicted interval and the ground truth:

$$\text{MPE} = \frac{1}{N} \sum_{v=1}^{N} \left| \frac{\hat{y}_v^{\text{low}} + \hat{y}_v^{\text{up}}}{2} - y_v \right|.$$

This evaluates whether the intervals are centered correctly. A model with good coverage and sharpness may still perform poorly if it consistently shifts intervals away from the true value.

**Sharpness** Sharpness (Gneiting & Raftery, 2007) refers to the concentration of prediction intervals and penalizes unnecessarily wide intervals:

$$\text{Sharpness} = \frac{1}{N} \sum_{v=1}^{N} \left( \hat{y}_v^{\text{up}} - \hat{y}_v^{\text{low}} \right)^2.$$

It is a stricter version of MPIW that emphasizes outliers by squaring the width. A well-calibrated model should aim for minimal sharpness under valid PICP.

**Winkler Score (WS)** WS (Winkler, 1994) is a proper scoring rule that evaluates both the width of the interval and whether it covers the true target. It imposes a linear penalty on width and an additional penalty on uncovered samples:

$$\text{WS} = \frac{1}{N} \sum_{v=1}^{N} \left[ \left( \hat{y}_v^{\text{up}} - \hat{y}_v^{\text{low}} \right) + \frac{2}{\alpha} \cdot \max(0, \ \hat{y}_v^{\text{low}} - y_v, \ y_v - \hat{y}_v^{\text{up}}) \right].$$

WS is interpretable, sensitive to both undercoverage and over-conservativeness, and widely used in applied settings.

**Combinational Coverage Width-based Criterion (CWC)** CWC (Khosravi et al., 2010b) balances calibration and sharpness with an asymmetric penalty. It penalizes undercoverage exponentially, making it highly sensitive to violations of the target coverage:

$$\text{CWC} = \text{NMPIW} \cdot \left( 1 + \gamma \cdot e^{-\eta(\text{PICP}-\mu)} \right),$$

where $\mu = 1 - \alpha$, $\gamma = 1$, and $\eta = 10$. When PICP falls below $\mu$, the exponential term grows rapidly, strongly penalizing the score.

**Normalized Mean Prediction Interval Width (NMPIW)** NMPIW normalizes MPIW to make interval widths comparable across datasets with different label ranges:

$$\text{NMPIW} = \frac{1}{N} \sum_{v=1}^{N} \frac{\hat{y}_v^{\text{up}} - \hat{y}_v^{\text{low}}}{y_{\max} - y_{\min}}.$$

It is especially useful for cross-dataset comparisons and is a key component of CWC.

## I COMPARED BASELINE MODELS

We compare QpiGNN against five representative baseline models that span distinct paradigms of uncertainty quantification: quantile regression, Bayesian approximation, and conformal prediction. All baselines are adapted to graph-based settings using the same GNN backbone (except CF-GNN$^{opt}$, which includes its own calibration-specific components).

**Simultaneous Quantile Regression (SQR)**   SQR (Tagasovska & Lopez-Paz, 2019) models conditional quantiles by treating the quantile level $\tau \in (0, 1)$ as an input feature. To construct prediction intervals, the model is evaluated twice at $\tau_{\text{low}} = \alpha/2$ and $\tau_{\text{up}} = 1 - \alpha/2$, where $\alpha$ is the miscoverage level. The model is trained using the standard quantile (pinball) loss. While SQR allows simultaneous learning of multiple quantiles through randomized sampling over $\tau$, the model requires explicitly specified quantile levels and may suffer from quantile crossing (Zhou et al., 2020a).

**Relaxed Quantile Regression (RQR)**   RQR (Pouplin et al., 2024) directly predicts both lower and upper bounds of prediction intervals using a shared architecture. The training objective incorporates a composite loss that balances calibration and compactness:

$$\mathcal{L}_{\text{RQR}} = \mathcal{L}_{\text{RQR-W}} + \gamma_{\text{order}} \cdot \text{ReLU}(\hat{\mathbf{y}}^{\text{low}} - \hat{\mathbf{y}}^{\text{up}}),$$

where $\mathcal{L}_{\text{RQR-W}}$ penalizes miscoverage and excessive width, and the ReLU term enforces a soft constraint to avoid interval crossing. $\gamma_{\text{order}} \geq 0$ is a hyperparameter. This ordering penalty is *not included in the original formulation* but added here to improve interval validity in GNN-based tasks. Despite its design, RQR tends to produce overly smooth and wide intervals when applied to GNNs, due to their intrinsic neighborhood averaging and representation homogeneity (Rusch et al., 2023).

**Bayesian Neural Networks (BayesianNN)**   BayesianNN (Kendall & Gal, 2017) models uncertainty via posterior distributions over network weights. During inference, multiple stochastic forward passes are performed to estimate the predictive mean $\mu$ and standard deviation $\sigma$, forming prediction intervals as:

$$[\mu - t \cdot \sigma, \ \mu + t \cdot \sigma], \quad t \text{ chosen to match the desired confidence level.}$$

This method captures both epistemic and aleatoric uncertainty but is computationally expensive and often slow to converge.

**Monte Carlo Dropout (MC dropout)**   MC dropout (Gal & Ghahramani, 2016) offers an approximate Bayesian alternative by retaining dropout during inference. Similar to BayesianNN, the model performs multiple forward passes to compute predictive mean and variance. It is simpler to implement and more scalable, but its uncertainty estimates can be unstable in high-variance regimes.

**Conformalized GNN (CF-GNN)**   CF-GNN (Huang et al., 2023) separates prediction and calibration by first training a base GNN regressor $\hat{\mathbf{y}}$, and then post-calibrating the intervals using conformal prediction (CP) methods. The calibrated prediction intervals is:

$$[\hat{\mathbf{y}} - \hat{q}, \ \hat{\mathbf{y}} + \hat{q}],$$

where $\hat{q}$ is the $(1-\alpha)$-quantile of residuals on a held-out calibration set. CF-GNN is model-agnostic and guarantees valid marginal coverage under the exchangeability assumption. However, it requires an additional calibration dataset and may be sensitive to distribution shifts. We use the official PyTorch-Geometric implementation[14] provided by the authors.

Table 8 summarizes the comparative properties of the five baselines. These models reflect a wide range of UQ approaches—quantile-based, Bayesian, and CP—and highlight the trade-offs in coverage validity, interval compactness, computational efficiency, and graph-awareness. For fairness, we implement all models using the same GNN encoder as QpiGNN, except CF-GNN$^{opt}$, which uses its official architecture and calibration setup.

## J   SUPPLEMENTARY METRIC RESULTS

In addition to the primary evaluation metrics—PICP and MPIW—we report results on four supplementary UQ metrics to provide a more comprehensive evaluation. Table 9 presents results on MPE and CWC. MPE captures the alignment between the center of each predicted interval and the corresponding ground truth value, serving as a proxy for point-prediction accuracy. CWC combines normalized interval width with an exponential penalty on undercoverage, allowing us to assess how

---

[14]Source code: `https://github.com/snap-stanford/conformalized-gnn`

Table 8: Summary of baseline models in terms of strengths, limitations, and computational complexity.

| Model | Strengths | Limitations | Comp. Cost |
|---|---|---|---|
| **SQR** | Fast and simple; enables arbitrary quantile estimation without sampling; compact implementation | No coverage guarantee; intervals may cross; sensitive to quantile choices | Low |
| **RQR** | Predicts both bounds jointly; soft ordering constraint avoids interval crossing; promotes compact intervals | Requires tuning of penalties; prone to over-smoothing when used with GNNs | Moderate |
| **CF-GNN** | Post-hoc calibrated intervals with valid marginal coverage; model-agnostic; flexible scoring rules | Needs separate calibration set; assumes exchangeability; less adaptive to node-level uncertainty | Moderate |
| **BayesianNN** | Theoretically grounded; captures both epistemic and aleatoric uncertainty | High computational cost due to sampling; slow convergence; less scalable | High |
| **MC Dropout** | Easy to integrate; empirically effective in practice; low overhead at training time | Multiple forward passes at inference; unstable under high variance; approximate posterior | Moderate |

Table 9: Comparison of Mean Prediction Error (MPE) and Combinational Coverage Width-based Criterion (CWC) on synthetic and real datasets. Lower values indicate better performance for both metrics. The best result for each dataset is highlighted in **bold**, and the second-best is underlined.

Synthetic

| Model | Basic MPE | Basic CWC | Gaussian MPE | Gaussian CWC | Uniform MPE | Uniform CWC | Outlier MPE | Outlier CWC | Edge MPE | Edge CWC | BA MPE | BA CWC | ER MPE | ER CWC | Grid MPE | Grid CWC | Tree MPE | Tree CWC |
|---|---|---|---|---|---|---|---|---|---|---|---|---|---|---|---|---|---|---|
| SQR-GNN | 0.09 | 1.01 | **0.13** | 1.32 | 0.13 | 1.27 | **0.05** | **0.20** | **0.08** | 0.64 | 0.25 | 3.20 | 0.21 | 4.27 | 0.17 | 2.87 | **0.08** | 1.38 |
| RQR$^{adj.}$-GNN | 0.25 | 1.66 | 0.14 | 1.34 | 0.17 | 1.48 | 0.11 | 0.71 | 0.25 | 1.43 | 0.28 | 3.41 | 0.25 | 1.77 | 0.17 | 3.87 | 0.24 | 2.14 |
| BayesianNN | 0.39 | 4.15 | 0.31 | 4.63 | 0.35 | 4.39 | 0.09 | 3.94 | 0.42 | 4.22 | 0.40 | 4.43 | 0.28 | 4.34 | 0.30 | 4.53 | 0.29 | 4.79 |
| MC dropout | 0.28 | 71.71 | 0.15 | 34.7 | 0.21 | 60.18 | **0.07** | 8.18 | 0.27 | 58.17 | 0.24 | 47.61 | 0.28 | 157.92 | **0.15** | 55.25 | 0.27 | 85.17 |
| CF-GNN | 0.57 | 3.26 | 0.74 | 2.50 | 0.85 | 4.05 | 0.54 | 0.67 | 0.50 | 3.01 | 17.67 | 59.12 | 4.62 | 14.57 | 0.95 | 11.82 | 0.26 | 1.80 |
| QpiGNN ($\lambda^{0.5}$) | **0.08** | 0.63 | **0.13** | **1.15** | 0.11 | **1.02** | 0.18 | 0.99 | 0.10 | 0.65 | **0.11** | 0.75 | 0.15 | 0.97 | **0.16** | **1.39** | **0.08** | **0.70** |
| QpiGNN ($\lambda^{0.1}$) | 0.25 | 1.33 | 0.15 | 1.36 | 0.18 | 1.35 | 0.23 | 1.04 | 0.26 | 1.34 | 0.32 | 1.56 | 0.24 | 1.36 | 0.17 | 1.57 | 0.09 | 0.94 |
| QpiGNN ($\lambda^{opt.}$) | **0.08** | **0.62** | 0.14 | 1.18 | 0.14 | 1.16 | 0.19 | 1.00 | 0.14 | 0.91 | **0.11** | **0.74** | 0.15 | 0.97 | 0.15 | 1.44 | **0.08** | **0.70** |

Real

| Model | Education MPE | Education CWC | Election MPE | Election CWC | Income MPE | Income CWC | Unemploy. MPE | Unemploy. CWC | Twitch MPE | Twitch CWC | Chameleon MPE | Chameleon CWC | Crocodile MPE | Crocodile CWC | Squirrel MPE | Squirrel CWC | Anaheim MPE | Anaheim CWC | Chicago MPE | Chicago CWC |
|---|---|---|---|---|---|---|---|---|---|---|---|---|---|---|---|---|---|---|---|---|
| SQR-GNN | **0.09** | 0.55 | 0.12 | 1.02 | 0.06 | 0.56 | 0.08 | 1.01 | 0.03 | 20.77 | 0.02 | 2.39 | 0.01 | 0.75 | **0.02** | 10.73 | 0.10 | 1.06 | 0.06 | 0.58 |
| RQR$^{adj.}$-GNN | 0.15 | 0.87 | 0.14 | 1.17 | 0.10 | 0.74 | 0.09 | 0.99 | 0.10 | 1.30 | 0.07 | 0.59 | 0.04 | 0.27 | 0.07 | 0.31 | 0.20 | 1.76 | 0.10 | 0.77 |
| BayesianNN | 0.16 | 3.11 | 0.37 | 4.25 | 0.14 | 3.96 | 0.16 | 5.15 | 0.18 | 6.88 | 0.08 | 6.24 | 0.08 | 4.53 | 0.08 | 4.06 | 0.15 | 5.36 | 0.10 | 4.27 |
| MC dropout | 0.15 | 64.77 | 0.14 | 35.59 | 0.11 | 57.30 | 0.09 | 44.11 | 0.12 | 44.28 | 0.04 | 26.19 | 0.03 | 12.55 | 0.04 | 27.65 | 0.18 | 58.96 | 0.10 | 82.75 |
| CF-GNN | 0.72 | 0.81 | 0.26 | 1.34 | 0.75 | 0.97 | 0.75 | 0.87 | 0.77 | 1.26 | - | - | - | - | - | - | 0.89 | 1.37 | 0.75 | 0.89 |
| CF-GNN (opt.) | 0.76 | 0.98 | 0.22 | 1.03 | 0.71 | 0.81 | 0.67 | **0.73** | 0.57 | **0.75** | - | - | - | - | - | - | 1.00 | 1.37 | 0.67 | 0.65 |
| QpiGNN ($\lambda^{0.5}$) | 0.11 | 0.61 | 0.15 | 1.17 | **0.07** | **0.56** | 0.18 | 1.29 | **0.04** | 3.79 | 0.03 | 3.56 | 0.02 | **0.16** | 0.03 | 0.58 | **0.12** | 0.95 | 0.07 | **0.55** |
| QpiGNN ($\lambda^{0.1}$) | 0.23 | 0.96 | 0.17 | 1.39 | 0.13 | 0.96 | 0.23 | 1.61 | 0.11 | 1.28 | 0.09 | 0.91 | 0.06 | 0.80 | 0.08 | 0.66 | 0.19 | 1.37 | 0.11 | 0.87 |
| QpiGNN ($\lambda^{opt.}$) | 0.12 | 0.64 | 0.14 | 1.15 | 0.07 | 0.59 | 0.18 | 1.28 | 0.08 | 1.00 | 0.06 | **0.55** | 0.03 | 0.26 | 0.04 | **0.29** | 0.12 | **0.94** | 0.07 | **0.55** |

well a model balances compactness with reliability. A lower CWC indicates better trade-off handling between sharpness and valid coverage. Table 10 reports results on Sharpness and WS, both of which focus on interval compactness and informativeness. Sharpness penalizes overly wide intervals through a quadratic term, while WS additionally incorporates a coverage-sensitive penalty, making it particularly useful for evaluating practical utility. Models like SQR-GNN and MC dropout achieve low sharpness and WS, indicating tight intervals—but often at the expense of undercoverage.

As discussed in Section 5, such models fail to meet the target coverage threshold (e.g., $1 - \alpha = 0.9$) as shown in Table 1, thereby exposing a critical trade-off between reliability and precision. These additional metrics thus offer a more nuanced view of model performance, enabling a clearer understanding of uncertainty behavior across diverse graph datasets. They also help identify whether a method's compact intervals are meaningfully calibrated or merely overconfident.

# K  QUALITATIVE ANALYSIS

To complement our quantitative evaluation, we present a qualitative comparison of prediction intervals across all models. Figure 9 shows visualizations for nine synthetic datasets, while Figure 10 displays corresponding results for ten real-world graph datasets.

Table 10: Comparison of Sharpness and Winkler Score (WS) on synthetic and real datasets. Lower values indicate better performance for both metrics. The best result for each dataset is highlighted in **bold**, and the second-best is underlined.

| | Model | Basic | | Gaussian | | Uniform | | Outlier | | Edge | | BA | | ER | | Grid | | Tree | |
|---|---|---|---|---|---|---|---|---|---|---|---|---|---|---|---|---|---|---|---|
| | | Sharpness | WS | Sharpness | WS | Sharpness | WS | Sharpness | WS | Sharpness | WS | Sharpness | WS | Sharpness | WS | Sharpness | WS | Sharpness | WS |
| **Synthetic** | SQR-GNN | 0.14 | 0.33 | 0.26 | **0.51** | 0.27 | 0.52 | 0.01 | 0.12 | 0.13 | **0.32** | 0.54 | 0.73 | 0.38 | 0.61 | 0.29 | 0.55 | **0.07** | **0.27** |
| | RQR$^{adj.}$-GNN | 0.67 | 0.82 | 0.29 | 0.54 | 0.46 | 0.68 | 0.13 | 0.37 | 0.68 | 0.84 | 0.57 | 0.76 | 0.60 | 0.78 | 0.25 | 0.52 | 0.46 | 0.68 |
| | BayesianNN | 9.08 | 3.01 | 8.92 | 2.98 | 9.02 | 3.00 | 8.72 | 2.95 | 9.37 | 3.06 | 9.50 | 3.08 | 9.09 | 3.01 | 9.11 | 3.01 | 9.02 | 3.00 |
| | MC dropout | **0.10** | 0.47 | **0.04** | 0.27 | **0.07** | 0.36 | **0.00** | 0.11 | 0.09 | 0.45 | **0.07** | 0.41 | 0.07 | 0.41 | **0.03** | 0.25 | 0.04 | 0.38 |
| | CF-GNN | 4.79 | 1.91 | 9.47 | 2.92 | 10.49 | 3.07 | 4.42 | 2.01 | 3.79 | 1.78 | 10514.81 | 68.69 | 638.45 | 17.26 | 14.91 | 3.19 | 1.31 | 0.98 |
| | QpiGNN ($\lambda^{0.5}$) | **0.10** | **0.30** | 0.31 | 0.55 | 0.19 | 0.44 | 0.23 | 0.48 | 0.17 | 0.39 | 0.31 | 0.49 | 0.42 | 0.63 | 0.76 | 0.87 | 0.15 | 0.39 |
| | QpiGNN ($\lambda^{0.1}$) | 0.86 | 0.93 | 0.71 | 0.84 | 0.79 | 0.89 | 0.56 | 0.75 | 0.94 | 0.97 | 1.03 | 1.01 | 0.85 | 0.92 | 0.95 | 0.98 | 0.35 | 0.59 |
| | QpiGNN ($\lambda^{opt.}$) | **0.10** | 0.31 | 0.42 | 0.65 | 0.39 | 0.62 | 0.25 | 0.49 | 0.30 | 0.54 | 0.30 | 0.48 | 0.42 | 0.63 | 0.87 | 0.93 | 0.15 | 0.39 |

| | Model | Education | | Election | | Income | | Unemploy. | | Twitch | | Chameleon | | Crocodile | | Squirrel | | Anaheim | | Chicago | |
|---|---|---|---|---|---|---|---|---|---|---|---|---|---|---|---|---|---|---|---|---|---|
| | | Sharpness | WS | Sharpness | WS | Sharpness | WS | Sharpness | WS | Sharpness | WS | Sharpness | WS | Sharpness | WS | Sharpness | WS | Sharpness | WS | Sharpness | WS |
| **Real** | SQR-GNN | 0.11 | 0.33 | 0.22 | 0.48 | 0.05 | 0.23 | 0.11 | 0.34 | **0.00** | 0.05 | **0.00** | 0.03 | **0.00** | 0.01 | **0.00** | 0.03 | 0.12 | 0.32 | 0.05 | 0.22 |
| | RQR$^{adj.}$-GNN | 0.24 | 0.50 | 0.29 | 0.55 | 0.13 | 0.37 | 0.15 | 0.39 | 0.18 | 0.42 | 0.02 | 0.16 | 0.01 | 0.09 | 0.02 | 0.16 | 0.25 | 0.51 | 0.09 | 0.31 |
| | BayesianNN | 8.78 | 2.96 | 8.93 | 2.98 | 8.83 | 2.97 | 8.87 | 2.98 | 9.44 | 3.07 | 8.73 | 2.95 | 9.45 | 3.07 | 8.85 | 2.97 | 8.70 | 2.94 | 8.96 | 2.99 |
| | MC dropout | **0.01** | **0.14** | **0.03** | 0.23 | **0.01** | **0.11** | **0.01** | 0.13 | 0.02 | 0.15 | **0.00** | 0.04 | **0.00** | 0.02 | **0.00** | 0.04 | **0.01** | **0.14** | 0.01 | 0.09 |
| | CF-GNN | 8.78 | 2.83 | 1.25 | 1.10 | 14.33 | 3.55 | 10.77 | 3.24 | 14.33 | 3.56 | - | - | - | - | - | - | 11.49 | 3.28 | 11.11 | 3.18 |
| | CF-GNN (opt.) | 11.82 | 3.14 | 0.90 | 0.96 | 8.63 | 2.97 | 6.88 | 2.69 | 5.58 | 2.38 | - | - | - | - | - | - | 8.17 | 2.86 | 5.56 | 2.31 |
| | QpiGNN ($\lambda^{0.5}$) | 0.35 | 0.57 | 0.62 | 0.78 | 0.19 | 0.41 | 0.56 | 0.74 | 0.01 | 0.10 | **0.00** | 0.05 | 0.01 | 0.08 | 0.01 | 0.08 | 0.18 | 0.40 | 0.15 | 0.36 |
| | QpiGNN ($\lambda^{0.1}$) | 0.80 | 0.90 | 0.93 | 0.97 | 0.52 | 0.72 | 0.87 | 0.93 | 0.32 | 0.54 | 0.18 | 0.41 | 0.32 | 0.54 | 0.24 | 0.47 | 0.56 | 0.74 | 0.37 | 0.60 |
| | QpiGNN ($\lambda^{opt.}$) | 0.38 | 0.59 | 0.61 | 0.78 | 0.22 | 0.44 | 0.55 | 0.73 | 0.15 | 0.37 | 0.06 | 0.23 | 0.03 | 0.16 | 0.04 | 0.19 | 0.18 | 0.40 | 0.16 | 0.36 |

Table 11: Comparison of average computational resource consumption across models during real-world dataset training.

| | Model | Education | Election | Income | Unemploy. | Twitch | Chameleon | Crocodile | Squirrel | Anaheim | Chicago |
|---|---|---|---|---|---|---|---|---|---|---|---|
| **Training Time (sec)** | SQR-GNN | 1.39 | 1.43 | 1.61 | 1.52 | 1.83 | 1.83 | 13.76 | 2.98 | 1.87 | 1.93 |
| | RQR$^{adj.}$-GNN | 2.06 | 1.99 | 1.79 | 2.39 | 2.10 | 2.00 | 31.51 | 7.81 | 1.78 | 1.71 |
| | BayesianNN | 1.32 | 1.28 | 1.31 | 1.32 | 1.34 | 1.41 | 21.34 | 5.36 | 1.48 | 1.26 |
| | MC dropout | 1.83 | 1.83 | 1.82 | 1.37 | 1.77 | 1.53 | 18.37 | 2.92 | 1.70 | 1.80 |
| | CF-GNN | 3.63 / 9.77 | 3.44 / 9.86 | 3.70 / 9.80 | 3.48 / 9.73 | 4.83 / 9.61 | - | - | - | 3.48 / 8.21 | 3.27 / 9.06 |
| | QpiGNN | 1.97 | 1.84 | 1.74 | 1.70 | 1.63 | 1.84 | 14.87 | 5.28 | 1.92 | 1.87 |
| **CPU (MB)** | SQR-GNN | 68.56 | 68.56 | 68.56 | 68.56 | 401.34 | 447.80 | 8069.80 | 2016.32 | 65.74 | 68.04 |
| | RQR$^{adj.}$-GNN | 286.70 | 286.70 | 286.70 | 286.70 | 382.78 | 425.81 | 7601.41 | 1966.33 | 82.11 | 167.08 |
| | BayesianNN | 68.47 | 68.47 | 68.47 | 68.47 | 382.77 | 425.79 | 7601.32 | 1966.30 | 65.72 | 67.99 |
| | MC dropout | 69.23 | 69.23 | 69.23 | 69.23 | 382.77 | 425.79 | 7601.61 | 1966.29 | 65.94 | 68.49 |
| | CP | 67.58 | 67.58 | 67.58 | 67.58 | 299.72 | 331.93 | 5712.37 | 1340.74 | 65.65 | 67.02 |
| | CF-GNN | 70.70 / 69.53 | 70.70 / 69.53 | 70.70 / 69.53 | 70.70 / 69.52 | 523.31 / 521.99 | - | - | - | 66.02 / 65.73 | 70.27 / 69.54 |
| | QpiGNN | 68.48 | 68.48 | 68.48 | 68.48 | 382.78 | 425.80 | 7601.44 | 1966.31 | 1306.90 | 68.00 |
| **Parameters (1,000)** | SQR-GNN | 9.28 | 9.28 | 9.28 | 9.28 | 414.27 | 409.41 | 1695.94 | 411.46 | 9.03 | 9.03 |
| | RQR$^{adj.}$-GNN | 9.22 | 9.22 | 9.22 | 9.22 | 414.21 | 409.35 | 1695.87 | 411.39 | 8.96 | 8.96 |
| | BayesianNN | 9.22 | 9.22 | 9.22 | 9.22 | 414.21 | 409.35 | 1695.87 | 411.39 | 8.96 | 8.96 |
| | MC dropout | 9.15 | 9.15 | 9.15 | 9.15 | 414.15 | 409.28 | 1695.81 | 411.33 | 8.90 | 8.90 |
| | CP | 9.15 | 9.15 | 9.15 | 9.15 | 414.15 | 409.28 | 1695.81 | 411.33 | 8.90 | 8.90 |
| | CF-GNN | 1.22 | 1.22 | 1.22 | 1.22 | 406.21 | - | - | - | 0.96 | 0.96 |
| | QpiGNN | 9.22 | 9.22 | 9.22 | 9.22 | 414.21 | 409.35 | 1695.87 | 411.39 | 8.96 | 8.96 |

QpiGNN achieves a favorable trade-off between calibration and compactness. On the Tree dataset, SQR and MC Dropout produce narrow intervals that miss several true targets—demonstrating under-coverage—while RQR generates globally smooth but overly wide intervals that fail to capture local uncertainty variations. BayesianNN and CF-GNN ensure full coverage by expanding the intervals significantly, resulting in conservative but less informative predictions. In contrast, QpiGNN adaptively adjusts its interval widths, slightly increasing them when needed to satisfy the target coverage $(1 - \alpha = 0.9)$, while preserving interval sharpness where uncertainty is low.

These trends persist on the real-world Anaheim dataset, where QpiGNN again balances calibration and informativeness. The ability to modulate interval width in response to local uncertainty allows QpiGNN to avoid both undercoverage and overconservativeness, unlike other methods that either overfit to fixed-width regimes or fail to generalize.

## L   COMPUTATIONAL EFFICIENCY AND COMPLEXITY ANALYSIS

Table 11 presents a comparison of computational resource usage across all models during training on real-world graph datasets, including training time, peak memory usage, and model size, all measured using an NVIDIA Tesla V100 GPU. The results highlight the trade-offs between efficiency and uncertainty estimation quality across different approaches. We present a theoretical comparison of

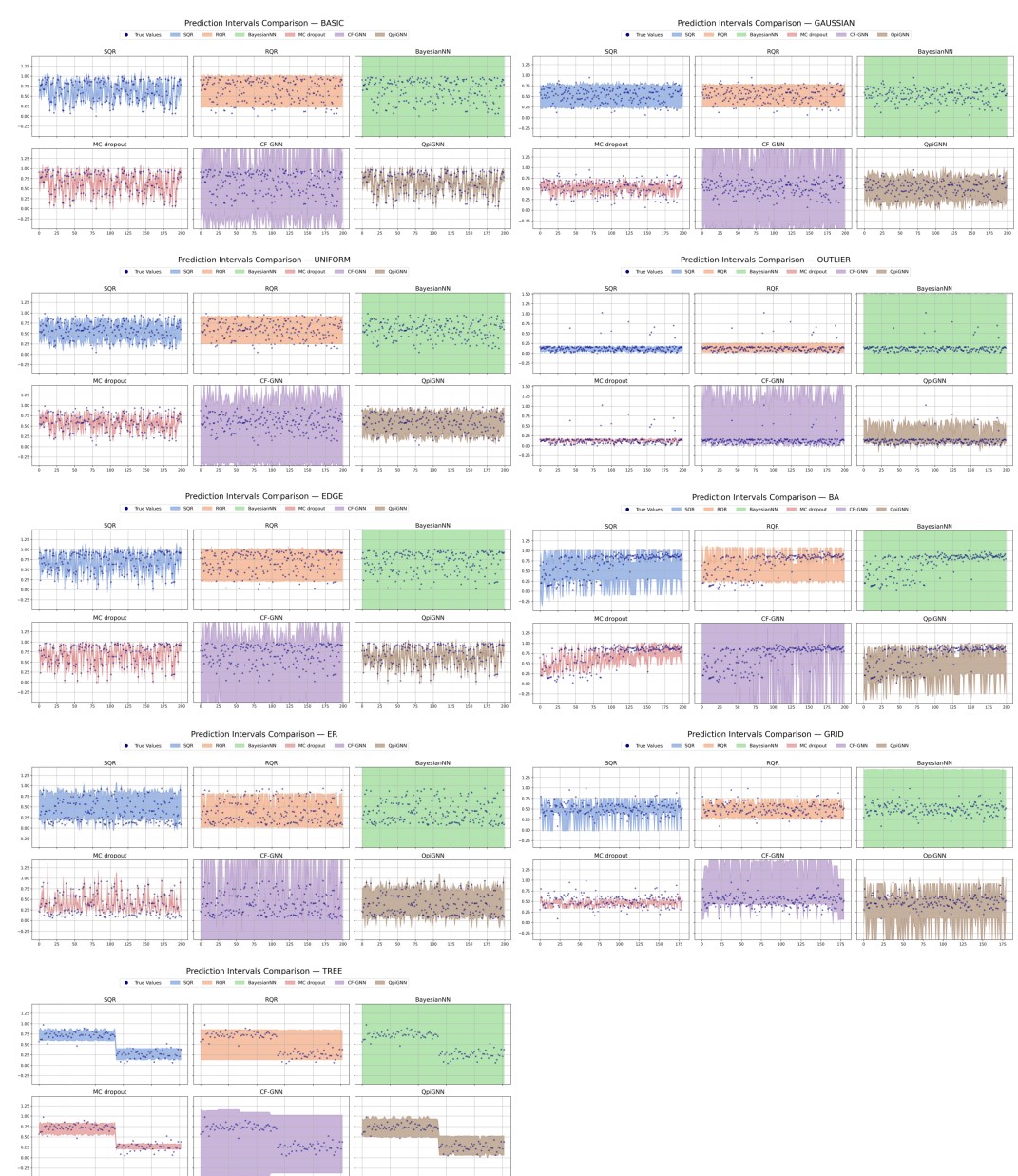

Figure 9: Prediction interval comparison across 9 synthetic datasets for qualitative analysis.

the time complexity of all baseline models considered in this study. Let $N$ and $E$ denote the number of nodes and edges in the graph and $d$ the hidden dimension of the GNN layers.

QpiGNN, RQR$^{adj.}$-GNN, and SQR-GNN share a common two-layer GraphSAGE (Hamilton et al., 2017) backbone and differ only in their output heads or loss formulations. Their time complexity is identical $\mathcal{O}(Ed + Nd^2)$. MC dropout requires $T$ stochastic forward passes at inference time to approximate uncertainty, resulting in a total complexity of $\mathcal{O}(T \cdot (Ed + Nd^2))$. Bayesian Neural Networks (BayesianNN) introduces posterior sampling in the final layer, incurring additional cost per forward pass $\mathcal{O}(Ed + Nd^2 + Nd)$. Conformalized GNN (CF-GNN) attaches a separate multi-layer GNN module (ConfGNN) for calibration on top of the base predictor. Let $d'$ and $L$ denote the hidden dimension and number of layers in ConfGNN, respectively. The resulting complexity is $\mathcal{O}(Ed + Nd^2 + L \cdot (Ed' + Nd'^2))$. This auxiliary component significantly increases both runtime and memory requirements, which may limit scalability, particularly for large or dense graphs.

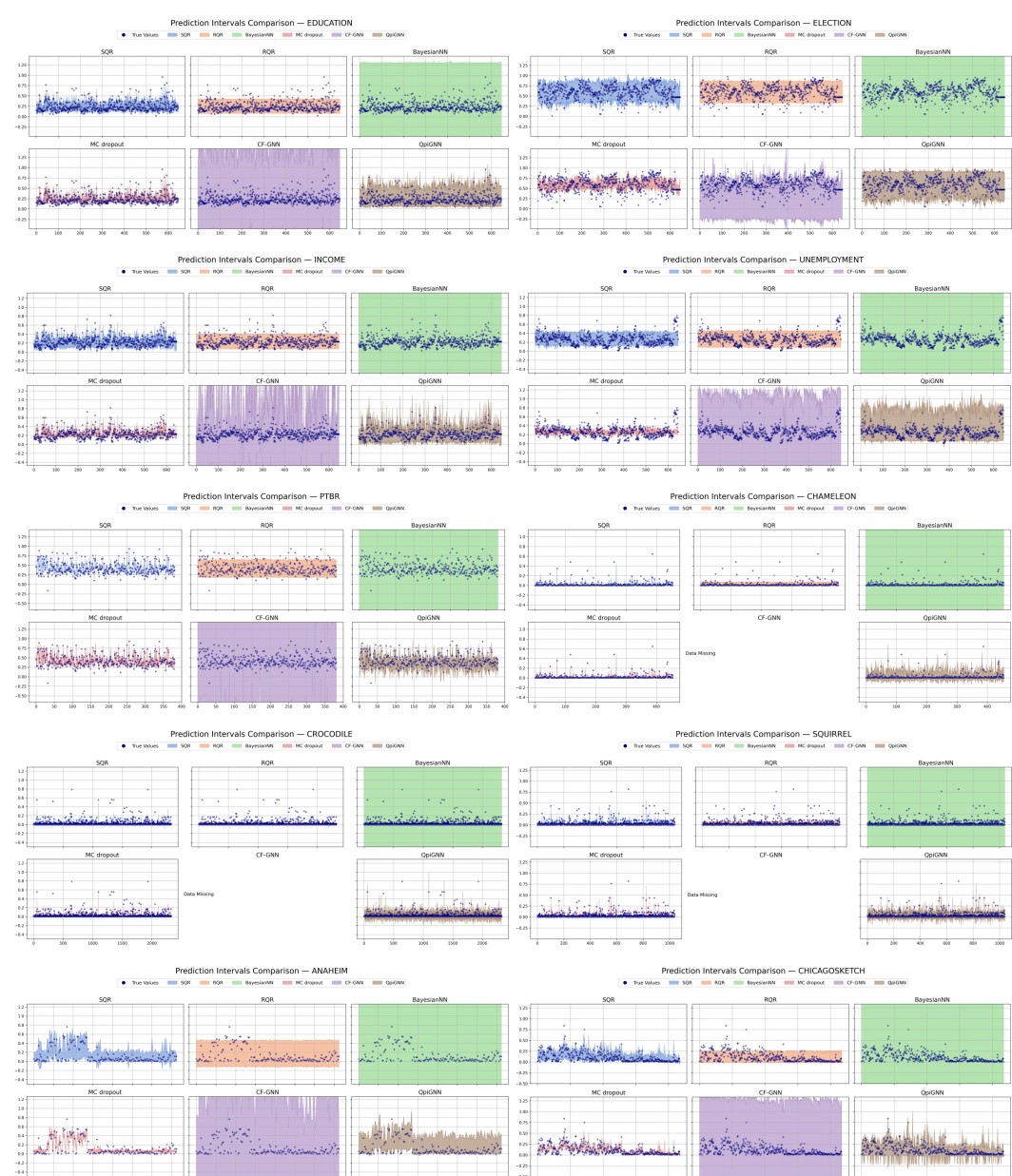

Figure 10: Prediction interval comparison across 10 real datasets for qualitative analysis.

QpiGNN and its quantile-based variants (RQR, SQR) offer strong computational efficiency by avoiding repeated forward passes or additional modules. In contrast, MC dropout and CF-GNN introduce notable overhead due to ensemble-style inference and dual-network design, respectively. This trade-off underscores the practical scalability advantage of QpiGNN for efficient uncertainty estimation in graph learning.

## M   STRUCTURAL SHIFT GENERALIZATION

To evaluate the generalization capacity of QpiGNN under distribution shift, we conduct an out-of-distribution (OOD) experiment across synthetic graph types. Specifically, we train the model on a single graph type and evaluate it on all other types, without further fine-tuning or recalibration. Table 12 presents the results. Each row indicates the graph type used for training, while columns

Table 12: Generalization performance across different synthetic graph datasets. Each cell reports the average PICP and MPIW over 10 runs with a fixed width penalty $\lambda_{\text{width}} = 0.5$.

| Target | PICP | | | | | MPIW | | | | |
|---|---|---|---|---|---|---|---|---|---|---|
| | BA | ER | basic | grid | tree | BA | ER | basic | grid | tree |
| BA | 0.8715 | 0.9720 | 0.7769 | 0.9527 | 0.9627 | 0.5549 | 1.2381 | 1.0600 | 1.3330 | 1.1843 |
| ER | 0.8324 | 0.9289 | 0.6719 | 0.8627 | 0.8892 | 0.7489 | 0.6280 | 0.8749 | 0.9655 | 0.9233 |
| basic | 0.1000 | 0.2750 | 0.9000 | 0.8277 | 0.4313 | 0.3060 | 0.4040 | 0.4018 | 0.5048 | 0.4799 |
| grid | 0.7064 | 0.6385 | 0.5250 | 0.8250 | 0.6578 | 0.8479 | 0.8186 | 0.8099 | 0.8072 | 0.8335 |
| tree | 0.1045 | 0.5625 | 0.2495 | 0.5644 | 0.9058 | 0.3884 | 0.4299 | 0.3389 | 0.3861 | 0.3905 |

Table 13: Ablation results on 19 synthetic and real datasets. ✓ denotes an enabled component. Each result is averaged over 5 runs of 500 epochs. The width penalty factor is set to $\lambda_{\text{width}} = 0.5$. The first column shows PICP and the second shows MPIW. **Bold** indicates models achieving PICP $\geq 0.9$ (target coverage $1-\alpha$), and among them, the configuration with the lowest MPIW is highlighted. *Note: The "No Loss" setting defaults to MSE loss over central prediction.*

| Model Variant | Dual | Coverage | Width | Basic | | Gaussian | | Uniform | | Outlier | | Edge | | ER | | BA | | grid | | tree | |
|---|---|---|---|---|---|---|---|---|---|---|---|---|---|---|---|---|---|---|---|---|---|
| | | | | PICP | MPIW | PICP | MPIW | PICP | MPIW | PICP | MPIW | PICP | MPIW | PICP | MPIW | PICP | MPIW | PICP | MPIW | PICP | MPIW |
| *Architecture-Level Ablation* | | | | | | | | | | | | | | | | | | | | | |
| Dual Head (learned diff) | ✓ | ✓ | ✓ | **0.91** | 0.31 | **0.92** | 0.53 | 0.88 | 0.43 | 0.88 | 0.47 | **0.93** | 0.43 | **0.98** | 0.58 | **0.98** | 0.45 | **0.98** | 0.86 | **0.96** | 0.39 |
| Fixed Margin (0.05) | ✓ | ✓ | ✓ | 0.72 | 0.10 | 0.29 | 0.10 | 0.25 | 0.10 | 0.68 | 0.10 | 0.77 | 0.10 | 0.64 | 0.10 | 0.43 | 0.10 | 0.18 | 0.10 | 0.32 | 0.10 |
| Fixed Margin (0.10) | ✓ | ✓ | ✓ | **0.94** | 0.20 | 0.51 | 0.20 | 0.51 | 0.20 | 0.68 | 0.20 | **0.97** | 0.20 | 0.84 | 0.20 | 0.73 | 0.20 | 0.38 | 0.20 | 0.67 | 0.20 |
| Single Head (low/high sep.) | ✗ | ✓ | ✓ | 0.46 | 0.07 | 0.25 | 0.09 | 0.25 | 0.10 | 0.71 | 0.23 | 0.58 | 0.07 | 0.78 | 0.19 | 0.32 | 0.11 | 0.26 | 0.14 | 0.06 | 0.02 |
| *Loss-Level Ablation* | | | | | | | | | | | | | | | | | | | | | |
| Full Loss | ✓ | ✓ | ✓ | **0.91** | 0.31 | **0.92** | 0.53 | 0.88 | 0.43 | 0.88 | 0.47 | **0.93** | 0.43 | **0.98** | 0.58 | **0.98** | 0.45 | **0.98** | 0.86 | **0.96** | 0.39 |
| Coverage Only | ✓ | ✓ | ✗ | **1.00** | 1.24 | **1.00** | 1.18 | **1.00** | 1.21 | **0.99** | 1.13 | **1.00** | 1.30 | **0.99** | 1.09 | **0.97** | 1.16 | **0.99** | 1.14 | **1.00** | 1.10 |
| Width Only | ✓ | ✗ | ✓ | 0.00 | 0.00 | 0.00 | 0.00 | 0.00 | 0.00 | 0.00 | 0.00 | 0.00 | 0.00 | 0.00 | 0.00 | 0.00 | 0.01 | 0.00 | 0.02 | 0.00 | 0.01 |
| No Loss (Sanity Check) | ✓ | ✗ | ✗ | **1.00** | 1.00 | **1.00** | 1.00 | **1.00** | 1.00 | **0.98** | 1.00 | **1.00** | 1.00 | **0.99** | 1.00 | **0.99** | 1.00 | **1.00** | 1.00 | **1.00** | 1.00 |

correspond to the test graph types. We report the PICP and MPIW, averaged over 10 independent runs using a fixed-width penalty $\lambda_{\text{width}} = 0.5$. Higher PICP reflects better coverage (calibration), and lower MPIW indicates sharper, more compact intervals.

QpiGNN shows the strongest generalization when trained on expressive graphs such as *BA* and *ER*, achieving high PICP scores on multiple unseen target types (e.g., BA→ER: 0.9720, ER→grid: 0.8627). However, this often comes at the cost of wider intervals (e.g., MPIW $\geq 1.2$). In contrast, models trained on simple graphs such as *basic* or *tree* tend to under-cover other graph types despite producing narrow intervals (e.g., basic→ BA: PICP = 0.1000, MPIW = 0.3060). Structurally rich graphs improve generalization but lead to wider intervals, while simple graphs yield narrower yet poorly calibrated intervals—highlighting the need for expressive source graphs in uncertainty transfer.

# N  ABLATION STUDY RESULTS

To better understand the contributions of individual components within QpiGNN, we conduct a comprehensive ablation study across nine synthetic graph datasets. Table 13 presents results from both architecture-level and loss-level ablations. Each configuration is evaluated over five runs for 500 training epochs, using a fixed width penalty coefficient $\lambda_{\text{width}} = 0.5$.

**Architecture-Level Ablation.** We first evaluate different architectural choices. The full *Dual Head* model with learned margin prediction (i.e., separate heads for mean and interval width) achieves strong performance across all datasets, consistently satisfying the target coverage level ($1 - \alpha = 0.9$) while minimizing interval width (lowest MPIW in most cases). In contrast:

- The Fixed Margin variants fail to adapt to dataset-specific uncertainty patterns, leading to either severe undercoverage (e.g., Gaussian, Uniform) or overestimation of width (e.g., Tree).

| Model Variant | Dual | Coverage | Width | Education | | Election | | Income | | Unemploy. | | Twitch | | Chameleon | | Crocodile | | Squirrel | | Anaheim | | Chicago | |
|---|---|---|---|---|---|---|---|---|---|---|---|---|---|---|---|---|---|---|---|---|---|---|---|
| | | | | PICP | MPIW | PICP | MPIW | PICP | MPIW | PICP | MPIW | PICP | MPIW | PICP | MPIW | PICP | MPIW | PICP | MPIW | PICP | MPIW | PICP | MPIW |
| *Architecture-Level Ablation* | | | | | | | | | | | | | | | | | | | | | | | |
| Dual Head (learned diff) | ✓ | ✓ | ✓ | **0.99** | 0.56 | **0.99** | 0.82 | **0.99** | 0.40 | **0.99** | 0.70 | 0.60 | 0.09 | 0.61 | 0.04 | **0.94** | 0.09 | 0.71 | 0.07 | **0.93** | 0.40 | **0.97** | 0.36 |
| Fixed Margin (0.05) | ✓ | ✓ | ✓ | 0.45 | 0.10 | 0.30 | 0.10 | 0.59 | 0.10 | 0.49 | 0.10 | 0.71 | 0.10 | **0.91** | 0.10 | **0.94** | 0.10 | 0.88 | 0.10 | 0.50 | 0.10 | 0.64 | 0.10 |
| Fixed Margin (0.10) | ✓ | ✓ | ✓ | 0.72 | 0.20 | 0.56 | 0.20 | 0.83 | 0.20 | 0.78 | 0.20 | **0.90** | 0.20 | **0.95** | 0.20 | **0.98** | 0.20 | **0.97** | 0.20 | 0.77 | 0.20 | 0.84 | 0.15 |
| Single Head (low/high sep.) | ✗ | ✓ | ✓ | 0.27 | 0.07 | 0.37 | 0.14 | 0.81 | 0.22 | 0.55 | 0.12 | 0.21 | 0.02 | 0.48 | 0.02 | 0.85 | 0.03 | 0.43 | 0.03 | 0.43 | 0.11 | 0.67 | 0.15 |
| *Loss-Level Ablation* | | | | | | | | | | | | | | | | | | | | | | | |
| Full Loss | ✓ | ✓ | ✓ | **0.99** | 0.56 | **0.99** | 0.82 | **0.99** | 0.40 | **0.99** | 0.70 | 0.60 | 0.09 | 0.61 | 0.04 | **0.94** | 0.09 | 0.71 | 0.07 | **0.93** | 0.40 | **0.97** | 0.36 |
| Coverage Only | ✓ | ✓ | ✗ | **1.00** | 1.12 | **1.00** | 1.13 | **1.00** | 1.11 | **1.00** | 1.12 | **1.00** | 1.17 | **1.00** | 1.01 | **1.00** | 1.03 | **1.00** | 1.06 | **1.00** | 1.13 | **1.00** | 1.11 |
| Width Only | ✓ | ✗ | ✓ | 0.00 | 0.00 | 0.00 | 0.00 | 0.00 | 0.00 | 0.00 | 0.00 | 0.00 | 0.00 | 0.21 | 0.00 | 0.29 | 0.00 | 0.08 | 0.00 | 0.08 | 0.00 | 0.05 | 0.01 |
| No Loss (Sanity Check) | ✓ | ✗ | ✗ | **1.00** | 1.00 | **1.00** | 1.00 | **1.00** | 1.00 | **1.00** | 1.00 | **1.00** | 1.00 | **1.00** | 1.00 | **1.00** | 1.00 | **1.00** | 1.00 | **1.00** | 1.00 | **1.00** | 1.00 |

Table 14: Optimal $\lambda_{\text{width}}$ and corresponding MPIW for each dataset obtained via Bayesian optimization.

| Synthetic Dataset | Best $\lambda_{\text{width}}$ | MPIW | Real Dataset | Best $\lambda_{\text{width}}$ | MPIW |
|---|---|---|---|---|---|
| Basic | 0.5000 | 0.39 | Education | 0.4898 | 0.44 |
| Gaussian | 0.4138 | 0.58 | Election | 0.5000 | 0.73 |
| Uniform | 0.4219 | 0.65 | Income | 0.2264 | 0.36 |
| Outlier | 0.4875 | 0.46 | Unemploy. | 0.1930 | 0.42 |
| Edge | 0.4876 | 0.48 | PTBR | 0.3761 | 0.21 |
| BA | 0.5000 | 0.65 | Chameleon | 0.2875 | 0.21 |
| ER | 0.5000 | 0.62 | Crocodile | 0.4871 | 0.39 |
| Grid | 0.3994 | 0.75 | Squirrel | 0.2875 | 0.21 |
| Tree | 0.4879 | 0.39 | Anaheim | 0.4871 | 0.40 |
| - | - | | Chicago | 0.5000 | 0.34 |

- The Single Head model, where upper and lower bounds are predicted independently, shows unstable behavior, often resulting in poor calibration and highly variable coverage.

These results confirm the importance of a dedicated dual-head architecture that allows flexible, learned interval widths conditioned on node features.

**Loss-Level Ablation.** Next, we assess the role of each loss component:

- The Full Loss (coverage + width + violation penalty) yields a consistent balance between calibration and compactness across all graphs.

- The Coverage-Only variant achieves perfect coverage on all datasets but produces excessively wide intervals (high MPIW and CWC), sacrificing informativeness for reliability.

- The Width-Only variant fails entirely, collapsing all intervals to zero, resulting in complete undercoverage—demonstrating that coverage loss is essential to meaningful UQ.

- The No Loss setting (MSE over center prediction, used as a sanity check) trivially learns to output wide enough intervals to contain all targets, but lacks meaningful control over sharpness or adaptive behavior.

Taken together, these results justify the design of QpiGNN's learning objective and architecture. Both the dual-head structure and the full composite loss are critical for achieving high-quality, well-calibrated, and compact prediction intervals across diverse graph topologies.