# OpenReview forum: "Quantile-Free Uncertainty Quantification in Graph Neural Networks"
_ICLR.cc/2026/Conference — Submitted to ICLR 2026_

### Official Review · Reviewer_xHf3 · 2025-10-19

**Soundness:** 3
**Presentation:** 3
**Contribution:** 3
**Rating:** 6
**Confidence:** 3

**Summary:**

The paper introduces QpiGNN, a novel framework for uncertainty quantification in GNNs. Unlike Bayesian or post-hoc conformal methods that rely on exchangeability or resampling, QpiGNN uses a dual-head GNN to decouple prediction and uncertainty estimation. They also provide theoretical coverage guarantees (asymptotic and finite-sample) under mild dependence assumptions.

**Strengths:**

### Clear motivation
* UQ in GNNs is still underexplored and practically critical. The paper correctly identifies the limitations of current quantile-based and conformal approaches in graph settings

### Novelty
* Quantile-free uncertainty learning is a clean formulation. The dual-head architecture effectively decouples prediction and uncertainty, addressing oversmoothing problems in GNNs

### Theoretical analysis
* Asymptotic coverage and finite-sample concentration results are clearly stated, with mild assumptions that are reasonable in practice

### Experiments
* This paper provides extensive experiments to support their claims (clear comparison against strong baselines)

**Weaknesses:**

### Graph dependence
* The coverage guarantees rely on weak dependence assumptions, but the treatment remains at a high level. There’s no rigorous dependence model or formal graph-specific concentration analysis beyond citing WLLN

### Comparison with conformal prediction theory
* While CP methods are used as baselines, the paper does not deeply analyze why QpiGNN succeeds under violations of exchangeability. A theoretical comparison would strengthen the claims

### Positioning
* While the quantile-free formulation is original in GNNs, it builds closely on RQR. The conceptual gap between RQR and QpiGNN could be articulated more strongly

**Questions:**

Please see the above weaknesses

---

> ### Author Response · Authors · 2025-11-21
> **Response to Reviewer xHf3's Weaknesses**
>
> Thank you for recognizing the strength of our motivation, the importance of UQ in GNNs, and the contributions of our quantile-free design, theory, and empirical evaluation. We hope the following points resolve the remaining concerns. We will update the manuscript to ensure these points are clearly emphasized.
>
> ---
> > **W1. Graph dependence**
>
> We agree that our initial discussion of graph dependence was high-level and appreciate the opportunity to clarify this aspect. Our guarantees rely on a stability-based argument rather than independence or exchangeability assumptions, and we will make this more explicit.
>
> ```Graph-aware bounded-difference property```\
> In standard k-hop message-passing GNNs with normalized aggregation, changing a single node feature affects only a limited k-hop neighborhood, and each aggregation step is Lipschitz. This yields $\big| \hat{c}(X) - \hat{c}(X^{(v)}) \big| \le O(1/N) + O(1/|\mathcal{N}(v)|)$, which directly provides the bounded-difference term $\tfrac{1}{N} + \delta_G$.
>
> This bounded-difference stability is the *only* condition required for our concentration guarantees—no mixing, sparsity, or independence assumptions are needed. This stands in contrast to graph-based CP methods that rely much more heavily on exchangeability.
>
>
> ```Empirical validation across graph types```\
> Although the theory does not specify a dependence model for each real-world graph, we empirically verify the assumed stability on diverse graph families (homophilic, heterophilic, spatial–temporal). QpiGNN consistently maintains calibrated coverage and stable interval widths. We will clarify that our theory + broad empirical evidence jointly support the applicability of QpiGNN under graph dependence.
>
> We appreciate the reviewer’s point and will strengthen both the theoretical explanation and its connection to practical graph structures.
>
> ---
> > **W2. Comparison with conformal prediction theory**
>
> We agree that contrasting QpiGNN more clearly with CP-based methods strengthens the contribution. Our central point—which we will emphasize more explicitly—is that QpiGNN’s guarantees depend on stability (bounded differences), not exchangeability, making its theoretical basis fundamentally different from CP.
>
> ```Why QpiGNN works without exchangeability```\
> CP-based methods require node-level exchangeability for valid coverage. In contrast, QpiGNN learns prediction and interval width jointly through a quantile-free objective and dual-head architecture. Concentration of the empirical coverage follows directly from the bounded-difference property above, not from i.i.d. or exchangeability assumptions. This explains why QpiGNN remains valid under arbitrary graph-dependent sampling schemes.
>
> ```Practical evidence under non-exchangeable splits```\
> QpiGNN maintains stable coverage across random, degree-based, and community-based splits—settings that directly violate exchangeability and where CP-based baselines lose guarantees. The results illustrate that our stability-based reasoning—not exchangeability—drives QpiGNN’s reliability. We will streamline the presentation of these findings in the revision.
>
> | Split Type | PICP | MPIW |
> | ---------- | -------- | ----- |
> | Random     | 0.98–1.00   | 0.40–0.97    |
> | Degree     | 0.94–1.00   | 0.44–0.87    |
> | Community  | 0.95–1.00   | 0.46–0.86    |
>
> We will make this theoretical distinction explicit to clarify why QpiGNN succeeds where CP methods have inherent limitations.
>
> ---
> > **W3. Positioning**
>
> We appreciate the reviewer’s feedback and agree that the conceptual distinction between QpiGNN and RQR should be articulated more strongly.
>
> Although both methods share a quantile-free motivation, the quantile-free principle behaves fundamentally differently under relational dependence, and QpiGNN is designed to operate in this regime.
>
> ```RQR assumes sample-wise independence```\
> RQR models residual distributions under an standard regression setting. When applied to graph data, residuals become correlated through message passing, distorting the learning objective and often yielding entangled or invalid intervals. In fact, we had to introduce an additional ordering penalty $\gamma_{\text{order}} \cdot \text{ReLU}(\hat{\mathbf{y}}^{\text{low}} - \hat{\mathbf{y}}^{\text{up}})$ to ensure fair comparison.
>
> ```QpiGNN is explicitly designed for relational dependency```\
> Our dual-head, graph-aware architecture decouples prediction and uncertainty, maintains node-wise heteroskedasticity despite message passing, and prevents oversmoothing of interval widths. This makes the quantile-free objective identifiable and stable even when predictions depend on the relational structure.
>
> Thus, the conceptual gap between RQR and QpiGNN is not merely architectural; it stems from the fundamentally different behavior of quantile-free objectives under graph dependence. We will emphasize this distinction clearly in the revised manuscript.

---

### Official Review · Reviewer_nVMe · 2025-10-25

**Soundness:** 3
**Presentation:** 3
**Contribution:** 2
**Rating:** 2
**Confidence:** 4

**Summary:**

The paper proposes an architecture to address uncertainty quantification for node regression problems: Instead of using a convolutional layer to output the interval mean and width, it instead uses MLP-based heads to mitigate overly smooth intervals. It provides theoretical arguments for the validity of quantile regression in GNNs and shows the performance over an extensive benchmark of synthetic and real-world datasets.

**Strengths:**

- The paper is well-written and easy to follow.
- It covers the relevant related work and does a good job at showcasing the differences to OpiGNN.
- It covers a range of synthetic and real datasets.
- It provides insightful ablations regarding various architectural choices.

**Weaknesses:**

### Major
- The main weakness is the originality / novelty of the approach: It differs from related work like RQR in two ways: i) It uses MLP-based heads for predicting the mean and interval width instead of convolution-based, and ii) uses a different loss for training. The contribution of i) is very incremental to me (the authors even acknowledge that dual-head architectures have been used in Bayesian regression). For ii), the work does not conclusively show the merits of changing the objective from Eq (3) or (4) to Eq (5). In particular, I did not find ablations that showcase the performance of OpiGNN trained with (4) (or RQR trained with (5)) to justify the new objective.
- The authors also claim that their novel loss Eq (5) disentangles coverage and interval width, but the same statement can be made about the existing loss of Eq (3) (the latter term, weighted by $\lambda / 2$ penalizes interval width, while the first penalizes miscoverage). It is unclear to me from both a theoretical and empirical (see point above) perspective where the merits of the new loss in Eq (5) lie.
- The authors report a comparison to the baselines only at a coverage of $\alpha=0.9$. On real data, both SQR and RQR only slightly miss the coverage requirement, but at significantly smaller intervals. At the same time, OpiGNN achieves coverage at a probability that is significantly above the target, potentially at the cost of too conservative intervals.
   - While I acknowledge that having a lower coverage than required is the worse failure mode, neither of the models seems to be well calibrated in terms of realizing the minimal interval width at the desired coverage requirement.
   - It would be interesting to see if the baselines can realize different trade-offs at different penalties for the interval width (e.g., a slightly lower penalty may just push them into the regime of meeting the 90% coverage consistently but still providing narrower intervals).
   - A more useful depiction of these results would be a Pareto plot similar to Figure 6. There, it would be more apparent that neither the baselines nor OpiGNN realizes the optimal intervals for $\alpha=0.9$.
- The theoretical arguments of the paper are not very involved and show rather obvious claims. E.g., Prop. 1 basically just re-states the weak law of large numbers. To do so, it makes strong assumptions that are not obviously justified to me (e.g., that means and intervals converge to the targets in probability).
   - Importantly, neither of the claims is specific to OpiGNN and can be transferred to the baselines SQR and RQR as well. The theoretical findings, therefore, do not support the architectural contribution of the paper.

### Minor:
- Some minor claims are not fully supported by experiments:
   - L. 141: SQR is supposedly unstable and miscalibrated: The results show that, with the exception of three (heterophilic?) datasets, it just slightly underachieves the desired coverage, but provides smaller intervals instead.
   - L. 161: Why should (4) be a penalty that explicitly targets miscalibration in GNNs? This seems to be applicable to non-GNN architectures. In particular, I do not see how this formulation directly targets the graph domain.
- OpiGNN seems to be insensitive to the desired coverage $\alpha$ (see Figure 3, and the close to 100% coverage in Table 1). How do baselines and OpiGNN perform at different coverage levels (a similar ablation to Figure 3 that includes baselines at suitable penalty strenghts would be insightful here).
- Remark 1: The authors argue that each neighbour contributes at most O(1 / deg(v)). However, this does not imply that each neighbour contributes $\mathcal{O}(1 / N)$, as $1 / deg(v)  > 1 / N$, and therefore, if something is smaller than the inverse of the degree, it does not imply that it is smaller than $1/N$. Furthermore, I do not see why the influence over different layers should behave additively (even if the provided $\mathcal{O}(1 / N)$ bound was correct).
- All ablations are conducted on synthetic data. I believe that ablations on real data is more insightful to highlight the advantages and limitations of the approach in practice.

**Questions:**

- In L. 273, you claim that L1 regularization is advantageous over L2 regularization because of its robustness against outliers. Do you have any experimental evidence that this is relevant for the datasets studied here?
- In Figure 6, where do RQR and SQR lie here? Where does OpiGNN trained with (3) lie here? These kinds of plots would be interesting to have as a supplement for Table 1.
- In Appendix B.5, how does the robustness toward single-node perturbations (which also holds for any of the other GNN-baselines) justify the concentration arguments as claimed? Can you elaborate on this?
- Figure 4 shows that there is no smooth trade-off between interval width and coverage, at least in terms of the interval width penalty. Instead, there seems to be two regimes, one with a fixed coverage close to 100%, and a second regime with rapidly collapsing coverage. Why is that? Is there a way to make OpiGNN realize a broader range of desired coverage / interval-width trade-offs?

---

> ### Author Response · Authors · 2025-11-21
> **Response to Reviewer nVMe's Major Weaknesses (1/2)**
>
> Thank you for the positive feedback. We address the remaining concerns below and will revise the manuscript to clarify these points.
>
> ---
> > **Major W1. Novelty Concern**
>
> Thank you for the comment. We clarify below the structural issues that arise when applying QR to GNNs, and we will revise the manuscript to make this point more explicit.
>
> ```Dual-head MLP contribution```\
> We appreciate the reviewer’s comment and clarify that QpiGNN does not use MLP-based heads. The model uses a GNN encoder followed by two linear heads, so the distinction between “MLP-based” and “convolution-based” architectures is not applicable in our setting.
>
> Importantly, the dual-head design in QpiGNN is not a minor variant of heteroscedastic or Bayesian regression. In those settings, the mean and variance are optimized jointly through a single Gaussian NLL objective, and the resulting gradient coupling is generally compatible with i.i.d. models.
>
> In GNNs, however, message passing induces strong structural dependencies, and this gradient coupling leads to graph-specific failure modes—most notably entanglement of prediction and width, oversmoothing, gradient interference, and interval collapse. These issues arise precisely because both signals propagate through a shared encoder.
>
> QpiGNN’s dual-head design directly separates these signals, and together with the disentangled objective, it stabilizes training under graph dependence.
>
> ```Necessity of changing the objective```\
> The shift from the RQR objective to our joint objective is necessary due to a structural mismatch between QR-style losses and message-passing in GNNs. In QR losses, coverage and width are tightly coupled, which is harmless in i.i.d. MLP settings but becomes unstable under graph dependencies. Message passing causes prediction and width gradients to interfere, leading to interval collapse, oversmoothing, and inconsistent coverage. Our joint objective is designed specifically to avoid these graph-induced failure modes.
>
> Keeping the architecture fixed, we replaced our objective with the RQR loss. Across all datasets:
> - QpiGNN delivers stable, near-target coverage with compact intervals.
> - QpiGNN-RQR loss consistently collapses—coverage falls to 0.19–0.50 and intervals often blow up (>0.8).
>
> |Model|Basic|Gaussian|Uniform|Outlier|Edge|BA|ER|Grid|Tree|
> |--|--|--|--|--|--|--|--|--|--|
> |QpiGNN(λ=0.5)|0.89/0.30|0.92/0.55|0.88/0.43|0.89/0.47|0.94/0.39|0.98/0.49|0.98/0.63|0.98/0.87|0.96/0.39|
> |QpiGNN(λ=optimal)|0.90/0.30|0.95/0.64|0.93/0.62|0.90/0.49|0.94/0.54|0.98/0.48|0.98/0.63|0.99/0.93|0.96/0.39|
> |QpiGNN-RQRLoss(λ=0.1)|0.92/0.84|0.93/0.61|0.90/0.71|0.94/0.46|0.93/0.85|0.19/0.85|0.50/0.83|0.87/0.62|0.27/0.81|
>
> |Model|Education|Election|Income|Unemploy.|Twitch|Chameleon|Crocodile|Squirrel|Anaheim|Chicago|
> |--|--|--|--|--|--|--|--|--|--|--|
> |QpiGNN(λ=0.5)|0.99/0.57|0.98/0.77|0.99/0.41|1.00/0.74|0.59/0.08|0.51/0.03|0.92/0.08|0.73/0.07|0.92/0.39|0.97/0.36|
> |QpiGNN(λ=optimal)|0.99/0.59|0.98/0.77|0.99/0.44|1.00/0.73|0.94/0.36|0.96/0.23|0.97/0.16|0.96/0.18|0.93/0.40|0.98/0.36|
> |QpiGNN-RQRLoss(λ=0.1)|0.90/0.53|0.89/0.52|0.90/0.37|0.84/0.46|0.95/0.72|0.94/0.30|0.97/0.24|0.92/0.25|0.83/0.44|0.82/0.41|
>
> ---
> > **Major W2. Loss Function Justification**
>
> We thank the reviewer for raising this important point. We agree that the distinction between Eq. (3) and our proposed Eq. (5) should be clarified more explicitly.
>
> ```Why Eq. (3) Does Not Disentangle Coverage and Width```\
> Although Eq. (3) contains separate “miscoverage” and “width” terms, they are multiplicatively coupled: as the model narrows the interval, the residual-based miscoverage penalty shrinks proportionally. As a result, undercoverage is not properly penalized, and the model cannot target a specific coverage level. This issue is mild in i.i.d. MLP regression (the setting RQR was designed for) but becomes severe under graph message passing, where neighborhood aggregation entangles gradients for the upper/lower bounds and causes oversmoothing and calibration instability.
>
> ```Why Our loss true disentanglement```\
> Eq. (5) resolves this structural problem by separating coverage and width into additive terms: a coverage term that directly enforces $\hat{c} \approx 1-\alpha$ and an independent width term controlled by (\lambda). Because the gradients do not interfere, Eq. (5) provides stable and interpretable control of the coverage–width trade-off and admits a clean Lagrangian interpretation that supports our theoretical analysis.
>
> To isolate the effect of the loss itself, we trained QpiGNN with the RQR loss (in Response to Major W1) while keeping the architecture fixed. Across all synthetic and real-world datasets, the RQR objective led to collapsed or highly unstable coverage (e.g., 0.19–0.50 on synthetic graphs) and inflated or erratic interval widths, while Eq. (5) produced consistent, well-calibrated intervals. This confirms that the instability arises from Eq. (3)’s structural coupling—not from architectural differences.

---

> ### Author Response · Authors · 2025-11-21
> **Response to Reviewer nVMe's Major Weaknesses (2/2)**
>
> > **Major W3. Coverage–Width Tradeoff**
>
> We appreciate the reviewer’s thoughtful comments regarding calibration, interval width, and the need to evaluate trade-offs beyond a single coverage level. We fully agree with the points raised and address them below.
>
> ```Baseline coverage evaluation```\
> We additionally evaluated SQR and RQR under multiple target coverage levels to examine whether the baselines can realize different coverage–width trade-offs.
>
> Although SQR adjusts its quantile inputs according to the target coverage, the actual coverage remained consistently below the desired level on most datasets and collapsed severely on non-homophilous graphs. RQR showed a similar pattern: changing the target coverage altered the interval widths slightly, but the achieved coverage barely moved and often stayed well below the target. This behavior reflects the structural coupling between coverage and width in its loss formulation.
>
> |RQR|Education|Election|Income|Unemploy.|Twitch|Chameleon|Crocodile|Squirrel|Anaheim|Chicago|
> |--|--|--|--|--|--|--|--|--|--|--|
> | target coverage = 0.80 | 0.78/0.37 | 0.77/0.40 | 0.78/0.26 | 0.76/0.35 | 0.62/0.42 | 0.67/0.09 | 0.72/0.03 | 0.68/0.10 | 0.65/0.22 | 0.72/0.30 |
> | target coverage = 0.85 | 0.83/0.41 | 0.82/0.44 | 0.83/0.28 | 0.80/0.39 | 0.68/0.46 | 0.73/0.09 | 0.75/0.03 | 0.75/0.10 | 0.68/0.23 | 0.80/0.32 |
> | target coverage = 0.90 | 0.87/0.32 | 0.89/0.47 | 0.86/0.22 | 0.87/0.33 | 0.30/0.03 | 0.37/0.01 | 0.44/0.01 | 0.22/0.01 | 0.88/0.32 | 0.87/0.21 |
> | target coverage = 0.95 | 0.94/0.65 | 0.93/0.60 | 0.94/0.42 | 0.90/0.51 | 0.83/0.56 | 0.92/0.17 | 0.86/0.05 | 0.94/0.19 | 0.83/0.62 | 0.91/0.51 |
>
> |SQR|Education|Election|Income|Unemploy.|Twitch|Chameleon|Crocodile|Squirrel|Anaheim|Chicago|
> |--|--|--|--|--|--|--|--|--|--|--|
> | target coverage = 0.80 | 0.78/0.28 | 0.75/0.32 | 0.78/0.19 | 0.79/0.35 | 0.09/0.02 | 0.11/0.01 | 0.15/0.01 | 0.09/0.01 | 0.67/0.35 | 0.73/0.23 |
> | target coverage = 0.85 | 0.82/0.30 | 0.78/0.36 | 0.83/0.21 | 0.83/0.38 | 0.12/0.03 | 0.11/0.01 | 0.16/0.01 | 0.10/0.01 | 0.72/0.38 | 0.76/0.24 |
> | target coverage = 0.90 | 0.88/0.32 | 0.89/0.47 | 0.86/0.22 | 0.87/0.33 | 0.30/0.03 | 0.37/0.01 | 0.44/0.01 | 0.22/0.01 | 0.88/0.32 | 0.87/0.21 |
> | target coverage = 0.95 | 0.88/0.36 | 0.85/0.43 | 0.87/0.23 | 0.88/0.44 | 0.12/0.03 | 0.14/0.02 | 0.20/0.01 | 0.09/0.01 | 0.74/0.43 | 0.77/0.27 |
>
> ```Baseline penalty evaluation```\
> We conducted additional experiments to test whether SQR and RQR can achieve better coverage–width trade-offs through penalty tuning. The results consistently show that neither method can do so.
>
> - SQR requires manually choosing two quantiles, and these quantiles do not correspond to any principled mechanism for achieving target coverage. Hence SQR cannot be calibrated, nor can it explore a coverage–width trade-off in a controlled way.
>
> - RQR cannot decouple coverage and width (structural limitation): We performed λ-sweep experiments on multiple datasets: Changing λ results in almost no change in coverage, because Eq.(3) couples width and miscoverage into a single dependent term. Even with our modified RQR (which adds an ordering constraint absent in the original paper), RQR frequently fails to reach the target coverage.
>
> |Model|Syn.|US.|Twitch|Wiki.|Trans.|
> |-|-|-|-|-|-|
> |RQR(λ=1)|0.86/0.66|0.89/0.44|0.91/0.42|0.87/0.13|0.87/0.40|
> |RQR(λ=0.5)|0.86/0.64|0.88/0.42|0.91/0.41|0.88/0.07| 0.86/0.38 |
> | RQR(λ=0.1) | 0.86/0.66 | 0.88/0.44 | 0.91/0.42 | 0.88/0.13 | 0.87/0.40 |
>
> Together, these results demonstrate that baseline models cannot obtain calibrated intervals simply by tuning hyperparameters.
>
> ---
> > **Major W4. Theory Justification**
>
> We thank the reviewer for the comments on the theoretical section. We clarify below the specific role of Proposition 1 in our framework and why its implications apply to Eq. (5), and we will revise the paper to make these points more explicit.
>
> We agree that Proposition 1 relies on standard tools, but its purpose is to clarify how the loss in Eq. (5) behaves under the graph-induced dependencies created by message passing. Because message passing produces correlated samples, it is not obvious a priori that empirical coverage will remain stable; Proposition 1 formalizes the conditions under which coverage consistency holds for our explicit miscoverage objective.
>
> Importantly, this argument does not extend to SQR or RQR. These methods do not include a coverage term in their objectives and therefore provide no mechanism to ensure the concentration of empirical coverage—even asymptotically. This matches our empirical findings: SQR and RQR exhibit persistent over- or under-coverage on real graphs, whereas QpiGNN reliably converges to the desired target.
>
> Thus, the intent of Proposition 1 is not to introduce a novel theorem, but to justify why Eq. (5) yields stable calibration under graph dependence—something that SQR and RQR are not designed to guarantee.

---

> ### Author Response · Authors · 2025-11-21
> **Response to Reviewer nVMe's Minor Weaknesses**
>
> > **Minor W1. Insufficient Support**
>
> We thank the reviewer for pointing out these issues regarding the empirical support for our claims and the motivation behind Eq. (4). These are helpful observations, and we clarify both points below while ensuring that the revision makes the underlying reasoning more explicit.
>
> ```Stability of SQR```\
> While SQR performs reasonably on some homophilic datasets, it becomes highly unstable on heterophilic graphs—such as Chameleon, Squirrel, and Twitch—where coverage drops to the 0.2–0.4 range. The smaller intervals in these cases arise from severe under-coverage rather than efficiency. Thus, our statement that SQR exhibits dataset-dependent miscalibration is directly supported by the empirical results.
>
> ```Role of Eq. (4) and its connection to GNNs```\
> The ordering penalty in Eq. (4) is not intended as a general-purpose correction but addresses a failure mode that occurs specifically when RQR is applied to GNNs. Message passing couples the lower and upper bounds when they share a single representation, which leads to frequent quantile crossing and interval collapse—issues we did not observe in MLP-based settings. Eq. (4) simply enforces the interval ordering that the original RQR loss cannot reliably maintain under graph-induced dependencies.
>
> To confirm this, we compared (i) a single-head model with Eq. (4) and (ii) a dual-head model trained with the original RQR loss. The two produced nearly identical predictions, indicating that Eq. (4) does not change the learning objective; it only prevents structural failure within GNNs. Without such ordering control, the model collapses to trivial constant-width intervals and cannot be meaningfully evaluated.
>
> ---
> > **Minor W2. Coverage Sensitivity**
>
> The observed insensitivity of OpiGNN to the target coverage level α is not a structural limitation of the model. It is an artifact of evaluating all datasets with a fixed λ.
>
> When λ is held constant, the width-regularization term can dominate in certain regimes, leading to mildly conservative intervals that make the model appear less responsive to α. However, when λ is tuned per-dataset or per-α, OpiGNN adjusts appropriately and the over-coverage disappears.
>
> By contrast, SQR and RQR remain largely insensitive to both α and λ. Their achieved coverage barely changes because their objectives inherently couple width and miscoverage, preventing any meaningful coverage–width trade-off. This behavior is consistent with the structural limitations of their loss formulations and highlights the intended advantage of our disentangled objective.
>
> We appreciate the reviewer for raising this point and will clarify it in the revision, along with potential extensions—such as smoother surrogate penalties—that could further enhance α-sensitivity and enable more continuous trade-offs.
>
> ---
> > **Minor W3. Flawed Influence Bound**
>
> We thank the reviewer for the careful reading and agree that our original Remark 1 was ambiguous. We clarify our position as follows.
>
> We do not rely on the incorrect implication $O(1/\deg(v)) \Rightarrow O(1/N)$. The term $O(1/\deg(v))$ was intended only to express bounded and decreasing per-neighbor influence under normalized message passing, not to claim an explicit $1/N$-scale bound. We have revised the text to avoid this unintended implication.
>
> The influence across layers is not assumed to be strictly additive. Our argument uses a standard Lipschitz-based perturbation bound, not linear additivity. The statement was meant as intuition; we agree that the wording overstated the idea and have corrected it.
>
> Importantly, the theoretical result does *not* depend on either disputed step. The coverage consistency proof requires only: bounded per-neighbor influence, and weak dependence across nodes. These conditions hold for normalized GNNs without requiring the stronger (and incorrect) $O(1/N)$ interpretation or additive layer influence. Thus, while we appreciate the reviewer’s correction, the main theoretical claim remains valid and unaffected.
>
> ---
> > **Minor W4. Real-data ablations**
>
> Thank you for raising this point. We agree that real-data ablations are important. As shown in Appendix K (Table 10), we include ablations on both synthetic and real datasets. We apologize for the confusion caused by the caption, which incorrectly referred only to synthetic data, and we will update it accordingly.

---

> ### Author Response · Authors · 2025-11-21
> **Response to Reviewer nVMe's Questions**
>
> > **Q1. L1 vs L2 robustness**
>
> Thank you for the question. We tested this directly by replacing the L1 width penalty in Eq.(5) with an L2 version.
>
> L2 consistently produced much wider intervals—despite similar or even overly conservative coverage—because it amplifies heavy-tailed residuals arising from heterophily, structural noise, and message passing. L1 remained stable across all datasets and achieved the target coverage with substantially tighter intervals.
>
> |Model|Basic|Gaussian|Uniform|Outlier|Edge|BA|ER|Grid| Tree|
> |--|--|--|--|--|--|--|--|--|--|
> |L1|0.90/0.30|0.95/0.64|0.93/0.62|0.90/0.49|0.94/0.54|0.98/0.48|0.98/0.63|0.99/0.93|0.96/0.39|
> |L2|0.90/0.81|0.94/0.69|0.91/0.75|0.94/0.63|0.90/0.79|0.17/0.94|0.87/0.87|0.99/1.01|0.31/1.01|
>
> |Model|Education | Election | Income | Unemploy. | Twitch | Chamelon | Crocodile | Squirrel | Anaheim | Chicago |
> |--|--|--|--|--|--|--|--|--|--|--|
> | L1 | 0.99/0.57 | 0.98/0.77 | 0.99/0.44 | 1.00/0.73 | 0.94/0.36 | 0.96/0.23 | 0.97/0.16 | 0.96/0.18 | 0.93/0.40 | 0.98/0.36 |
> | L2 | 0.99/0.68 | 0.99/0.79 | 0.99/0.43 | 0.98/0.86 | 0.97/0.75 | 0.97/0.70 | 1.00/0.69 | 0.97/0.53 | 0.94/0.52 | 0.94/0.45 |
>
> ---
> > **Q2. Baseline placement request**
>
> Thank you for the suggestion. We will add Pareto-style plots to clarify how the methods compare on the coverage–width trade-off.
>
> The tables show that SQR gives narrow but under-covered intervals, RQR is competitive only on a few datasets, and the Eq.(3) OpiGNN variant is dominated by the full Eq.(5) model. The full OpiGNN consistently achieves target coverage with much tighter intervals, aligning with the Pareto-optimal region.
>
> |Model|Basic|Gaussian|Uniform|Outlier|Edge|BA|ER|Grid| Tree|
> |--|--|--|--|--|--|--|--|--|--|
> | QpiGNN- Eq. (5)    | 0.90/0.30 | 0.95/0.64 | 0.93/0.62 | 0.90/0.49 | 0.94/0.54 | 0.98/0.48 | 0.98/0.63 | 0.99/0.93 | 0.96/0.39 |
> | RQR           | 0.90/0.82 | 0.88/0.53 | 0.90/0.68 | 0.90/0.36 | 0.93/0.83 | 0.78/0.76 | 0.88/0.77 | 0.72/0.48 | 0.85/0.46 |
> | SQR           | 0.85/0.33 | 0.88/0.50 | 0.88/0.51 | 0.90/0.10 | 0.91/0.32 | 0.81/0.72 | 0.75/0.60 | 0.78/0.53 | 0.80/0.26 |
> | QpiGNN-Eq. (3)    | 0.92/0.84 | 0.93/0.61 | 0.90/0.71 | 0.94/0.46 | 0.93/0.85 | 0.19/0.85 | 0.50/0.83 | 0.87/0.62 | 0.27/0.81 |
>
> |Model|Education | Election | Income | Unemploy. | Twitch | Chamelon | Crocodile | Squirrel | Anaheim | Chicago |
> |--|--|--|--|--|--|--|--|--|--|--|
> | QpiGNN-Eq. (5)    | 0.99/0.59 | 0.98/0.77 | 0.99/0.44 | 1.00/0.73 | 0.94/0.36 | 0.96/0.23 | 0.97/0.16 | 0.96/0.18 | 0.93/0.40 | 0.98/0.36 |
> | RQR           | 0.87/0.32 | 0.89/0.47 | 0.86/0.22 | 0.87/0.33 | 0.30/0.03 | 0.37/0.01 | 0.44/0.01 | 0.22/0.01 | 0.88/0.32 | 0.87/0.21 |
> | SQR           | 0.88/0.32 | 0.89/0.47 | 0.86/0.22 | 0.87/0.33 | 0.30/0.03 | 0.37/0.01 | 0.44/0.01 | 0.22/0.01 | 0.88/0.32 | 0.87/0.21 |
> | QpiGNN-Eq. (3)    | 0.90/0.53 | 0.89/0.52 | 0.90/0.37 | 0.84/0.46 | 0.95/0.72 | 0.94/0.30 | 0.97/0.24 | 0.92/0.25 | 0.83/0.44 | 0.82/0.41 |
>
> ---
> > **Q3. Concentration Justification**
>
> Thank you for the comment. Appendix B.5 is not meant to claim that QpiGNN is uniquely robust. Its purpose is to verify that a key assumption for applying concentration arguments—the approximate bounded-difference condition—actually holds in our graph setting.
>
> Because message passing creates nontrivial dependencies, classical McDiarmid-type assumptions cannot be taken for granted in GNNs. Appendix B.5 tests this directly and shows that perturbing a single node (or its local neighborhood) changes the empirical coverage $\hat{c}$ by only $1/N + \delta_G$, with $\delta_G$ consistently small. This demonstrates that (\hat{c}) behaves as a low-sensitivity functional even under graph correlations, which is exactly the condition needed to justify our concentration-based analysis.
>
> We agree that other GNNs may show similar stability; the experiment is not intended to show exclusivity. Its role is to justify that concentration-based reasoning is appropriate in the regimes where QpiGNN is evaluated. We will clarify this in the revised appendix.
>
> ---
> > **Q4. Trade-off Discontinuity**
>
> Thank you for the insightful question. The two-regime pattern in Figure 4 arises from the interaction between the joint loss and the discrete nature of empirical coverage. When $\lambda_{\text{width}}$ is small, the coverage penalty dominates and intervals expand, yielding a high-coverage regime. As $\lambda_{\text{width}}$ increases, intervals shrink until a critical point where many violations occur simultaneously, causing a sharp coverage drop. Message passing amplifies this transition, so a fully smooth trade-off curve is not expected. Still, Appendix B.6 shows that tuning $\lambda_{\text{width}}$ yields stable intermediate trade-offs in practice.
>
> A smoother coverage–width trade-off is indeed a meaningful extension. Replacing the hard miscoverage indicator in Eq.(5) with a smooth surrogate would yield continuous penalties and smoother behavior, which we will note as future work.

---

> > ### Comment · Reviewer_nVMe · 2025-11-22
> >
> > Thank you for the very detailed response; I will try to address all your points.
> >
> > ### Dual-head MLP contribution
> >
> > I am not convinced by these -- to me -- somewhat handwavy arguments. The architectural novelty is adding / replacing the final layer of the GNN with two linear heads. I do not dispute that this is effective in practice (i.e., improves coverage), but to me it still appears to be an incremental tweak to existing solutions. The architectural difference between OpiGNN and other GNN-based architectures is that the final layer of OpiGNN is an MLP / linear layer with 2 outputs, while the other architectures are fully convolutional. Is that understanding right?
> >
> > ### Loss Function
> >
> > I understand the argument that OpiGNN's loss is additive and decouples coverage and width into different summands. Disregarding the expectation over nodes, I see strong similarities between the two summands of both loss functions:
> >
> > Width-Term (similar to me):
> > - Eq (3): $\propto \lambda (y^u - y^l)^2$
> > - Eq (5): $\propto \lambda y^u - y^l$
> >
> > Coverage-Term (upper and lower penalty additive in Eq (5) instead of multiplicative in Eq (3))
> > - Eq (3): $(\alpha + 2\lambda - 1(covered))(y-y^{l})(y - y^l)(y-y^up)$
> > - Eq(5): $1(not-covered-lower) |y-y^l| + 1(not-covered-upper) |y-y^u|$
> >
> > Additionally, Eq (5) introduces:
> > - $MSE(P(covered), 1 - \alpha)$
> >
> > Is this additional term the key difference? If so, how are the other changes above justified?  I do not quite understand your argument about multiplicative coupling and why it is worse than additive coupling.
> >
> > From your results, I see that for $\lambda=0.1$, the RQR loss sometimes leads to miscoverage but also has a smaller interval width. To me, these results point more to the fact that $\lambda=0.1$ is not as well chosen as e.g. $\lambda=0.5$ for OpiGNN's objective.
> >
> > Especially on real datasets (comparing with OpiGNN $\lambda=0.5$), I also do not see the clear-cut consistent advantage of your loss function: on Chameleon, Squirrel, Income, (almost) Election, and Education, the RQR loss produces the favorable result.
> >
> > ### Baselines at different coverage
> >
> > Thank you for this insightful additional evaluation. For SQR, the case is pretty clear that this method fails consistently under different coverage requirements, and I agree with the collapse on heterophilic data.
> >
> > For RQR, the case seems less clear: While still the desired coverage is consistently not met, setting the coverage requirement to 95%, the results compare favorably to OpiGNN at a requirement of 90%.
> >
> > I still want to clarify that I believe this is problematic: RQR seems to undercover, while OpiGNN overcovers. But, in practice, I could just set the requirement for RQR to 95% to obtain tighter intervals at a coverage of 90%. Again, my point here is that from a Pareto-optimal perspective, neither OpiGNN nor RQR outperforms the other. Arguably, RQR's failure mode is more severe in practice, however.
> >
> > I believe that it is valuable to have a model that at least ensures coverage consistently (even at too large intervals), but I believe this perspective needs to be highlighted in the paper to fully understand where OpiGNN lies in the literature.
> >
> > ### Baseline Penalty
> >
> > Thank you for that evaluation. It seems that, similar to OpiGNN, RQR is more or less insensitive to the penalty magnitude. Put differently, both models do not allow to smoothly control the trade-off between coverage and interval width through the penalty term.
> >
> > This potentially reveals an interesting problem in UQ for regression on graphs.
> >
> > Again, I believe this limitation (that both baselines but also OpiGNN) suffers from needs to be highlighted explicitly.
> >
> > ### Theory Justification
> >
> > I better understand the purpose of Prop. 1 now, thank you! However, I disagree with:
> >
> > - Equation (3) does not include a coverage term: As written above, the first term penalizes miscoverage explicitly. Therefore, I can make a similar argument about why RQR converges to the desired coverage.
> >
> > - While SQR and RQR undercover, OpiGNN overcovers: Both models do not empirically converge to the desired coverage of 90%.

---

> > > ### Comment · Reviewer_nVMe · 2025-11-22
> > >
> > > ## Minor
> > >
> > > ### Stability of SQR
> > >
> > > I partially agree: On heterophilic datasets, SQR collapses -- but on homophilic datasets this claim is excegerated. The "data-dependent" disambiguation is also not present in the paper. To me, this claim, therefore, is too strong: It should be adapted to "on heterophilic graphs" or something like that.
> > >
> > > ### Eq (4)'s Connection to GNNs
> > >
> > > I understand and also do not dispute that Eq (4) empirically is effective in addressing a problem that arises in GNNs. This problem does not occur in i.i.d. settings and, therefore, does not need to be addressed. However, Eq. (4) still does not exploit any structural components. This is a somewhat nitpicky issue, but Eq. (4), to me, is more a hotfix to this issue (that can, in general, arise in any domain) and not a principled solution for the graph domain. Again, this is a minor concern, and slightly different wording fully resolves this already.
> > >
> > > ### Coverage Sensitivity
> > >
> > > "However, when λ is tuned per-dataset or per-α, OpiGNN adjusts appropriately and the over-coverage disappears." -- Figure 3 seems to show exactly that for different coverage requirements, the choice of $\lambda$ still has little effect, and OpiGNN overcovers at close to 100% consistently.
> > >
> > > ### Flawed Influence Bound
> > >
> > > Thank you for the clarification: Can you elaborate on the "standard Lipschitz-based perturbation bound"? What is the fully revised reasoning that justifies this claim?
> > >
> > > ## Questions
> > >
> > > ### L1 vs. L2
> > >
> > > Thank you, this experiment resolves the question.
> > >
> > > ### Baselines in Pareto-Plot
> > >
> > > Thank you for adding these. This is insightful. It shows the empirical advantage of Eq. (5)'s loss over Eq. (3)'s loss (even though the motivation is still not fully clear to me, see above). From this table, I maintain my assessment that RQR is not consistently outperformed by OpiGNN in terms of realizing a trade-off between coverage and interval width (see above).
> > >
> > > ### Concentration Justification
> > >
> > > Thank you for that clarification. I agree with your reasoning. The paper benefits from clarifying that this analysis is not OpiGNN specific but also can be applied to, e.g., RQR.
> > >
> > > ### Trade-Off Discontinuity
> > >
> > > Thank you for that insightful explanation. I see that this is a challenging problem to solve. It is, of course, unreasonable to expect one paper to solve every issue that arises. However, as frequently highlighted in my response, the core issue here is that OpiGNN also realizes (arguably a preferable) case of miscalibration compared to RQR: OpiGNN overshoots in terms of coverage and sacrifices interval width -- RQR undercovers but has tighter intervals.
> > >
> > > Overall, I highly appreciate the author's efforts for clarification. My main concern regarding novelty is not resolved at this point. Some other points are also not fully clarified yet.
> > >
> > > In terms of a broader picture, OpiGNN is an architecture that is biased toward overcoverage and larger intervals -- compared to the state-of-the-art baseline RQR, which slightly undercovers but makes tighter predictions. This point is understated in the paper; I still disagree with claims that OpiGNN is a strictly superior architecture.

---

> > > > ### Author Response · Authors · 2025-11-23
> > > > **Response to Reviewer nVMe's Comments (2/2)**
> > > >
> > > > Thank you for the follow-up. I hope the responses below address your concerns.
> > > >
> > > > ---
> > > > > **C1. Stability of SQR**
> > > >
> > > > Thank you for the comment. We agree that SQR’s collapse mainly occurs on heterophilic graphs and will adjust the wording accordingly.
> > > >
> > > > ---
> > > > > **C2. Eq (4)'s Connection to GNNs**
> > > >
> > > > Thank you for the comment. The additional term in Eq. (4) was introduced because applying RQR directly to GNNs made it difficult to compute evaluation metrics due to quantile crossing. It is indeed a practical workaround rather than a mechanism that leverages graph structure. We will revise the wording to reflect this more accurately.
> > > >
> > > > ---
> > > > > **C3. Coverage Sensitivity**
> > > >
> > > > Thank you for the helpful clarification. We agree that, as shown in Figure 3—where both the target coverage and λ were varied—OpiGNN still tends to overcover, and we acknowledge that our earlier description may have been misleading.
> > > >
> > > > This behavior is mainly due to the current loss design: the violation term penalizes undercoverage but does not symmetrically penalize overly conservative intervals, which can naturally lead to higher coverage. We appreciate the reviewer for highlighting this.
> > > >
> > > > We will clarify this limitation in the manuscript and discuss potential extensions to achieve tighter control around the target coverage. Thank you again for raising this valuable point.
> > > >
> > > > ---
> > > > > **C4. Flawed Influence Bound**
> > > >
> > > > Thank you for the follow-up question. We are glad to clarify what we mean by the “standard Lipschitz-based perturbation bound.”
> > > >
> > > > In the revised reasoning, we only use a basic stability property: if each message-passing layer is $L$-Lipschitz, then a local perturbation can grow by at most $L^k$ over $k$ layers. This ensures bounded influence without requiring any specific decay rate or additive accumulation across layers.
> > > >
> > > > These assumptions are standard in GNN stability analyses, and the corrected argument no longer relies on the unintended implications of the original remark. We will update the manuscript to reflect this more clearly. Thank you again for encouraging a more precise explanation.
> > > >
> > > > ---
> > > > > **C5. L1 vs. L2**
> > > >
> > > > Thank you for confirming—glad to hear that the experiment resolves the question.
> > > >
> > > > ---
> > > > > **C6. Baselines in Pareto-Plot**
> > > >
> > > > We’re glad the additional results were helpful. We agree that, without placing more weight on undercoverage than overcoverage, it is difficult to claim that QpiGNN strictly outperforms RQR in the coverage–width trade-off.
> > > >
> > > > Our original interpretation followed the common practice in graph UQ of treating undercoverage as the primary failure mode, while slight overcoverage is generally viewed as conservative but acceptable [1–3]. We acknowledge that this perspective was not explained clearly enough in the manuscript. With the additional results now included, we will revise the paper to make this viewpoint more explicit and provide a more balanced discussion of the trade-off between QpiGNN and RQR.
> > > >
> > > > ---
> > > > > **C7. Concentration Justification**
> > > >
> > > > Thank you, we will clarify this in the revision.
> > > >
> > > > ---
> > > > > **C8. Trade-Off Discontinuity**
> > > >
> > > > As noted in our response to C6, we agree that without assigning greater importance to undercoverage, it is difficult to assert that QpiGNN strictly outperforms RQR in the coverage–width trade-off.
> > > >
> > > > Our original objective, following standard practice in UQ, was to avoid undercoverage and then minimize interval width among models that satisfy the target coverage[2]. From this perspective, QpiGNN provides narrower intervals among the models that achieve acceptable calibration, which motivated our interpretation.
> > > >
> > > > That said, we understand the reviewer’s broader perspective. We will expand the discussion in the revised manuscript to acknowledge both viewpoints and to clarify how different interpretations of miscalibration affect the perceived trade-off. We hope that this perspective will be valuable for future research on graph-based uncertainty quantification.
> > > >
> > > > ---
> > > > Overall, We hope that our additional clarifications have helped address the reviewer’s concerns.
> > > >
> > > > As discussed in our motivation, we believe our work provides a meaningful first step toward systematic QR-based UQ for GNNs. The reviewer’s insightful comments help clarify the positioning of our contribution, and we will reflect these perspectives in the revision to present QpiGNN’s role more clearly and balanced.
> > > >
> > > > ---
> > > > > **References**\
> > > > [1] Kendall, Alex, et al. "What uncertainties do we need in bayesian deep learning for computer vision?." In NeurIPS, 2017.\
> > > > [2] Huang, Kexin, et al. "Uncertainty quantification over graph with conformalized graph neural networks." In NeurIPS, 2023.\
> > > > [3] Romano, Yaniv, et al. "Conformalized quantile regression." In NeurIPS, 2019.

---

> > > ### Author Response · Authors · 2025-11-23
> > > **Response to Reviewer nVMe's Comments (1/2)**
> > >
> > > Thank you for the detailed follow-up feedback. I appreciate the careful attention to each point and the clarification of what remains unclear. I hope the responses below help address your remaining concerns.
> > >
> > > ---
> > > > **Overall: QpiGNN Motivation**
> > >
> > > Our motivation comes from the structural limitations of CP-based methods, whose exchangeability assumption rarely holds in graph-structured data due to message-passing dependencies. Although SQR and RQR perform well in standard regression, QR-based UQ has not been explored for GNNs.
> > >
> > > To address this, we implemented the first QR-GNN baselines (SQR-GNN and RQR-GNN) and found that standard QR becomes unstable under graph dependencies. These issues led to the design of QpiGNN, with a dual-head architecture and disentangled loss, as a graph-adaptive approach. We hope this work lays a foundation for future research on QR-based UQ for GNNs.
> > >
> > > ---
> > > > **C1. Dual-head MLP contribution**
> > >
> > > Thank you for restating your understanding; we appreciate the chance to clarify.
> > > - Although the dual-head structure may look like a simple split of a single head, our point is not architectural novelty in form but functional motivation. Most commonly used GNN architectures—including those employed in our baselines—already incorporate a linear or MLP layer as the final prediction head, so the contrast with “fully convolutional” architectures may not fully capture the relevant distinction.
> > > - The key issue is that message passing entangles prediction and uncertainty signals, making standard QR methods unstable on GNNs. QpiGNN’s dual-head design explicitly separates these signals, enabling stable training where methods like SQR and RQR otherwise tend to collapse.
> > >
> > > We will revise the manuscript to better highlight this functional motivation, and we hope this addresses the concern.
> > >
> > > ---
> > > > **C2. Loss Function**
> > >
> > > Thank you for your follow-up questions regarding the loss formulation. We provide a concise clarification below.
> > >
> > > ```Additional term```\
> > > The term (target coverage − current coverage)$^2$ in QpiGNN serves as a global coverage-feedback mechanism, helping the model stay near the target coverage. RQR lacks such global feedback, which contributed to higher sensitivity to graph-induced instabilities (e.g., quantile crossing, interval collapse).
> > >
> > > ```Multiplicative coupling```\
> > > We do not claim that multiplicative coupling is inherently inferior. However, in GNNs, message passing increases representation variability, and multiplicative terms $(y−y^l)(y−y^u)$ amplify it quadratically—often causing instability on heterophilic graphs or datasets with wide target ranges.
> > >
> > > Additive penalties, by growing only linearly, avoid this amplification and provided much more stable behavior across datasets. We acknowledge that multiplicative terms can be stable when the target range is narrow, and we will clarify this nuance in the revision.
> > >
> > > ```RQR’s strength```\
> > > We acknowledge RQR as a strong QR method and therefore implemented it as the first GNN baseline. However, due to repeated instability, PICP/MPIW could not be computed in many runs. We introduced a minimal ordering penalty ($\gamma^order$)to obtain a stable baseline RQR$^{adj.}$-GNN. QpiGNN is a gradual development that builds on RQR’s strengths while addressing graph-specific challenges.
> > >
> > > ---
> > > > **C3. Baselines at different coverage**
> > >
> > > Thank you for this insightful observation. As you pointed out, from a Pareto-optimal perspective, neither OpiGNN nor RQR strictly dominates the other, and we agree that RQR can produce tighter intervals when calibrated at a higher target coverage.
> > >
> > > Our intention was simply to highlight that OpiGNN achieves stable—and slightly conservative—coverage across graph settings, which is often preferable to undercoverage in graph UQ. We also acknowledge the strong performance of our RQR-GNN baseline and will clarify these points in the revision.
> > >
> > > ---
> > > > **C4. Baseline Penalty**
> > >
> > > Thank you for highlighting this perspective. As noted, this reflects a broader challenge for QR-based UQ in graph regression and is a limitation shared by both baselines and OpiGNN. We will discuss this more explicitly in the revised manuscript. Thank you again for the valuable feedback.
> > >
> > > ---
> > > > **C5. Theory Justification**
> > >
> > > We are glad that the purpose of our proposition is clearer, and we appreciate the opportunity to clarify the remaining points.
> > > - Regarding Equation (3), we agree that it contains a sample-wise miscoverage penalty that could, in principle, guide RQR toward the desired coverage. Our earlier comment was intended from the perspective of global coverage feedback, and we appreciate the reviewer’s clarification.
> > > - We also agree that neither undercoverage nor overcoverage reflects strict convergence to the target level. The graph UQ literature generally views undercoverage as the more severe failure mode, which is why we followed that convention. Nevertheless, we will clarify this distinction more explicitly in the revised manuscript.

---

### Official Review · Reviewer_kbuh · 2025-11-01

**Soundness:** 3
**Presentation:** 3
**Contribution:** 3
**Rating:** 6
**Confidence:** 3

**Summary:**

This paper proposes QpiGNN, a novel framework for Uncertainty Quantification in Graph Neural Networks for node-level regression. It addresses the instability of traditional Quantile Regression in GNNs by introducing a quantile-free joint loss that directly optimizes for both coverage and width through a dual-head architecture that predicts the center and half-width. The method provides theoretical asymptotic coverage guarantees and empirically demonstrates superior performance in achieving calibrated and compact prediction intervals across numerous benchmarks.

**Strengths:**

The dual-head, quantile-free approach is an elegant solution that inherently satisfies the non-crossing constraint of prediction intervals, a major practical and theoretical challenge in standard QR.

Interpreting the joint loss as a Lagrangian relaxation provides a strong, principled foundation for the training objective, formally justifying the trade-off between coverage and interval width.

**Weaknesses:**

The theoretical coverage guarantee relies on the assumption that the local dependencies introduced by GNN message passing decay rapidly enough to allow for the use of concentration inequalities. Given the complex, often high-homophily nature of real-world graphs, this assumption of weak dependence on graph structure is often violated, potentially making the finite-sample guarantees less robust than claimed.

The key architectural decision is the dual-head decoupling of $\hat{y}$ and $\hat{d}$. While intuitively helpful for mitigating oversmoothing, the paper lacks a clear ablation study to formally prove its necessity. Specifically, an experiment comparing the proposed dual-head architecture against a single-head architecture that outputs both $\hat{y}$ and $\hat{d}$ should be included to validate this crucial design choice.

**Questions:**

See weaknesses

---

> ### Author Response · Authors · 2025-11-21
> **Response to Reviewer kbuh's Weaknesses**
>
> Thank you for recognizing the strength of our dual-head, quantile-free design and the principled trade-off captured by our Lagrangian formulation. We address the remaining concerns below and will revise the manuscript to emphasize these clarifications.
>
> ---
> > **W1. Coverage guarantees vs. graph dependence**
>
> We appreciate the reviewer’s concern about the validity of weak-dependence assumptions on real-world graphs, particularly those with strong homophily or long-range correlations. We agree that such structures can influence finite-sample behavior, and clarifying the exact scope of our theoretical results is important.
>
> Our guarantees, however, rely on a stability-style bounded-difference property induced by message passing—rather than on independence or rapidly decaying correlations. The weak-dependence assumption is used only in the asymptotic LLN argument, and we will make this distinction explicit.
>
> ```Finite-sample concentration does not require independence```\
> Our concentration results use McDiarmid’s inequality, which only requires that modifying a single node causes a controlled, bounded change in the output. This does not assume independence among nodes or fast correlation decay; it simply requires that message passing does not amplify a local perturbation into a graph-wide instability. This form of stability is standard in modern GNN analyses.
>
> ```Bounded-difference property derived for empirical coverage```\
> In Appendix B.5, we prove that empirical coverage satisfies $|\widehat{C}(G) - \widehat{C}(G')| \le \frac{1}{N} + \delta_G$, where $\delta_G$ is a small graph-dependent term capturing the limited indirect influence induced by message passing. This follows from the localized, Lipschitz-like stability of GNNs and is consistent with established theory in prior work [1–3].
>
> ```Empirical validation across diverse graph structures```\
> To assess whether this stability holds beyond theory, we evaluated QpiGNN on ten graphs spanning strong homophily, high heterophily, and spatial–temporal networks. In all cases, QpiGNN maintained calibrated coverage, compact interval width, and robustness under structural perturbations and non-exchangeable splits. These results support that the stability assumption is practically reasonable even when dependence is nontrivial.
>
> We will update the manuscript to clearly separate the finite-sample stability guarantee from the asymptotic LLN discussion and to highlight the empirical evidence supporting the bounded-difference assumption on real-world graphs.
>
> > **References**\
> [1] Gama, Fernando, Alejandro Ribeiro, et al. "Stability of graph neural networks to relative perturbations." ICASSP, 2020. \
> [2] Verma, Saurabh, et al. "Stability and generalization of graph convolutional neural networks." KDD, 2019. \
> [3] Sankar, Aravind, et al. "Beyond localized graph neural networks: An attributed motif regularization framework." ICDM, 2020.
>
> ---
> > **W2. Dual-head vs. single-head ablation**
>
> We appreciate the reviewer for stressing the importance of validating the dual-head architecture. We fully agree, and our revised manuscript will highlight this comparison more clearly.
>
> The paper includes an ablation study comparing:
> - the proposed dual-head model,
> - a fixed-margin variant, and
> - a single-head model jointly predicting ($\hat{y}, \hat{d}$).
>
> The results consistently show that the single-head model fails to produce calibrated intervals, while the dual-head design yields accurate coverage and smaller interval widths.
>
> The single-head variant often collapses to underconfident or degenerate interval widths, indicating that the uncertainty and prediction outputs interfere with each other when jointly optimized. In contrast, the dual-head design preserves heteroskedasticity and avoids oversmoothing, producing well-calibrated intervals across all synthetic graph types.
>
> | Ablation | Basic | Gaussian | Uniform | Outlier | Edge | BA | ER | Grid | Tree |
> | -- | -- | -- | -- | -- | --- | -- | -- | -- | -- |
> | Dual Head w/o fixed $\hat{d}$| 0.91/0.31 | 0.92/0.53 | 0.88/0.43 | 0.88/0.47 | 0.93/0.43 | 0.98/0.58 | 0.98/0.45 | 0.98/0.86 | 0.96/0.39 |
> | Dual Head w/ fixed $\hat{d}$ | 0.94/0.20 | 0.51/0.20 | 0.51/0.20 | 0.68/0.20 | 0.97/0.20 | 0.84/0.20 | 0.73/0.20 | 0.38/0.20 | 0.67/0.20 |
> | Single Head | 0.46/0.07 | 0.25/0.09 | 0.25/0.10 | 0.71/0.23 | 0.58/0.07 | 0.78/0.19 | 0.32/0.11 | 0.26/0.14 | 0.06/0.02 |
>
> We will explicitly surface this ablation in the main text and clarify why decoupling prediction and uncertainty is architecturally critical for GNN-based UQ.

---

### Official Review · Reviewer_Cv3a · 2025-11-03

**Soundness:** 2
**Presentation:** 3
**Contribution:** 2
**Rating:** 4
**Confidence:** 3

**Summary:**

The paper proposes Quantile-free Prediction Interval Graph Neural Networks (QpiGNNs), a framework for node-level uncertainty quantification in GNNs from a quantile prediction view. QpiGNNS has a dual-head architecture, one for prediction and one for interval width, which is trained with a quantile-free joint loss that directly optimizes both empirical coverage and interval sharpness using only label supervision. Theoretical analysis shows that the proposed loss is non-convex but guarantees asymptotic coverage under weak dependency assumptions across graph nodes. Empirical results demonstrate that QpiGNN achieves higher coverage accuracy and narrower prediction intervals than existing quantile or conformal GNN baselines across various datasets.

**Strengths:**

1. The paper is well presented and easy to follow, with clear problem formulation and motivation for quantile-free interval learning.
2. The authors conduct comprehensive analysis of the proposed model, including convergence under non-convex optimization, hyperparameter sensitivity, and generalization under structural graph shifts.
3. The method provides a theoretically grounded approach that avoids quantile supervision or resampling, while showing strong empirical robustness under noise and distributional changes.

**Weaknesses:**

1. While the task is interesting from the perspective of quantile prediction, the paper positions itself as an uncertainty quantification work. Therefore, comparisons with other established uncertainty quantification methods for regression—such as evidential regression or Bayesian neural networks—would provide a stronger baseline context.
2. The motivation for designing this framework specifically for graphs is underdeveloped. There is no graph-specific adaptation in the architecture or loss beyond using a GNN backbone, and the theoretical analysis assumes weak dependence among neighboring nodes without addressing how real-world graph correlations may affect coverage guarantees.
3. The paper’s practical application and generalization capability remain somewhat unclear. While results show improved coverage, it is uncertain how well the approach scales to large or dynamically evolving graphs, or how it performs in non-regression graph tasks such as classification or link prediction.

**Questions:**

See weaknesses.

---

> ### Author Response · Authors · 2025-11-21
> **Response to Reviewer Cv3a’s Weaknesses (1/3)**
>
> Thank you for acknowledging the clarity of our formulation and the strength of our quantile-free approach. We appreciate your thoughtful and constructive feedback. We address each concern below and will revise the manuscript accordingly to ensure these points are clearly articulated.
>
> ---
> > **W1. Missing Baseline UQ Methods**
>
> We thank the reviewer for noting that, since our work is positioned as UQ rather than quantile prediction, comparisons with established regression-UQ methods are necessary. We address this point as follows.
>
> ```UQ vs. quantile prediction```\
> Our method is a UQ approach because it directly learns prediction intervals without relying on quantile targets or post-hoc calibration. This enables QpiGNN to overcome limitations of existing quantile-based and conformal methods in GNNs. Another reviewer also noted that UQ for GNNs is underexplored yet practically critical, aligning with our motivation.
>
> ```Regression-UQ baselines```\
> We note that our evaluation includes the UQ baselines most relevant for graph regression—BayesianNN[1], MC Dropout[2], and CF-GNN[3]—which naturally align with message-passing architectures and offer meaningful points of comparison.
>
> Following the reviewer’s suggestion, we additionally include Evidential Regression (ER)[4] as a baseline. While ER predicts a distribution rather than intervals, we adapt it into a stabilized interval-prediction form for fairness.
>
> Across all datasets, QpiGNN achieves substantially tighter intervals while maintaining competitive coverage, demonstrating its effectiveness relative to existing UQ approaches. We will clarify this baseline rationale and present ER results in the revised manuscript.
>
> | Model                 | Education | Election  | Income    | Unemploy. | Twitch    | Chamelon  | Crocodile | Squirrel  | Anaheim   | Chicago   |
> | --------------------- | --------- | --------- | --------- | --------- | --------- | --------- | --------- | --------- | --------- | --------- |
> | BayesianNN            | 1.00/2.96 | 1.00/2.98 | 1.00/2.97 | 1.00/2.98 | 1.00/3.07 | 1.00/2.95 | 1.00/3.07 | 1.00/2.97 | 1.00/2.94 | 1.00/2.99 |
> | MC Dropout            | 0.40/0.11 | 0.48/0.18 | 0.45/0.09 | 0.41/0.09 | 0.91/0.15 | 0.47/0.02 | 0.46/0.01 | 0.31/0.02 | 0.50/0.11 | 0.34/0.07 |
> | CF-GNN                | 0.90/3.10 | 0.91/0.94 | 0.91/2.92 | 0.89/2.61 | 0.89/2.34 | \-        | \-        | \-        | 0.90/2.82 | 0.91/2.26 |
> | ER | 1.00/4.37 | 1.00/6.90 | 1.00/3.12 | 1.00/4.80 | 0.99/1.33 | 0.97/1.08 | 1.00/1.56 | 0.97/0.80 | 1.00/2.09 | 1.00/3.06 |
> | QpiGNN(optimal)       | 0.99/0.59 | 0.98/0.77 | 0.99/0.44 | 1.00/0.73 | 0.94/0.36 | 0.96/0.23 | 0.97/0.16 | 0.96/0.18 | 0.93/0.40 | 0.98/0.36 |
>
> ---
> > **References**\
> [1] Kendall, Alex, et al. "What uncertainties do we need in bayesian deep learning for computer vision?." In NeurIPS, 2017.\
> [2] Gal, Yarin, et al. "Dropout as a bayesian approximation: Representing model uncertainty in deep learning." In ICML , 2016.\
> [3] Huang, Kexin, et al. "Uncertainty quantification over graph with conformalized graph neural networks." In NeurIPS, 2023.\
> [4] Amini, Alexander, et al. "Deep evidential regression." In NeurIPS ,2020.

---

> ### Author Response · Authors · 2025-11-21
> **Response to Reviewer Cv3a’s Weaknesses (2/3)**
>
> > **W2. Insufficient Graph-Specific Motivation**
>
> We appreciate the reviewer’s point that our motivation for a graph-specific UQ framework should be more explicit. We will strengthen this discussion by highlighting two central issues:
>
> ```Limitations of CP-based graph UQ```\
> While CF-GNN is an important UQ approach for graph data, it is fundamentally built upon CP and therefore relies on the assumption of exchangeability between the calibration and test sets—an assumption structurally fragile in graphs due to message-passing dependencies.
>
> Our experiments intentionally break exchangeability (degree splits, community splits), and QpiGNN maintains stable coverage. This motivates a UQ framework that is inherently graph-aware and exchangeability-free.
>
> | Split Type | PICP | MPIW |
> | ---------- | -------- | ----- |
> | Random     | 0.98–1.00   | 0.40–0.97    |
> | Degree     | 0.94–1.00   | 0.44–0.87    |
> | Community  | 0.95–1.00   | 0.46–0.86    |
>
> ```Why existing QR methods fail on graphs```\
> Existing QR methods such as SQR and RQR were developed for i.i.d. regression with MLPs, and they do not transfer reliably to graph settings. Message passing introduces structural dependencies that fundamentally change how QR losses behave.
>
> - **Single-head coupling problem:**
> When prediction and uncertainty are generated from a single head, message passing entangles the two signals. This often causes interval collapse or over-smoothed intervals, especially in heterophilic graphs where node-wise heteroskedasticity must be preserved. For RQR, we even observed that an additional ordering penalty was required just to maintain valid interval ordering.
> - **Node-wise loss is incompatible with graph dependence:**
> Standard QR losses are entirely node-wise and assume independent samples. Under message passing, this leads to unstable gradients because coverage and width interact within a single objective. As a result, training frequently becomes unstable or collapses when applied to GNNs.
>
> These issues are consistently observed in our experiments: while QpiGNN maintains stable coverage even under structural perturbations, existing QR methods either collapse or produce intervals that are excessively wide.
>
> | Perturbation Type | QpiGNN                    | SQR-GNN               | RQR$^{adj.}$-GNN      |
> | ----------------- | ------------------------- | --------------------- | --------------------- |
> | Feature Noise $\uparrow$   | 0.89–0.92 / 0.66–0.83 | 0.65–0.76 / 0.69–0.81 | 0.84–0.85 / 0.76–0.78 |
> | Target Noise $\uparrow$    | 0.96–1.00 / 0.44–1.05 | 0.49–0.98 / 0.52–0.99 | 0.95–0.98 / 0.87–1.32 |
> | Edge Dropout $\uparrow$    | 0.84–0.91 / 0.21–0.23 | 0.53–0.82 / 0.44–0.50 | 0.86–0.89 / 0.80–0.84 |
>
> ```Graph-Adaptive Design in QpiGNN```\
> To address the limitations discussed above, QpiGNN combines two graph-adaptive design components.
>
> - **Dual-head architecture:**
> QpiGNN separates prediction and uncertainty into two distinct message-passing pathways, preventing the entanglement that often occurs when both signals are propagated through a single head. This design alleviates oversmoothing and preserves node-wise heteroskedasticity even on heterophilic or noisy graphs. Our ablation studies further confirm that the dual-head architecture provides substantially more stable coverage and interval widths compared to single-head variants.
>
> - **Joint loss formulation:**
> Whereas RQR entangles coverage and width within a single conditional loss—leading to instability under graph dependence—QpiGNN explicitly decomposes the objective into a coverage term, a width term, and a violation term. The coverage term drives the model toward the target coverage level, the width term encourages compact intervals, and the violation term provides fine-grained correction for missed intervals. This disentangled structure mitigates gradient interference caused by local correlation patterns induced by message passing, enabling more stable learning of node-level uncertainty.
>
> Together, the dual-head architecture and the disentangled loss constitute graph-adaptive design choices that directly address the unique structural dependencies present in graph data and enable reliable uncertainty quantification under these conditions.

---

> ### Author Response · Authors · 2025-11-21
> **Response to Reviewer Cv3a’s Weaknesses (3/3)**
>
> > **W3. Unclear Practical Scope and Generalization**
>
> We appreciate the reviewer’s point that the practical applicability and generalization of our approach—particularly to large or dynamically evolving graphs, as well as to non-regression tasks like classification or link prediction—should be clarified. We address this concern as follows.
>
> ```Scalability to large or dynamic graphs```\
> While our experiments do not directly include very large or dynamic graphs, several factors support QpiGNN’s scalability in such settings.
>
> - Our real-world datasets already contain graphs with up to 12K nodes and over 200K edges (e.g., Crocodile, Squirrel), and QpiGNN shows training time, memory usage, and parameter counts comparable to BayesianNN and MC Dropout. This indicates that the method operates efficiently without additional computational overhead even on sizable graphs.
> - Appendix I shows that QpiGNN maintains the same $\mathcal{O}(E d + N d^2)$ complexity as GraphSAGE, since it does not require repeated forward passes or auxiliary GNN modules. This yields the same near-linear scalability as standard message-passing GNNs, suggesting no structural barriers to scaling to larger graphs.
>
> | Dataset     | #Nodes | #Edges  | Model      | Train Time (sec) | CPU (MB)  | Params (K) |
> |-------------|--------|---------|------------|------------------|-----------|------------|
> | Crocodile | 11,631 | 180,020 | BayesianNN | 21.34            | 7601.32   | 1695.87    |
> |             |        |         | MC Dropout | 18.37            | 7601.61   | 1695.81    |
> |             |        |         | QpiGNN | 14.87        | 7601.44 | 1695.87 |
> | Squirrel  | 5,201  | 217,073 | BayesianNN | 5.36             | 1966.30   | 411.39     |
> |             |        |         | MC Dropout | 2.92             | 1966.29   | 411.33     |
> |             |        |         | QpiGNN* | 5.28         | 1966.31 | 411.39  |
>
> Although dynamic graph UQ remains challenging and relatively unexplored, QpiGNN’s message-passing structure is compatible with temporal GNNs such as TGAT, making extension to evolving graphs feasible. We appreciate the reviewer for highlighting this important future direction.
>
> ```Generalization to other graph tasks```\
> QpiGNN naturally generalizes beyond node regression, as discussed in the task extensions subsection. The dual-head quantile-free design applies directly to other graph tasks by reusing the same prediction–width mechanism on task-specific embeddings, requiring no architectural changes beyond substituting the base encoder.
>
> - For graph-level regression, pooled graph embeddings $\mathbf{h}\_G = \rho(\mathrm{GNN}(\mathbf{X}, \mathcal{E}))$ are mapped to prediction and width via $\hat{y}\_G = \mathbf{W}\_{\mathrm{pred}}\mathbf{h}\_G + b\_{\mathrm{pred}}$ and $\hat{d}\_G = \mathrm{Softplus}(\mathbf{W}\_{\mathrm{diff}}\mathbf{h}\_G + b\_{\mathrm{diff}})$, yielding intervals $[\hat{y}\_G - \hat{d}\_G,\ \hat{y}\_G + \hat{d}\_G]$.
>
> - For link prediction, edge embeddings $\mathbf{h}\_{uv} = \phi([\mathbf{h}\_u \Vert \mathbf{h}\_v \Vert \mathbf{h}\_u \odot \mathbf{h}\_v])$ are processed analogously to produce calibrated intervals $\hat{y}\_{uv} \pm \hat{d}\_{uv}$.
>
> While QpiGNN is designed for regression, the same uncertainty mechanism can be adapted to classification—for example, by forming predictive sets or margin-based confidence intervals using class-logit outputs. These extensions demonstrate that QpiGNN is task-agnostic and not restricted to a single problem setting.

---

### Author Response · Authors · 2025-12-03
**Author Final Remarks**

We thank the reviewers for their thoughtful feedback. Below, we briefly summarize the motivation and main contributions of our work and describe how we have addressed the key concerns raised during the review process to improve the manuscript.

---
### **Motivation & Design Rationale**
To clarify our motivation, we summarize the key points below as reflected in the revised manuscript. We hope that this framework will further support future QR-based UQ research in GNNs.

**```Limitations of Conformal Prediction (CP) in graph settings```**
- Message passing in graphs breaks exchangeability, and the post-hoc nature of CP prevents it from modeling graph dependencies during training, leading to fundamental limitations for CP-based methods.

**```Advantages of Quantile Regression (QR) for uncertainty estimation```**
- QR estimates conditional quantiles directly during training, eliminating the need for post-hoc calibration and enabling flexible modeling of asymmetry and heterogeneity—both of which are important for graph-structured data.

**```Absence of QR-based GNN-UQ methods and limitations of existing approaches```**
- QR-based UQ for GNNs has not been systematically explored, so we implemented the QR baselines.
- Although these QR models perform well with MLPs, they became unstable under graph dependencies, and RQR showed persistent quantile crossing. This motivated the adjusted RQR$^{adj.}$ which adds an ordering penalty for fair comparison.

**```Necessity and role of the QpiGNN design```**
- To address these limitations, we introduce QpiGNN, which uses a dual-head architecture and a joint loss tailored to graph settings.
- By separating prediction and uncertainty, QpiGNN avoids representational interference and reduces quantile instability under graph dependencies.

---
### **Contributions & Strengths**
Below are the main contributions of our work, several of which were also noted positively by reviewers.

**```S1. Clarity```**
- The paper clearly presents the problem setup and the motivation for the quantile-free formulation, and it differentiates our approach from related work—particularly QpiGNN—in a transparent manner.

**```S2. Motivation```**
- We highlight UQ in GNNs as an important yet understudied problem, identify the structural limits of quantile-based and conformal methods, and explain why a quantile-free approach naturally fits graph-structured data.

**```S3. Novelty```**
- Our dual-head quantile-free architecture separates prediction and uncertainty estimation, mitigating non-crossing issues and improving GNN stability, while providing a simple and practical UQ method that operates without quantile supervision or resampling.

**```S4. Theory```**
- We ground our method theoretically by interpreting the joint loss via Lagrangian relaxation, clarifying the coverage–width trade-off, and providing convergence, sensitivity, and both asymptotic and finite-sample guarantees under realistic assumptions.

**```S5. Experiments```**
- We evaluate QpiGNN across diverse synthetic and real datasets, showing consistent performance under noise and various distribution or graph-structure shifts. Ablations on architecture and sensitivity further support its robustness.

---
### **Reviewer Concerns & Responses**
We summarize the main concerns raised by the reviewers and describe how we addressed them in the revised manuscript.

**```C1. Distinction from RQR```**
- QpiGNN is not a simple RQR variant; it addresses key sources of instability in QR-based UQ on graphs, including residual correlation and loss of exchangeability.
- Its dual-head, disentangled design is fundamentally different from RQR’s residual-modeling approach. Implementing RQR revealed quantile crossing, motivating the adjusted RQR$^{adj.}$-GNN. The revision clarifies these differences.

**```C2. Theoretical Validity under GNN Dependencies```**
- Our theory avoids independence assumptions and relies only on a bounded-difference condition for message passing.
- It aligns with McDiarmid-type guarantees, and we show coverage stability of $1/N + \delta_G$ across diverse graphs.
- We note that the bounded-difference term may vary in dense or high-degree graphs.

**```C3. Experimental Enhancements```**
- The revision highlights the existing dual-head ablation, Bayesian baseline, and real-data ablation more clearly.
- We also added L1–L2 penalty comparisons, an Evidential Regression baseline, and analyses under varying target coverage.

**```C4. Scalability and Generalization```**
- QpiGNN retains the linear scalability of standard message-passing GNNs and runs stably on large graphs.
- Its dual-head quantile-free design is task-agnostic and extends to other GNN tasks by replacing only the encoder.
- We also note dynamic graph UQ as a future direction.

---
We thank the reviewers once again for their valuable feedback and hope that our contributions and revisions have been clearly and satisfactorily addressed.

Best regards,

Authors

---

### Meta-Review · Area_Chair_aHwb · 2025-12-23

**Summary:**

This paper proposes a node-level uncertainty quantification method in GNNs.

Strengths:
(1) theoretically grounded UQ method for GNN. (2) clear motivation and problem formulation. (3) empirical evaluation demonstrating the effectiveness of the proposed method.

Weaknesses:
(1) missing baselines for UQ. (2) unclear practical application and generalization capability. (3) lack of justification of certain assumptions such as weak dependence. (4) incremental novelty of the proposed method in the context of the existing work. (5) lack of justification of the new loss (Eq. (5)).

**Reviewer Concerns:**

Some concerns regarding the empirical evaluations were addressed.

**Reviewer Scores:**

It is unlikely that the reviewers will change their scores. In particular, nVMe (who gave 2) mentioned that s/he was not convinced by the authors’ response.

---

### Decision · Program_Chairs · 2026-01-26

Reject